# Inhalable cardiac targeting peptide modified nanomedicine prevents pressure overload heart failure in male mice

Haobo Weng[1,2,3,7], Weijuan Zou[2,4,7], Fangyan Tian[1,3,5,7], Huilin Xie[1,3], Ao Liu [1,3], Wen Liu[1], Yu Liu[1], Nianwei Zhou[1], Xiaojun Cai [2,4], Jianrong Wu [2,4] ✉, Yuanyi Zheng [2,4] ✉ & Xianhong Shu[1,3,6] ✉

Heart failure causes considerable morbidity and mortality worldwide. Clinically applied drugs for the treatment of heart failure are still severely limited by poor delivery efficiency to the heart and off-target consumption. Inspired by the high heart delivery efficiency of inhaled drugs, we present an inhalable cardiac-targeting peptide (CTP)-modified calcium phosphate (CaP) nanoparticle for the delivery of TP-10, a selective inhibitor of PDE10A. The CTP modification significantly promotes cardiomyocyte and fibroblast targeting during the pathological state of heart failure in male mice. TP-10 is subsequently released from TP-10@CaP-CTP and effectively attenuates cardiac remodelling and improved cardiac function. In view of these results, a low dosage (2.5 mg/kg/2 days) of inhaled medication exerted good therapeutic effects without causing severe lung injury after long-term treatment. In addition, the mechanism underlying the amelioration of heart failure is investigated, and the results reveal that the therapeutic effects of this system on cardiomyocytes and cardiac fibroblasts are mainly mediated through the cAMP/AMPK and cGMP/PKG signalling pathways. By demonstrating the targeting capacity of CTP and verifying the biosafety of inhalable CaP nanoparticles in the lung, this work provides a perspective for exploring myocardium-targeted therapy and presents a promising clinical strategy for the long-term management of heart failure.

Heart failure is the leading cause of morbidity and mortality in humans, affecting more than 26 million people worldwide[1]. As the world's population ages, the global burden of heart failure is inevitably increasing, and it is expected to substantially increase in the future. Recently, remarkable advances have been made during the exploration of the molecular biology and targets involved in heart failure. However, many patients still have a poor prognosis and low quality of life[1,2]. Additionally, numerous preclinical studies based on novel therapeutic targets have failed to meet their primary efficacy endpoints[3–5], with low cardiac specificity and delivery efficacy, which

[1]Department of Echocardiography, Shanghai Institute of Medical Imaging, Zhongshan Hospital, Fudan University, Shanghai, PR China. [2]Shanghai Key Laboratory of Neuro-Ultrasound for Diagnosis and Treatment, Shanghai Sixth People's Hospital Affiliated to Shanghai Jiao Tong University School of Medicine, Shanghai, PR China. [3]Department of Cardiology, Shanghai Institute of Cardiovascular Disease, Zhongshan Hospital, Fudan University, Shanghai, PR China. [4]Department of Ultrasound in Medicine, Shanghai Institute of Ultrasound in Medicine, Shanghai Sixth People's Hospital Affiliated to Shanghai Jiao Tong University School of Medicine, Shanghai, PR China. [5]Department of Ultrasound Medicine, The Affiliated Hospital of Guizhou Medical University, Guiyang, China. [6]Department of Ultrasound in Medicine, Shanghai Xuhui District Central Hospital, Shanghai, PR China. [7]These authors contributed equally: Haobo Weng, Weijuan Zou, Fangyan Tian. ✉e-mail: wujianrong028@shsmu.edu.cn; zhengyuanyi@sjtu.edu.cn; shu.xianhong@zs-hospital.sh.cn

lead to the low efficacy of available therapeutic agents, being amongst the critical explanations[6]. Consequently, it is imperative to explore alternative delivery strategies to prevent heart failure.

The oral route is the most widely used route of administration for daily heart failure therapies because it is non-invasive and promotes patient compliance with long-term medication[6–8]. However, several physical barriers (e.g., the intestinal epithelium, intestinal mucus, intestinal microorganisms and capillary barrier) limit the efficient transport of drugs across the intestinal membrane. In addition, chemical substances in the gastrointestinal tract (including gastric juice, proteases, digestive juice, or bile salts) and metabolic reactions in the liver might prevent therapeutics from extravasating the target tissues[8]. These shortcomings render oral delivery unreliable for the long-term treatment of heart failure. Other invasive delivery strategies, such as intravenous (I.V.) injection, surgical approaches or catheter-based approaches, partly improve the efficiency of drug delivery to the heart but may lead to poor patient compliance and adherence, and they are usually accompanied by the inevitable complications of myocardial injury[6,7]. However, an encouraging study by Miragoli et al.

demonstrated the feasibility of using inhalation as an approach for delivering therapeutic peptide-loaded calcium phosphate (CaP) nanoparticles to diseased hearts[9]. Due to their substantial surface area and good epithelial permeability and the abundant blood flow in the lungs, small particles that are deposited in the lungs are rapidly absorbed into the systemic circulation at high levels[10,11]. Consequently, inhalation allows high drug accumulation in the heart within a short time[9]. Recently, Modica et al. verified the application of these CaP nanoparticles as carriers of therapeutic microRNAs for heart failure[12]. However, the potential lung complications of chronic regimens, which may limit the wide application of drug administration via inhalation clinically[11], still need to be explored[9,11–14]. According to previous studies, the long-term use of inhalable insulin leads to wheezing and bronchoconstriction[15] and results in gradually increased loss of lung function[11,16,17]. In long-term clinical trials, lung cancer was reported in some patients who received inhalable insulin[18,19]. In addition, inhalable levodopa has been utilized for the treatment of Parkinson's disease, but its clinical use is limited due to the increased risk of bronchospasm[20]. Thus, avoiding lung complications is crucial for

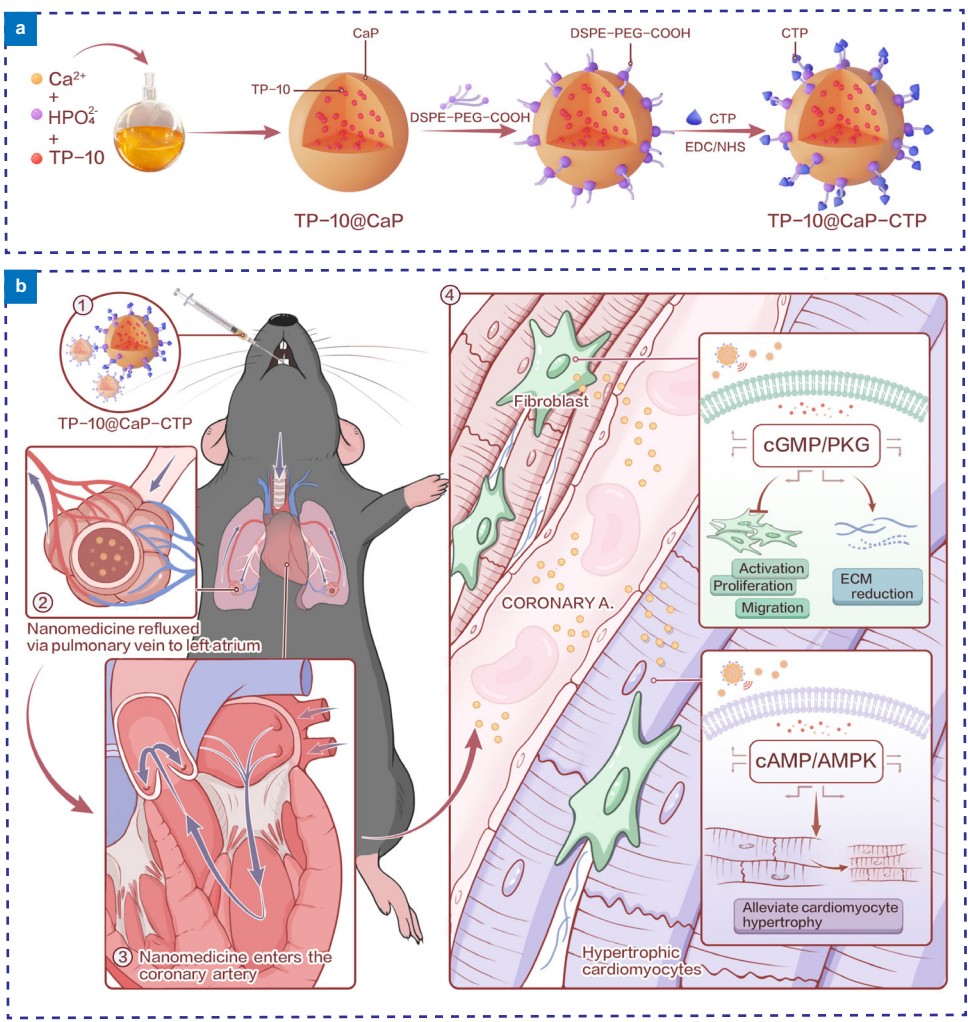

**Fig. 1 | Scheme of the heart failure prevention strategy involving the inhalation of TP-10@CaP-CTP nanoparticles. a** Fabrication of TP-10@CaP-CTP. CaP was first loaded with TP-10 (TP-10@CaP) through a biomineralization-inspired strategy, followed by incubation with DSPE-PEG-COOH to obtain TP-10@CaP-DSPE-PEG-COOH. Then, CTPs was connected to TP-10@CaP-DSPE-PEG-COOH through an EDC/NHS coupling reaction. **b** Inhalation delivery strategy for heart failure. 1) TP-10@CaP-CTP nanoparticles were administered through via inhalation with a high-pressure microsprayer aerosolizer. 2) TP-10@CaP-CTP nanoparticles crossed the

alveolar capillaries and were absorbed into systematic circulation. 3) Once they entered the systemic circulation, the TP-10@CaP-CTP nanoparticles first arrived in the left atrium and were then distributed in the myocardium through the coronary arteries and cardiac microvasculature. 4) TP-10@CaP-CTP nanoparticles specifically target the failing myocardium. The released TP-10 effectively attenuated cardiac hypertrophy and inhibited cardiac fibroblast proliferation, migration, and activation, as well as ECM synthesis.

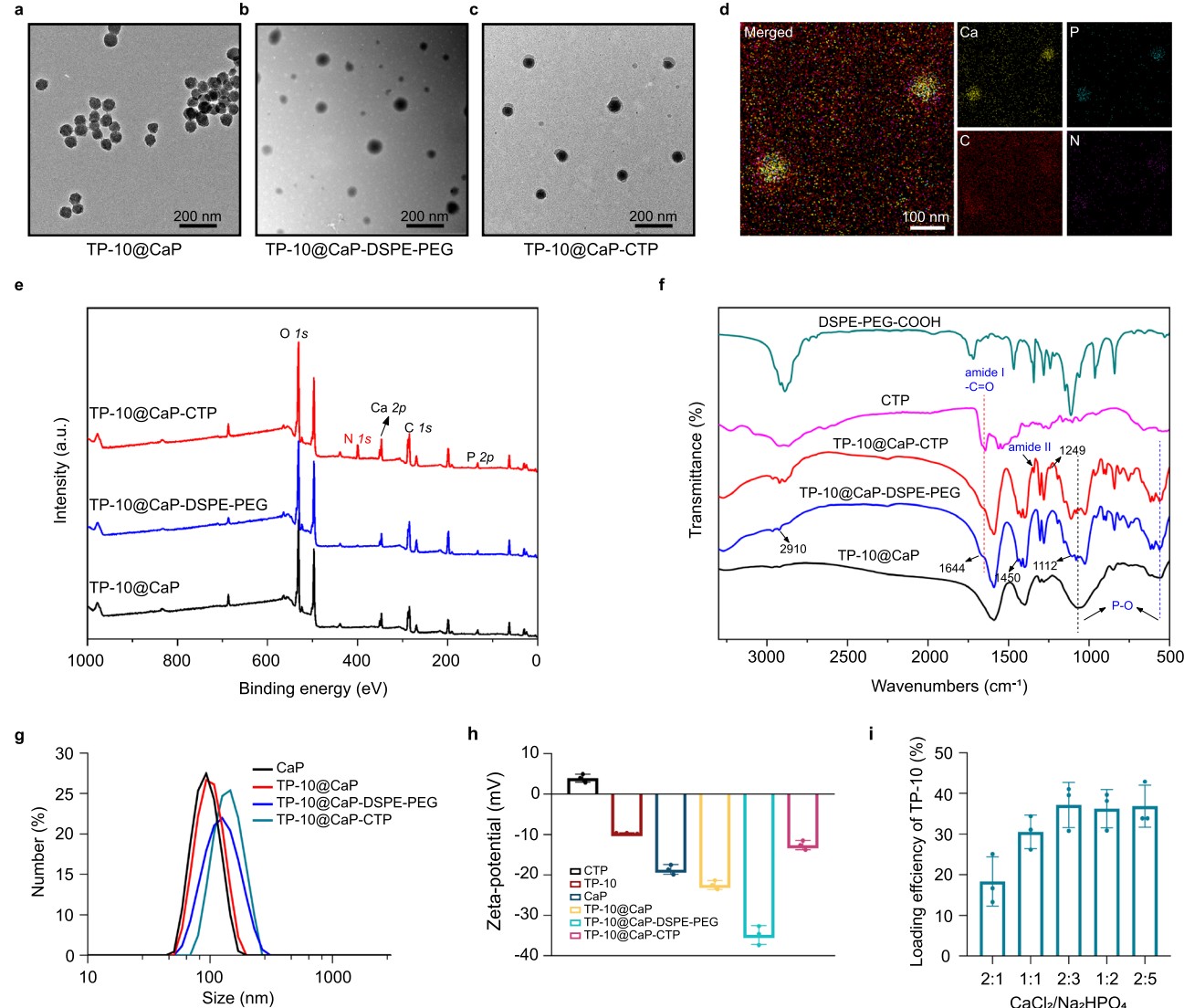

**Fig. 2 | Characterization of different nanoparticles prepared in each step.** TEM image of different intermediate products, including **a**, TP-10@CaP, **b**, TP-10@CaP-DSPE-PEG, and **c**, TP-10@CaP-CTP nanoparticles collected at 12 h after preparation. **d** Elemental mappings of Ca, P, C, and N in TP-10@CaP-CTP. **e** XPS and **f** FT-IR spectra of different prepared nanoparticles. **g** Hydrodynamic diameter of different prepared nanoparticles in water (0.4 mg/mL). **h** Zeta potentials of different prepared nanoparticles in PBS at pH 7.4. $n = 3$ independent samples. **i** TP-10 loading in TP-10@CaP-CTP analysis with different ratio of CaCl$_2$/Na$_2$HPO$_4$. $n = 3$ independent samples. In **a**–**d**, experiments were performed three times (**b**, **c**), with similar results and a representative image is shown. The results are presented as the mean ± SD. Source data are provided as a Source Data file.

fulfilling the requirement of efficient drug delivery via inhalation. Recent studies have shown that a cardiac-targeting peptide (CTP) with the sequence CSTSMLKAC filtered by phage display technology exhibits remarkable selectivity for the pathological myocardium[21,22]. Although the potential binding mechanism remains unclear, drug delivery carriers modified with CTP displayed clearly increased accumulation of therapeutic agents in the pathological myocardium[21,22]. Therefore, we hypothesized that the CTP modification strategy would enable heart-targeted drug delivery to improve the accumulation of therapeutic agents in the heart rather than administering a high dose or multiple doses of each inhaled medication, thus reducing the required therapeutic dose to be administered.

Here, a CTP-modified biodegradable CaP nanoparticle loaded with TP-10 (denoted TP-10@CaP-CTP) was fabricated and delivered to the myocardium via inhalation for the long-term management of chronic heart failure (Fig. 1). TP-10 is a selective inhibitor of PDE10A that has been shown to be involved in the pathological process of heart failure[23]. The use of TP-10 in a chronic heart failure mouse model was

indicated to attenuate cardiomyocyte (CM) hypertrophy and inhibit cardiac fibroblast (CF) activation, proliferation and migration and extracellular matrix (ECM) synthesis, thereby reversing the established cardiac remodelling and improving cardiac function[23]. Compared with other PDE inhibitors, TP-10 affects both CMs and CFs by regulating intracellular cAMP and cGMP levels[23]. Furthermore, PED10A was proven to be a safe therapeutic target in phase II clinical trials[23]. These dual therapeutic effects and safety properties make TP-10 a promising small molecule to prevent the pathogenesis of heart failure.

In this study, we showed that CTP modification combined with inhalation delivery strikingly promoted TP-10 accumulation in the heart. The TP-10@CaP-CTP nanoparticles also regulated intracellular cAMP and cGMP levels in both CMs and CFs. Exploiting the heart-targeting capacity of CTP, the inhaled TP-10@CaP-CTP nanoparticles effectively attenuated pathological cardiac hypertrophy and cardiac fibrosis remodelling, thereby improving cardiac function in a pressure overload-induced heart failure mouse model. We elucidated that TP-10@CaP-CTP might contribute to CM pathological hypertrophy via

cAMP/AMPK signalling and inhibit CF activation in a cGMP/PKG-dependent manner. Moreover, the long-term safety profile of the nanoparticles in the lungs was evaluated by observing lung injury and fibrosis during 18 weeks of inhalation therapy. Our study highlights that using the engineered TP-10@CaP-CTP nanoparticles modified with CTP and administered via the cardiopulmonary circulation could be a promising approach for the long-term management of chronic heart failure.

## Results

### Fabrication and characterization of the TP-10@CaP-CTP nanoparticles

TP-10@CaP-CTP nanoparticles were synthesized based on the procedures shown in Fig. 1a. In brief, TP-10@CaP nanoparticles were prepared via a biomineralization-inspired strategy by mixing TP-10 with $CaCl_2$ and $Na_2HPO_4$ according to previous studies by Miragoli et al.[9,12,24]. The transmission electron microscopy (TEM) image of TP-10@CaP (Fig. 2a) showed that these nanoparticles had a homogeneous and spherical morphology with a diameter of approximately 73 nm. Then, the TP-10@CaP nanoparticles were modified with 1,2-distearoyl-sn-glycero-3-phosphoethanolamine conjugated polyethylene glycol acid (DSPE-PEG-COOH) to obtain carboxyl-functionalized CaP (TP-10@CaP-DSPE-PEG). In this step, the phosphate head of DSPE coordinated with the calcium ions of the TP-10@CaP nanoparticles, forming a lipid layer and introducing carboxyl groups for further modification. Unlike TP-10@CaP, TP-10@CaP-DSPE-PEG showed a smooth organic surface by TEM (Fig. 2b), indicating the formation of a lipid layer. Afterwards, TP-10@CaP-DSPE-PEG was reacted with the amino group of CTP (CSTSMLKAC, Supplementary Fig. 1) by an EDC/NHS coupling reaction to form the final product TP-10@CaP-CTP. The CTP with a relative molecular weight of 943 g/mol was used after identification by high-performance liquid chromatography (HPLC) and mass spectrometry (MS) analysis. Similarly, the as-obtained TP-10@CaP-CTP nanoparticles were also monodispersed and exhibited a uniform spherical topology with a larger particle size of $85 \pm 3.7$ nm (Fig. 2c). Furthermore, elemental mapping based on high-angle annular dark-field scanning TEM showed that C, N, P, and Ca were homogeneously distributed in the TP-10@CaP-CTP nanoparticles (Fig. 2d). The X-ray photoelectron spectroscopy (XPS) elemental analysis showed a typical N$1s$ peak in TP-10@CaP-CTP due to the presence of nitrogen in CTP, while there were no obvious N signals in the spectra of TP-10@CaP-DSPE-PEG and the original TP-10@CaP nanoparticles (Fig. 2e). Moreover, a significant increase in the C content was observed in TP-10@CaP-CTP and TP-10@CaP-DSPE-PEG compared to that in TP-10@CaP, suggesting the conjugation of DSPE and CTP, as further proven by energy dispersive X-ray (EDX) spectroscopy (Supplementary Fig. 2). Fourier transform infrared (FT-IR) spectra were obtained for the particles prepared in each step (Fig. 2f). Characteristic peaks at 1038 and 602 $cm^{-1}$ were observed in the spectra of both CaP-based nanoparticles, corresponding to the $v_3$ and $v_4$ P-O vibrations of $PO_4^{3-}$ in CaP. Compared with the spectrum of TP-10@CaP, the TP-10@CaP-DSPE-PEG spectrum showed new peaks corresponding to PEG ($v_{C-H}$, 2910 $cm^{-1}$; $\delta_{C-H}$, 1450 $cm^{-1}$; and $v_{C-O}$, 1112 $cm^{-1}$) and -COOH ($v_{C=O}$, 1644 $cm^{-1}$), indicating that DSPE-PEG was successfully grafted onto the surface of TP-10@CaP. After CTP modification, there was a new band at 1249 $cm^{-1}$ ($v_{C-N}$), together with distinct stretching vibrations at 1644 and 1343 $cm^{-1}$ (C = O, CTP), belonging to the O = C-N-H group connecting the PEG and CTP moieties, confirming the successful combination of TP-10@CaP-DSPE-PEG with CTP by the EDC/NHS reaction. After surface modification, the hydrodynamic diameter of the TP-10@CaP-CTP nanoparticles was approximately 140 nm, which was larger than that of the TP-10@CaP (92.9 nm) and TP-10@CaP-DSPE-PEG (125.6 nm) nanoparticles, as indicated by dynamic light scattering (DLS) measurements (Fig. 2g). Meanwhile, it can be observed that a broad absorption band at approximately 602–618 $cm^{-1}$, indicating the

amorphous of TP-10@CaP. FT-IR spectra of of TP-10@CaP-DSPE-PEG and TP-10@CaP-CTP exhibited two bands at 602 and 618 $cm^{-1}$ ($v_4$ P-O vibrations of $PO_4^{3-}$), which indicated that amorphous TP-10@CaP turned into a more crystalline state, following a surface modification process. The amorphous and crystalline states of these CaP-based materials were also confirmed by the measurement of splitting factors (Supplementary Fig. 3). Additionally, the TP-10@CaP and bare CaP nanoparticles showed similar surface charges of approximately −20 mV due to the negligible impact of drug encapsulation on the surface charge. However, modification of the CaP nanoparticles with DSPE-PEG-COOH changed the zeta potential to −34.8 ± 1.9 mV due to the negative charge of the carboxyl groups, confirming the formation of the lipid layer. In contrast, due to the presence of primary amino groups on CTP, the zeta potential of TP-10@CaP-CTP decreased to −12.6 ± 1.2 mV after the formation of amide bonds (Fig. 2h). These results demonstrated that the CaP nanoparticles could be efficiently loaded with TP-10 and modified with DSPE-PEG-COOH and CTP on the surface. The obtained TP-10@CaP-CTP nanoparticles were well dispersed in different media (water, PBS, and DMEM) and showed a stable mean hydrodynamic diameter and surface charge during 7 days of incubation, indicating good colloidal stability (Supplementary Fig. 4).

To evaluate TP-10 loading, the total amount of TP-10 in the TP-10@CaP-CTP nanoparticles was quantified. First, the amount of TP-10 in TP-10@CaP was determined to be 57.6% by HPLC−MS/MS, the loading capacity of TP-10 in TP-10@CaP-CTP was determined to be 37.2%. It should be noted that we synthesized TP-10@CaP with a drug loading of about 37.2% for subsequent experiments, which is mainly to maintain consistency with TP-10@CaP-CTP at the dosage level of monolithic structure. Moreover, the optimized encapsulation efficiency (EE) of TP-10 was calculated to be 51.4 ± 3.1% with a $CaCl_2$/$Na_2HPO_4$ ratio of 2:3 (Fig. 2i). The CTP bonding efficiency and EE were evaluated by a similar procedure and finally determined by the BCA assay, which showed the high binding of 22.3% (w/w%) and an EE% of 62.9 ± 0.9% (Supplementary Fig. 5).

Then, we further evaluated the capacity of the TP-10@CaP-CTP nanoparticles to release TP−10 under different pH conditions (Supplementary Fig. 6). As expected, the release of TP-10 from the CaP structure was dependent on the pH. The release of TP-10 from both TP−10@CaP and TP−10@CaP-CTP reached equilibrium within 12 h with <10% release at pH 7.4, indicating the high stability of CaP under neutral conditions. The percentage of TP−10 released from TP−10@CaP-CTP dramatically increased to 46.2% at pH 6.5 and 67.7% at pH 5.5 within 12 h. In contrast, the amount of TP-10 released from TP−10@CaP over a span of 12 h increased to ≈49.3% and ≈72.5% at pH 6.5 and 5.5, respectively. It follows that surface modification with DSPE-PEG and CTP may slow the degradation rate of the CaP structure and subsequent drug release. As the microenvironment of the failing myocardium during heart failure is mildly acidic[25,26], the pH-responsive release of TP−10 will benefit the application of TP-10@CaP-CTP nanoparticles for heart failure therapy.

### In vivo accumulation of TP-10@CaP-CTP in the heart after nebulized inhalation

After the nanoparticles were generated and characterized, the feasibility of delivering TP-10@CaP-CTP by inhalation was systematically evaluated to ensure that TP-10 was accurately delivered to the heart. To this end, Cy5.5-labelled TP−10@CaP and TP-10@CaP-CTP nanoparticles were prepared. The stability of the Cy5.5-labelled TP-10@CaP-CTP or TP-10@CaP nanoparticles was first evaluated by monitoring the UV–Vis-NIR spectra (Supplementary Fig. 7). After dispersion in PBS for 7 days, no obvious change in the absorption peak of Cy5.5 was observed in the UV–Vis-NIR spectrum, indicating that the Cy5.5 label was stable. Then, the same dose of these two nanoparticle dispersions were given to sham mice or mice that were subjected to transverse aortic constriction (TAC) intravenously or via inhalation. The mean

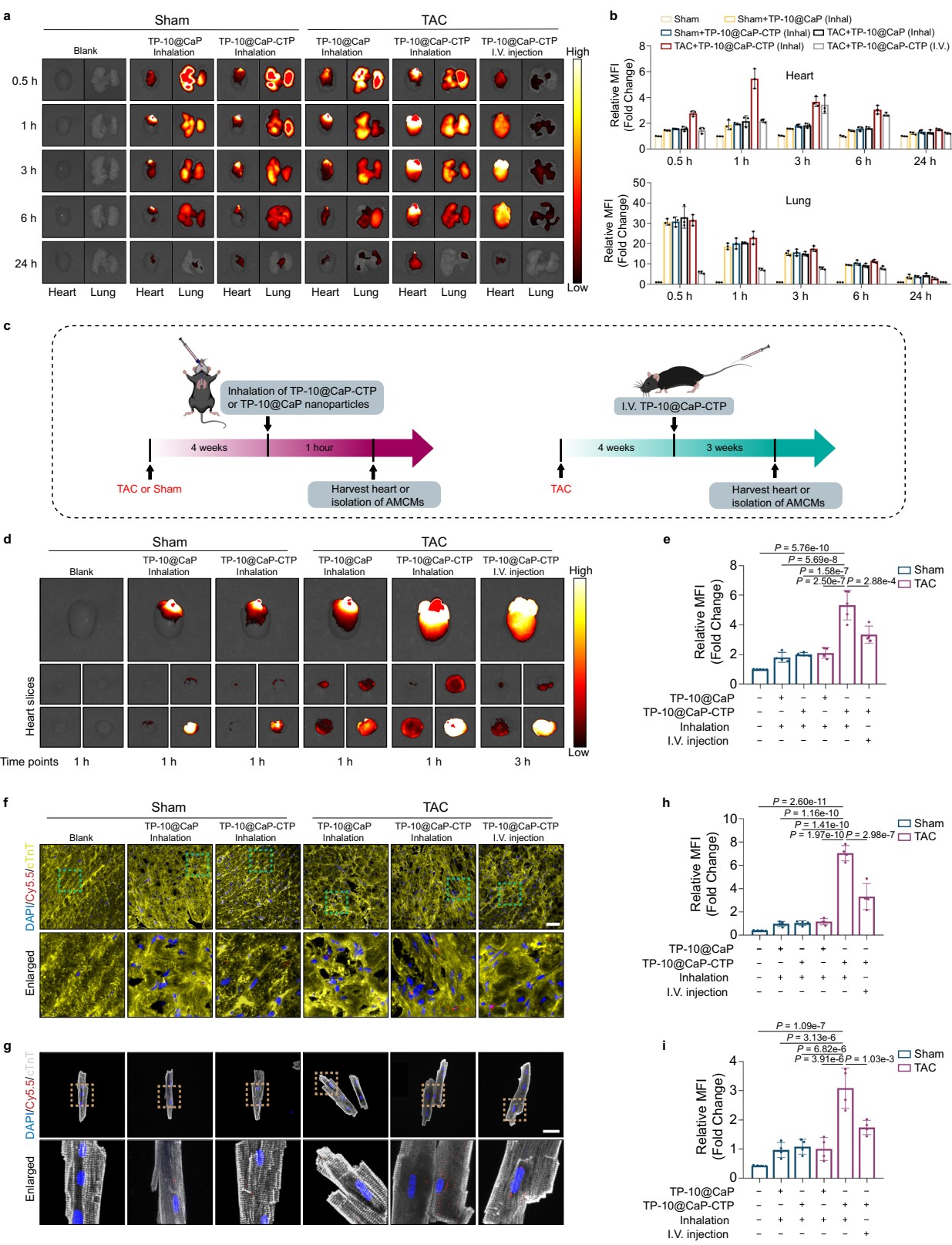

fluorescence intensity, which reflects the accumulation of the different formulations, was determined at different time points by using an in vivo imaging system. Ex vivo fluorescence imaging revealed that the myocardial accumulation of the nanoparticles over time was accompanied by a decaying signal in the lungs, suggesting the passage of the nanoparticles across the pulmonary barrier (Fig. 3a). In addition, TP−10@CaP-CTP inhaled by TAC-operated mice presented the highest

signal at every time point (Fig. 3b). Similarly, free Cy5.5 also passed through the pulmonary barrier, resulting in a decaying signal in the lungs, but was unable to be retained in the myocardium due to rapid diffusion and/or metabolism (Supplementary Fig. 8). Furthermore, inhalation delivery also resulted in faster and greater accumulation of the nanoparticles in the myocardium than I.V. injection (Fig. 3a, b). In addition, no fluorescence signal was detected 36 h after inhalation in

**Fig. 3 | Heart targeting capacity of TP-10@CaP-CTP at the physiological, histological and cellular levels. a** Cy5.5-labelled TP-10@CaP or TP-10@CaP-CTP (50 μL) was administered via inhalation or I.V. injection for ex vivo imaging to evaluate the distribution of different nanoparticles in TAC and Sham mice at different time points (0.5, 1, 3, 6, and 24 h). **b** Time-course quantification of fluorescence signals from the heart and lung tissue. $n = 3$ hearts or lungs in each group. **c** Schematic of the heart-targeting capacity of TP-10@CaP and TP-10@CaP-CTP, which were administered via inhalation or I.V. injection. **d** IVIS images of the hearts of TAC and Sham mice that received Cy5.5-labelled TP-10@CaP or TP-10@CaP-CTP via inhalation or I.V. injection at the peak time points (the peak for the inhalation approach was 1 h, while that for I.V. injection was 3 h). **e** Quantitative assay. $n = 5, 4,$

4, 4, 5 and 4 hearts respectively. **f** Representative images showing the distribution of Cy5.5-labelled TP-10@CaP and TP-10@CaP-CTP in heart tissue from mice after inhalation treatment (1 h) or I.V. injection (3 h). $n = 4$ hearts in each group. Scale bar, 50 μm. **g** Representative images showing the distribution of Cy5.5-labelled TP-10@CaP and TP-10@CaP-CTP in primary CMs isolated from mice after inhalation treatment (1 h) or I.V. injection (3 h). $n = 4$ hearts in each group. Scale bar, 50 μm. Quantitative assessment of fluorescence signals from **h**, heart tissue and **i**, primary CMs. The results are presented as the mean ± SD. For **e**, **h**, and **i**, statistical analysis was performed using one-way ANOVA with the Bonferroni multiple comparison correction. Source data are provided as a Source Data file.

the lungs of the sham or TAC-operated mice, which suggested that there was no pulmonary retention of the nanoparticles (Supplementary Fig. 9).

The maximum signals were observed at 1 h and 3 h in the inhalation group and the I.V. injection group, respectively (Fig. 3b). Therefore, the tissues were harvested at these times for subsequent comparative analysis (Fig. 3c). The administration of TP-10@CaP-CTP via inhalation resulted in significantly greater myocardial accumulation than did the administration of TP-10@CaP-CTP via I.V. injection in TAC model mice at the peak time point (Fig. 3d, e). Moreover, the accumulation of TP-10@CaP-CTP in the hearts of TAC mice was significantly greater than that in the hearts of sham mice and TP-10@CaP-treated TAC mice. Furthermore, there were no differences in myocardial accumulation between TP-10@CaP with or without CTP modification-treated sham mice and TP-10@CaP-treated TAC mice. To verify the accumulation of nanoparticles in the myocardium, hearts were harvested from mice in different groups, and frozen sections were generated. Confocal laser scanning microscopy (CLSM) was performed, and the results demonstrated that the delivery of TP-10@CaP-CTP to TAC model mice by inhalation resulted in the highest fluorescence intensity (Fig. 3f, h). Similar results were obtained after evaluating primary mouse CMs isolated from different groups (Fig. 3g, i). These results indicated that CTP only targeted hearts in a pathophysiological state. This selectivity of CTP might facilitate accurate drug delivery to the failing heart.

In addition, we observed the distribution of TP-10@CaP-CTP in other major organs (including the liver, brain, spleen, and kidney). Compared with I.V. injection, inhalation resulted in less nanoparticle accumulation in other organs but greater accumulation in the heart (Supplementary Fig. 10). Moreover, mice that inhaled Cy5.5-labelled TP-10@CaP-CTP showed greater fluorescence in heart tissue and less fluorescence in the other major extrapulmonary organs than did mice that received Cy5.5-labelled TP-10@CaP or free Cy5.5 (Supplementary Fig. 11). The TP-10 levels in the myocardium and other major organs were subsequently quantified via HPLC-ESI-MS/MS. Consistent with the above results, the amount of TP-10, expressed as the injected dose per gram of tissue (%ID/g), in the heart was calculated to be 5.74 ± 1.17% in the TP-10@CaP-CTP group, which is notably greater than that in the TP-10@CaP and free TP-10 groups (Supplementary Table 1). Moreover, some TP-10 was detected in the liver and kidneys, as CaP nanoparticles can accumulate in the liver due to absorption by the mononuclear phagocyte system; those in the kidney likely accumulate due to renal excretion[27]. Together, these data indicated that more TP-10@CaP-CTP accumulated in the myocardium after delivery by inhalation than after I.V. injection and CTP modification facilitated the specific targeting of the CaP nanoparticles to the failing heart, which may reduce nanoparticle accumulation in other major organs.

**In vitro efficacy of TP-10@CaP-CTP in attenuating adult mouse CM (AMCM) pathological hypertrophy**
Pathological cardiac hypertrophy is a critical pathological condition that precedes heart failure[28,29]. We explored whether CTP can target CMs in vitro. Consistent with the data from the in vivo study, CTP

modification significantly promoted the ability of TP-10@CaP-CTP to target hypertrophic neonatal rat ventricular myocytes (NRVMs) stimulated with phenylephrine (PE; 100 μM) in a CTP concentration-dependent manner (Supplementary Fig. 12). Compared to the substantial cellular uptake of TP-10@CaP-CTP by PE-induced NRVMs, no significant difference in nanoparticle accumulation in TP-10@CaP-treated PBS- or PE-induced NRVMs or TP-10@CaP-CTP-treated normal NRVMs was observed (Fig. 4a, b). These results confirmed the ability of CTP to specifically target pathological CMs.

We then verified the therapeutic effect of TP-10@CaP-CTP nanoparticles on PE-induced AMCMs. As a selective inhibitor of PDE10A, TP-10 was reported to regulate intracellular cAMP and cGMP levels[23,30]. Thus, we first measured the intracellular levels of cAMP and cGMP in AMCMs by enzyme-linked immunosorbent assays (ELISAs). After stimulation with PE for 24 h to induce pathological hypertrophy, the intracellular cAMP and cGMP levels were strikingly lower (Fig. 4c). However, treatment with TP-10@CaP-CTP abolished the ability of PE to reduce the intracellular levels of cAMP and cGMP, while the effects were weak in the TP-10@CaP groups (Fig. 4c). We further determined the AMCM phenotype distribution in each group. We found that the CM cell surface area was significantly increased after PE stimulation (Fig. 4d, e). Moreover, the expression of hypertrophic marker genes, including *Nppa*, *Nppb* and *β-MHC*, increased (Fig. 4f–h). These changes in phenotype could be attenuated by treatment with TP-10@CaP-CTP nanoparticles but not CaP-CTP or TP-10@CaP nanoparticles.

Numerous previous studies have demonstrated the effect of the cAMP/AMPK axis on various cell types after AMPK activation[31]. AMPK is a natural energy sensor in mammalian cells that plays an essential role in energy homeostasis, can protect against cardiac hypertrophy[32–34] and has numerous beneficial effects on heart failure[31,32,35,36]. Moreover, cAMP can lead to AMPK activation in a protein kinase A (PKA)- or exchange protein activated by cAMP (EPAC)-dependent manner[31]. This evidence prompted us to explore whether the therapeutic effect of TP-10@CaP-CTP was dependent on cAMP-induced AMPK activation. First, we determined the phosphorylation level of AMPK in AMCMs by WB. AMPK phosphorylation was significantly reduced in PE-stimulated AMCMs, but this effect was reversed by treatment with TP-10@CaP-CTP (Fig. 4i, j). Then, we used the AMPK inhibitor compound C (CC) and found that the therapeutic effect of TP-10@CaP-CTP on cardiac hypertrophy was blocked by CC, as indicated by the CM cell surface area (Fig. 4k, l). Moreover, CC abolished the effects of TP-10@CaP-CTP on the downregulation of hypertrophic marker genes (Fig. 4m–o). These results indicated that TP-10@CaP-CTP nanoparticles can effectively attenuate pathological hypertrophy of CMs via the cAMP/AMPK signalling pathway.

**In vitro efficacy of TP-10@CaP-CTP in CFs**
Given that TP-10@CaP-CTP nanoparticles significantly accumulated in the fibrotic regions of failing hearts, while TP-10@CaP not (Supplementary Fig. 13). We explored the effects of TP-10@CaP-CTP nanoparticles on adult mouse CFs. In vitro, we found that TP-10@CaP-CTP also possessed the capacity to target CFs stimulated by TGF-β, a fibrosis stimulant. The TP-10@CaP-CTP-treated TGF-β-induced CFs

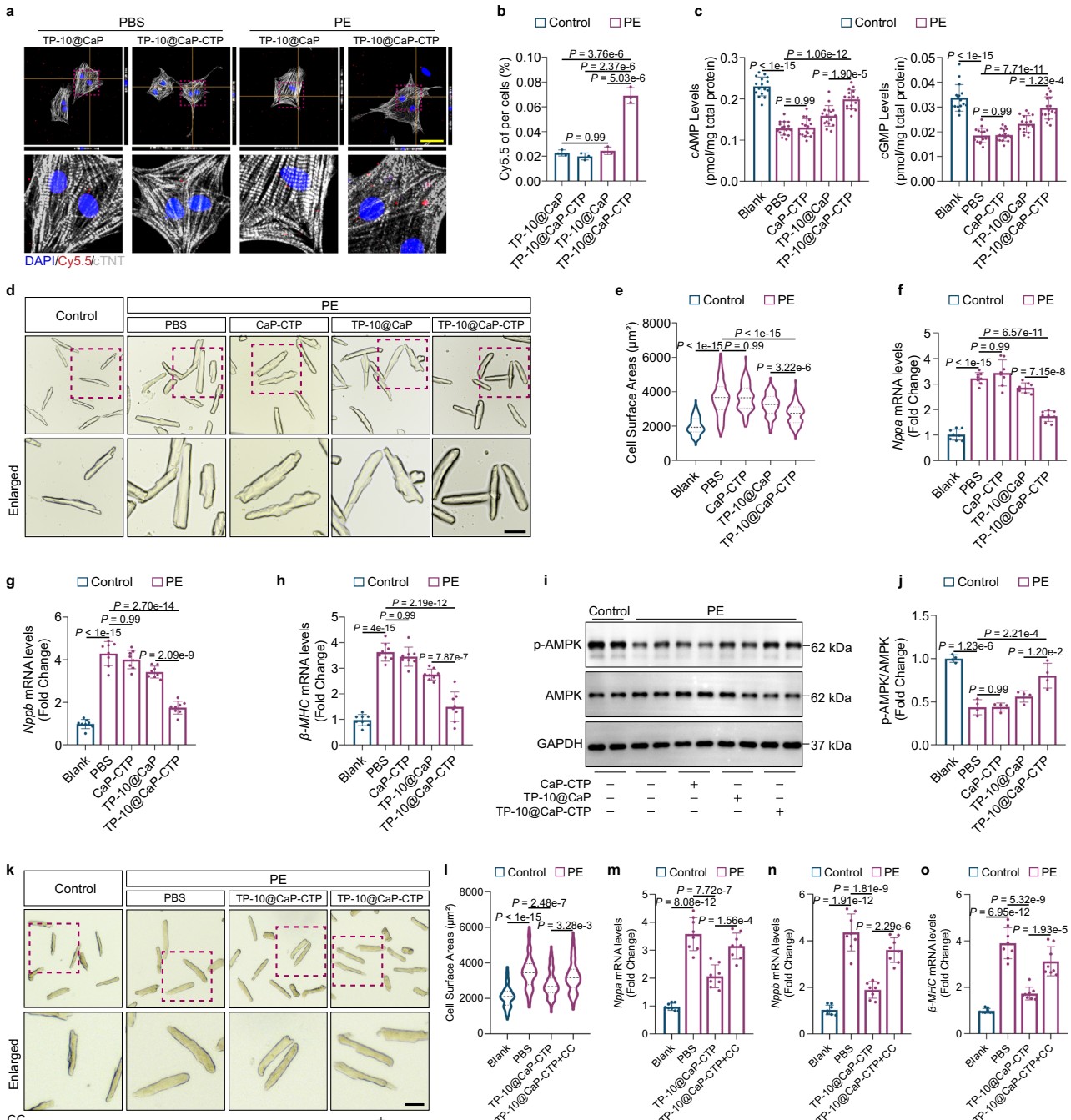

**Fig. 4 | TP-10@CaP-CTP attenuated cardiomyocyte pathological hypertrophy in vitro. a** Representative images of the intracellular uptake of Cy5.5-labelled TP-10@CaP and TP-10@CaP-CTP in PBS- or PE-induced (pretreated with 100 μM PE for 24 h) neonatal rat ventricular myocytes (NRVMs). Scale bar, 50 μm. **b** Quantitative analysis of fluorescence signals from NRVMs. *n* = 3 biologically independent samples in each group. **c** Intracellular cAMP and cGMP levels in CMs determined by ELISA. *n* = 16 biologically independent samples in each group. **d** Representative images of CMs after treatment. CMs were isolated from the left ventricle of adult mice and pretreated with 25 μg/mL CaP-CTP, TP-10@CaP, TP-10@CaP-CTP, or vehicle for 2 h, followed by treatment with 100 μM PE or vehicle for 24 h. Then, the CMs were fixed and photographed under a microscope. Scale bar, 50 μm. **e** CM cell surface area. *n* = 91–102. **f**–**h** mRNA levels of hypertrophic marker genes in CMs after the indicated treatments. *n* = 8 biologically independent samples in each

group. The results are presented as the mean ± SD. **i**, **j** Western blot analysis and quantification of AMPK expression and AMPK phosphorylation. GAPDH was used as a loading control. *n* = 4 biologically independent samples in each group.
**k** Representative images of CMs after different treatments. CMs were pretreated with 25 μg/mL CaP-CTP, TP-10@CaP, TP-10@CaP-CTP, TP-10@CaP-CTP with the AMPK inhibitor compound C (CC, 5 μM) or vehicle for 2 h, followed by treatment with 100 μM PE or vehicle for 24 h. Scale bar, 50 μm. **l** CM cell surface area after different treatments. *n* = 92–94. **m**–**o** mRNA levels of hypertrophic marker genes in CMs after different treatments. *n* = 8 biologically independent samples in each group. The results are presented as the mean ± SD. For **b**, **c**, **e**, **f**, **g**, **h**, **j**, **l**, **m**, **n** and **o**, statistical analysis was performed using one-way ANOVA with the Bonferroni multiple comparison correction. Source data are provided as a Source Data file.

presented significantly greater nanoparticle accumulation. However, upon comparing TP-10@CaP-treated TGF-β-induced CFs with TP-10@CaP-CTP- or TP-10@CaP-treated normal CFs, no overt difference was observed in nanoparticle accumulation (Supplementary Fig. 14). This indicated that CTP might also target CFs in a pathological state.

To explore the therapeutic effect of TP-10@CaP-CTP on CFs, we firstly isolated CFs from adult mouse hearts and then treated them with TGF-β to promote fibrosis. We found that the intracellular cAMP and cGMP levels both decreased after TGF-β stimulation, but this effect was effectively reversed by TP-10@CaP-CTP (Supplementary Fig. 15). We then determined the effects of TP-10@CaP-CTP on CF activation and proliferation. A scratch wound healing assay and a Boyden chamber migration assay showed that the TP-10@CaP-CTP nanoparticles significantly inhibited TGF-β-induced CF migration (Fig. 5a–d). Moreover, the CCK8 proliferation assay confirmed that CF proliferation was significantly reduced after treatment with TP-10@CaP-CTP nanoparticles (Fig. 5e). α-Smooth muscle actin (α-SMA) is a biomarker of CF activation. Immunostaining (Fig. 5f, g) and WB analysis (Fig. 5h, i) confirmed the inhibitory effects of TP-10@CaP-CTP nanoparticles on TGF-β-induced CF activation. The expression level of the ECM protein α-1 type 1 collagen (Col1α1) was consistently decreased after treatment with the TP-10@CaP-CTP nanoparticles (Fig. 5h, j). Real-time quantitative PCR confirmed the inhibitory effects of TP-10@CaP-CTP on the expression of *Acta1* and the ECM-related genes *Col1α1, Col3α1* and *Fn1* (Fig. 5k).

According to the above results, TP-10@CaP-CTP reversed the reduction in cAMP and cGMP levels in CFs stimulated by TGF-β (Supplementary Fig. 15). As an effector of cGMP, activated PKG exerts antifibrotic effects on heart failure, thus inhibiting cardiac fibrosis and improving cardiac function[37,38]. This finding prompted us to explore whether the inhibitory effect of TP-10@CaP-CTP nanoparticles on TGF-β-stimulated CFs was cGMP/PKG dependent. We used a selective PKG inhibitor, DT-2, and found that all of the effects of the TP-10@CaP-CTP nanoparticles on CFs were blocked by treatment with DT-2 (Fig. 5). These results demonstrated that TP-10@CaP-CTP nanoparticles could inhibit CF activation, migration, and proliferation via a cGMP/PKG-dependent mechanism.

### Efficacy of inhaled TP-10@CaP-CTP in a pressure overload-induced mouse model of heart failure

To evaluate the therapeutic effects of TP-10@CaP-CTP nanoparticles on chronic heart failure, eight-week-old C57 male mice were subjected to TAC surgery to establish a chronic heart failure mouse model, and then the mice were treated with inhalable TP-10@CaP-CTP nanoparticles intratracheally (Fig. 6a). After the mice inhaled different treatments (PBS, CaP-CTP, TP-10, TP-10@CaP and TP-10@CaP-CTP), it was determined that TP-10@CaP-CTP treatment via inhalation significantly improved the long-term survival rate of the heart failure model mice (Fig. 6b). Moreover, six weeks after treatment, the PBS and CaP-CTP treatment groups exhibited strikingly larger hearts (Fig. 6c, d) and greater heart weight/body weight ratios (Fig. 6h) than did the sham group. In contrast, these effects were partially attenuated by inhalation of TP-10 or TP-10@CaP but significantly improved by inhalation of TP-10@CaP-CTP. Moreover, Masson staining and Sirius staining revealed widespread cardiac fibrosis in PBS- and CaP-CTP-treated mice (Fig. 6e, f, i). However, there was less cardiac fibrosis in mice that received TP-10@CaP-CTP via inhalation. Additionally, wheat germ agglutinin (WGA) staining showed that the cross-sectional areas of CMs from PBS- and CaP-CTP-treated TAC model mice were obviously larger than those of CMs from sham mice. However, TP-10@CaP-CTP inhalation treatment significantly improved CM hypertrophy and exhibited better therapeutic efficacy than the TP-10 and TP-10@CaP treatments (Fig. 6g, j). The plasma level of ANP, a biomarker for heart failure, was also significantly reduced in the mice that received TP-10@CaP-CTP via inhalation (Supplementary Fig. 16f).

Consistently, real-time quantitative PCR demonstrated that TAC-induced hypertrophic (Fig. 6k) and fibrotic (Fig. 6l) marker expression was strikingly downregulated in the myocardia of mice that inhaled TP-10@CaP-CTP. In addition, only when the dosage increased threefold (e.g., inhalation of 7.5 mg/kg/2 days) did the TP-10@CaP nanoparticles show therapeutic efficiency similar to that of the TP-10@CaP-CTP nanoparticles at the current dosage (2.5 mg/kg/2 days) (Supplementary Fig. 17). Furthermore, the effects of different doses of TP-10 on heart failure were evaluated (Supplementary Fig. 18). Inhalation of a high dosage of free TP-10 (3 mg/kg/2 days), which was similar to the dosage used in previous studies[23], partially ameliorated the pathological changes associated with heart failure. However, this effect was still inferior to that of TP-10@CaP-CTP inhalation at the dosage of 2.5 mg/kg/2 days (the dose of TP-10 was approximately 0.75 mg). Finally, we explored the therapeutic efficacy of TP-10@CaP-CTP nanoparticles administered via I.V. injection. The effect on heart failure after contrasting TAC was less pronounced after I.V. injection of TP-10@CaP-CTP than after inhalation of TP-10@CaP-CTP (Supplementary Fig. 19). These data suggested that TP-10@CaP-CTP inhalation treatment effectively attenuated the pathological process of chronic heart failure at a relatively low dosage.

### Effects of inhaled TP-10@CaP-CTP on cardiac function

We then evaluated the effects of the inhaled nanoparticles on cardiac function. Mice that received different nanodrug therapies via inhalation for six consecutive weeks after TAC surgery were monitored by mouse ultrasonic cardiography, and then CMs were isolated and subjected to a shortening/relengthening assay (Fig. 7a). Echocardiography suggested that the time-dependent worsening of cardiac systolic function parameters, including ejection fraction (EF%) and fraction shortening (FS%), was significantly improved by TP-10@CaP-CTP treatment (Fig. 7b–d and Supplementary Fig. 16b, c). Moreover, the left ventricular end-systolic diameter (LVESD) and left ventricular end-diastolic diameter (LVEDD) were clearly ameliorated in mice that received TP-10@CaP-CTP (Fig. 7e, f and Supplementary Fig. 16d, e). These indices represent the size and blood pumping functionality of the heart, and the results were consistent with the pathological changes described above (Fig. 6c). In addition, the cardiac diastolic function parameter E/A ratio was also significantly improved by inhalation of TP-10@CaP-CTP (Fig. 7g). Furthermore, AMCMs were isolated from all the groups to evaluate their shortening/relengthening capacity. Consistent with the echocardiography findings, AMCMs from TP-10@CaP-CTP-treated mice presented notably improved contractile function, as indicated by increased peak shortening (PS) and maximal velocity of shortening/relengthening (±dL/dt) and shortened time to peak shortening (TPS) and time to 90% relengthening (TR$_{90}$) (Fig. 7h–m). These results demonstrated that TP-10@CaP-CTP inhalation treatment effectively improved cardiac systolic and diastolic function in mice with TAC-induced heart failure and suggested that TP-10@CaP-CTP inhalation therapy exerted significant cardio-protection effect.

### Biosafety evaluation of inhaled TP-10@CaP-CTP nanoparticles

Drug-induced side effects in the lungs are a critical factor to consider for widely used inhalation therapy. In addition, chronic heart failure patients usually require medication to be administered over the long term. Thus, the long-term safety of inhaled TP-10@CaP-CTP nanoparticles in the lungs was carefully considered. Healthy male C57 mice were administered TP-10@CaP-CTP nanoparticles (2.5 mg/kg/2 days) via inhalation for 18 consecutive weeks to verify lung safety after long-term inhalation treatment. We harvested lung tissues from mice that had received treatment for 0, 3, 6, 12, and 18 weeks and subjected them to H&E and Masson staining. Compared with the control group, no obvious inflammatory infiltration or fibrosis was observed in the TP-10@CaP-CTP inhalation treatment group (Fig. 8a). Moreover, the lung

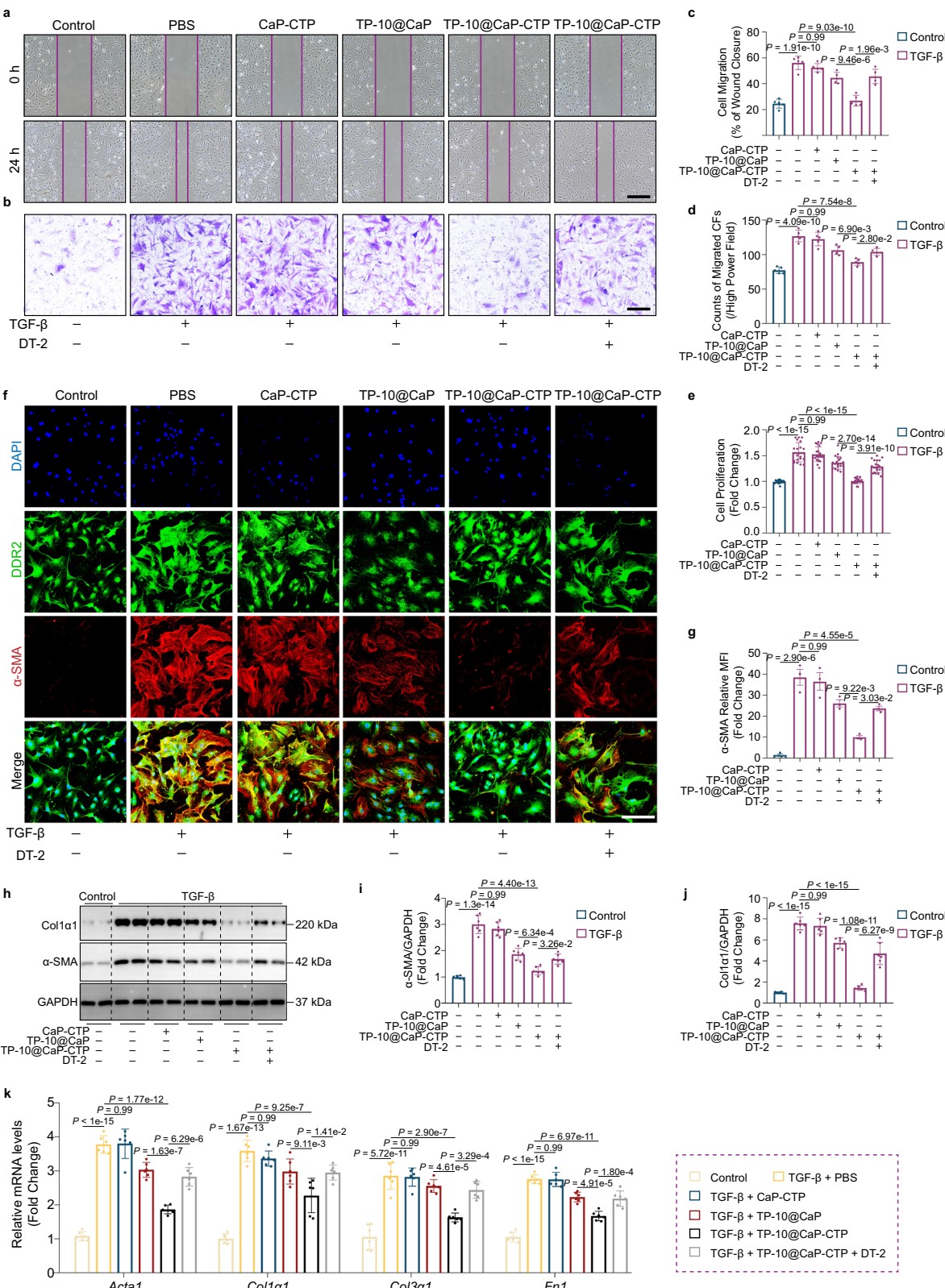

W/D ratio, which indicates the degree of lung oedema, did not significantly increase in the inhalation treatment group at the different time points (Fig. 8b). In addition, we collected bronchoalveolar lavage fluid (BALF) from the mice that received inhalation treatment, and BCA assays showed that protein levels in the BALF were not significantly elevated (Fig. 8c). Moreover, ELISAs demonstrated that there were no obvious differences in the IL-1β, IL-6, or TNF-α levels between the inhalation treatment group and the control group (Fig. 8d–f). Furthermore, we harvested lung tissue after six weeks of treatment. As shown in Supplementary Fig. 20, TAC operation-induced chronic heart failure would lead to pulmonary congestion and inflammatory infiltration. However, inhalation treatment with a low dosage of TP-10@CaP-CTP (2.5 mg/kg/2 days) or a high dosage of TP-10@CaP (7.5 mg/kg/2 days) ameliorated lung injury in the animals with chronic

**Fig. 5 | The effect of TP-10@CaP-CTP on cardiac fibroblasts.** CFs were isolated from adult mice, serum starved for 24 h, and pretreated with 25 μg/mL CaP-CTP, TP-10@CaP, TP-10@CaP-CTP, TP-10@CaP-CTP with 1 μM DT-2 or vehicle for 2 h prior to treatment with TGF-β (10 ng/mL) or vehicle for 24 h. **a** Representative images of CF wound closure after the indicated treatments for 24 h. Scale bar, 500 μm. **b** Representative images of migrated CFs after the indicated treatments for 24 h. Scale bar, 200 μm. **c** Statistical analysis of CF wound closure. n = 5. **d** Statistical analysis of migrated CFs. n = 5 biologically independent samples in each group. **e** CCK8 assay to evaluate the cell proliferation capacity of adult mouse CFs. n = 20 biologically independent samples in each group. **f** Representative images of α-SMA expression in adult mouse CFs after the indicated treatments.

Scale bar, 200 μm. **g** Quantitative assessment of fluorescence intensity. n = 3 biologically independent samples in each group. **h** Western blotting analysis of α-SMA and Col1a1 in adult mouse CFs after the indicated treatments. GAPDH was used as a loading control. **i** Quantitative assessment of α-SMA expression. n = 6 biologically independent samples in each group. **j** Quantitative assessment of Col1α1 expression. n = 6 biologically independent samples in each group. k, mRNA levels of the indicated genes in adult mouse CFs after the indicated treatments. n = 6 biologically independent samples in each group. The results are presented as the mean ± SD. For **c, d, e, g, i, j,** and **k,** statistical analysis was performed using one-way ANOVA with the Bonferroni multiple comparison correction. Source data are provided as a Source Data file.

heart failure. These data indicated that long-term inhalation treatment at a low dose (2.5 mg/kg/2 days for mice) or short- to medium-term inhalation treatment at a high dose (7.5 mg/kg/2 days for mice) might not lead to lung injury.

Finally, we verified the biocompatibility of the TP-10@CaP-CTP nanoparticles. First, the toxicity of the TP-10@CaP-CTP nanoparticles was evaluated by performing a CCK8 assay on AC16 cells, which are a proliferating human CMs. No obvious cytotoxicity was observed when the AC16 cells were treated with a high concentration of TP-10@CaP-CTP nanoparticles (200 μg/mL), as shown in Supplementary Fig. 21a. Then, we collected serum and tissue samples from healthy mice 4 h after inhalation treatment and control mice to detect the inflammatory response. The serum IL-1β, IL-6, and TNF-α levels were not significantly elevated in the inhalation treatment group (Supplementary Fig. 21b). H&E staining revealed no obvious histopathological changes in the major organs (lung, liver, kidney, spleen, and brain) in the inhalation treatment group (Supplementary Fig. 21c). In addition, we explored whether TP-10@CaP-CTP can lead to complement activation. Plasma samples were collected from the mice and coincubated with TP-10@CaP-CTP. Then, the plasma levels of C3 and its activated fragments (C3b and C3c) were detected by ELISA. There was no significant increase in plasma C3, C3b or C3c levels (Supplementary Fig. 22a–c). Moreover, plasma samples from mice that received TP-10@CaP-CTP inhalation treatment were also subjected to ELISA. As expected, there was no increase in plasma C3, C3b or C3c levels (Supplementary Fig. 22d–f). These results demonstrated the good biocompatibility and biosafety of the TP-10@CaP-CTP nanoparticles.

## Discussion

Patients with heart failure, a chronic cardiovascular disease, require lifelong treatment. Most commonly, drugs to treat heart failure are administered orally. However, poor transport efficacy and first-pass metabolism render oral delivery unreliable[8]. Moreover, drugs administered via I.V. injection might lead to poor patient adherence in the long-term management of heart failure. Delivering drugs via inhalation can solve these problems. However, two major challenges must be overcome to allow the use of inhalation delivery in the long-term management of chronic heart failure. First, both biodegradability and biocompatibility are essential for the prolonged use of drug delivery systems in the body. Second, avoiding lung complications is highly important for efficient inhaled drug delivery. To address these issues, in this study, we designed a biodegradable CaP nanoparticle-based delivery system that supports the targeted delivery of drugs to the heart via the cardiopulmonary circulation. Inhalation of the nanoparticles led to greater and faster cardiac accumulation than I.V. administration (Fig. 3a), which is consistent with previous studies[9,13,14]. Moreover, modification with CTP further enhanced nanoparticle accumulation in failing hearts. Furthermore, we demonstrated the therapeutic effects of our nanoparticles in vitro and in vivo. At the same dosage, inhaled TP-10@CaP-CTP exerted better therapeutic effects than inhaled TP-10 or TP-10@CaP. In addition, the dosage of TP-10@CaP-CTP nanoparticles inhaled in this study (2.5 mg/kg/2 days,

with a content of TP-10 that is equivalent to approximately 0.75 mg/kg/ 2 days) was much lower than that used in a previous study (3.2 mg/kg/ day TP-10 for the prevention of heart failure by I.V. injection)[23]. These results may be attributed to the fact that the effective myocardial accumulation of the TP-10@CaP-CTP nanoparticles can facilitate TP-10 delivery. Another explanation is that CaP nanoparticles can prolong drug retention in the heart. In a previous study, the clearance of CaP nanoparticles loaded with a mimetic peptide (MP) was delayed in the heart[9]. The authors of this study found that inhaled CaP loaded with MP had a much greater effect than that of free MP in the treatment of diabetic cardiomyopathy. However, in our study, there were no significant differences in the improvement of the pathological process of heart failure or cardiac function between the free TP-10 inhalation group and the TP-10@CaP group (Figs. 6 and 7). This discrepancy may be due to the high protein binding of TP-10[39,40]. Thus, the levels of TP-10 in the myocardium after administration of TP-10@CaP or free TP-10 were far from the minimal effective concentrations due to the high protein binding nature of TP-10 in vivo. In contrast, TP-10@CaP-CTP had good effects on preventing heart failure. Consequently, this inhalable TP-10@CaP-CTP nanoparticle-mediated therapeutic strategy is expected to be used to improve the efficacy of other small molecule drugs, which has certain clinical relevance for reducing the drug dosage used and minimizing side effects.

The main purpose of this study was to propose a heart-targeted peptide (CTP) engineering strategy to promote nanodrug accumulation in the failing myocardium. CTP was discovered via an in vivo phage display technique and was determined to be an ischaemic myocardium-homing peptide[22]. In previous studies, CTP was widely used for targeted drug delivery in ischaemic heart disease[21,41,42]. However, few studies have explored the application of CTP for chronic heart failure. The in vitro and in vivo studies demonstrated that CTP strikingly promoted CaP nanoparticle targeting of the failing myocardium in a pressure overload heart failure mouse model, and such targeting only occurred with hearts in a pathological state. Our data suggested that both CMs and CFs are targeted under pathological stimulation. Interestingly, pulmonary epithelial cells and fibroblasts were not targeted by this CTP-modified nanomedicine (Supplementary Fig. 23 and Fig. 24). We hypothesized that the targeting capacity of CTP might be mediated by a specific combination of proteins or receptors in CMs and CFs under pathological conditions; however, more evidence is needed to confirm this hypothesis.

Clinically, adverse drug-induced respiratory reactions are a critical limitation of inhalable drugs for systemic delivery. In previous studies, inhalable CaP nanoparticles were utilized as carriers of therapeutic peptides or microRNAs for the treatment of diabetic cardiomyopathy[9] and heart failure[12], and the biosafety of only the CaP nanoparticles was demonstrated. More recently, Alogna et al. performed a 1-week daily inhalation toxicity study and demonstrated the safety of inhalable dry powder CaP (dpCaP) nanoparticles. After 2 weeks of inhalation of dpCaP nanoparticles, chronic heart failure-induced lung congestion and damage were significantly improved[43]. In this study, we investigated the adverse effects of TP-10@CaP-CTP on

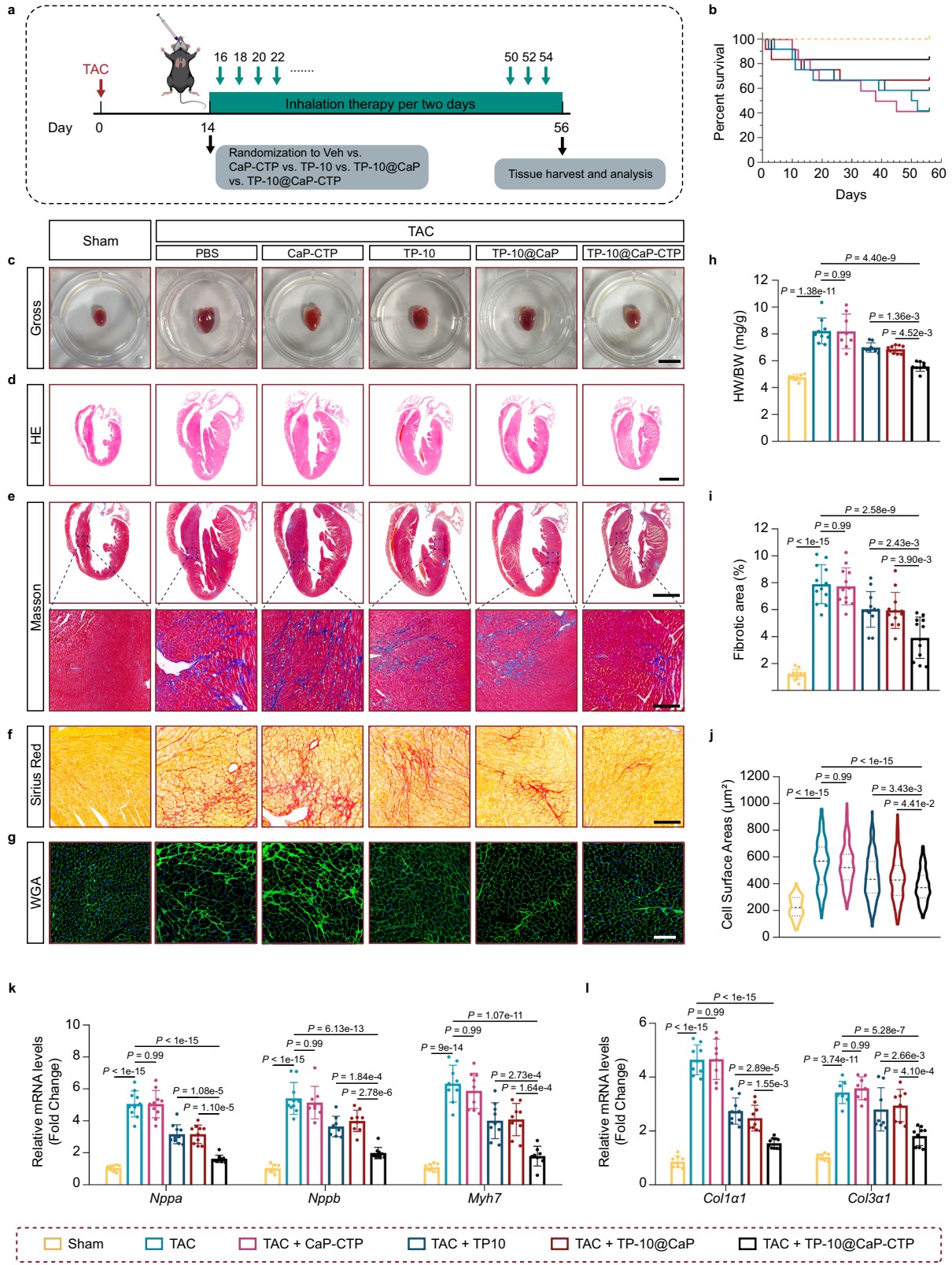

the lungs with 18 weeks of monitoring. After long-term inhalation treatment, no obvious lung injury was observed (Fig. 8). This may be due to the targeting effect of the failing myocardium-targeting peptide CTP in the nanoparticles, as achieving greater myocardial accumulation reduces the therapeutic dosage and attenuates pulmonary burden during nebulization. In addition, consisted with previous study[40], we found that both inhalation treatment of TP-10@CaP in the high dosage

and TP-10@CaP-CTP in the low dosage could ameliorate pulmonary congestion and injury, which induced by chronic heart failure (Supplementary Fig. 20). These results demonstrated that long-term inhalation treatment at a low dose or short- to medium-term inhalation treatment at a high dose were safe for lung. However, more observation may be required to verify the safety of inhalable drugs in the future.

**Fig. 6 | Efficacy of inhaled TP-10@CaP-CTP in a pressure overload-induced heart failure mouse model. a** Schematic overview of inhalation treatment. Two weeks after TAC or sham operation, the mice were treated with PBS, TP-10 (0.75 mg/kg/2 days), CaP-CTP (2.5 mg/kg/2 days), TP-10@CaP (2.5 mg/kg/2 days) or TP-10@CaP-CTP (2.5 mg/kg/2 days) for six weeks via inhalation. **b** Survival curves of the TAC and sham-operated mice after the indicated treatments. $n = 12$. **c** Representative images of the gross appearances of whole hearts. Scale bar, 1 cm. **d** Representative images of heart vertical sections stained with H&E. Scale bar, 2 mm. **e** Representative Masson staining images. Scale bars, 2 mm (top) and 200 μm (bottom). **f** Representative Sirius red staining images. Scale bar, 200 μm.

**g** Representative wheat germ agglutinin (WGA) staining images of mice. Scale bar, 100 μm. **h** Ratios of heart weight to body weight (HW/BW). $n = 7, 9, 8, 9, 9$ and $9$ hearts respectively. **i** Statistical analysis of the fibrotic area of the myocardium. $n = 12$ mice in each group. **j** Statistical analysis of the cell surface areas. $n = 130-136$ CMs from 6–7 mice in each group. **k, l** mRNA levels of the indicated genes in heart tissue after the indicated treatments. $n = 8-10$ biologically independent samples in each group. The results are presented as the mean ± SD. Experiments were performed three times (**d–g**) with similar results. For **h–l**, statistical analysis was performed using one-way ANOVA with the Bonferroni multiple comparison correction. Source data are provided as a Source Data file.

In addition, chronic pressure overload in heart failure patients is usually complicated by an increase in muscularized lung vessels, myofibroblast proliferation, and general lung fibrosis[44,45]. Since CTP possesses the ability to target CFs, it was necessary to explore whether CTP also targeted lung fibroblasts. In vitro, we confirmed that CTP did not target lung fibroblasts with or without TGF-β stimulation (Supplementary Fig. 24) or lung epithelial cells (Supplementary Fig. 23). Moreover, in vivo, we did not observe a significant difference in pulmonary nanoparticle retention between the sham and TAC groups. Furthermore, the nanoparticles were not retained in the lung over time (Supplementary Fig. 9). These findings indicated that CTP does not target lung fibroblasts and that the CaP nanoparticles would not be retained in pulmonary tissue. CTP was first discovered to target the ischaemic myocardium by an in vivo phage display technique[22]. The specific proteins or receptors highly expressed in the failing myocardium (in the CMs and CFs) rather than in the lung might be involved in CTP homing. These results confirmed the promising application of inhalable TP-10@CaP-CTP nanoparticles in chronic heart failure patients who have specific lung conditions.

We also revealed the potential mechanism underlying the therapeutic effects of the TP-10@CaP-CTP nanoparticles, showing that they attenuate cardiac hypertrophy and cardiac fibrosis via the cAMP/AMPK and cGMP/PKG signalling pathways, respectively. TP-10 is a selective inhibitor of PDE10A that can hydrolyse both cAMP and cGMP[23]. A recent study showed that PDE10A is expressed in low levels in the normal heart but highly expressed in failing hearts. In addition, PDE10A inhibition or deficiency inhibited cardiac remodelling and improved cardiac function[23]. However, the potential underlying mechanism is still unclear. In this study, we verified that TP-10@CaP-CTP regulated the intracellular cAMP and cGMP levels in both CMs and CFs, which is consistent with a previous study[23]. In addition, the therapeutic effects were blocked by an AMPK inhibitor in CMs and a PKG inhibitor in CFs. These results indicated that TP-10@CaP-CTP attenuated CM hypertrophic changes in a cAMP/AMPK-dependent manner, while CFs relied on cGMP/PKG signalling. Although these experiments were preliminary, the results partially elucidated the potential mechanism underlying the effects of TP-10@CaP-CTP on CMs and CFs.

Despite the encouraging results of this study, the therapeutic effect of the currently developed strategy has not shown a substantial advancement over current clinical practice; thus, future investigations should focus on the systematic optimization of inhalation-based approaches, including various parameters involving atomization efficiency, drug loading, and accumulation in the heart by targeted design. In addition, some pathological lung conditions (such as chronic obstructive pulmonary disease) might hinder drug absorption via cardiopulmonary circulation. Future studies are needed to investigate and explore these challenges.

In summary, we presented an inhalable heart-targeting nanoparticle, TP-10@CaP-CTP, for the long-term treatment of chronic heart failure. The high delivery efficiency of administration via inhalation and CTP modification allows targeting of the pathological myocardium and improve the local accumulation of the nanomedicine,

thus achieving better therapeutic outcomes with TP-10 and avoiding adverse lung reactions. Upon delivery into the failing myocardium, the CaP shell decomposes and TP-10 is released, which can effectively attenuate cardiac remodelling and improve cardiac function. Moreover, inhaled TP-10@CaP-CTP displayed an improved therapeutic effect in a pressure overload-induced mouse model. This study verified the feasibility of using CTP for the treatment of pressure overload heart failure and confirmed the biosafety of inhalable CaP nanoparticles for long-term treatment. Although this proof-of-concept work is still in an early phase, we envision that this strategy based on cardiopulmonary circulation delivery holds great promise for the long-term management of chronic heart failure.

## Methods
### Materials and reagents
Calcium chloride (CaCl₂), sodium citrate (Na₃Cit), sodium phosphate dibasic (Na₂HPO₄), N-hydroxysuccinimide (NHS) and sodium hydroxide (NaOH) were obtained from the Aladdin Reagent Co. (Shanghai, China). N-(3-dimethylaminopropyl)-N'-ethylcarbodiimide hydrochloride (EDC•HCl), 1,2-distearoyl-sn-glycero-3-phosphoethanolamine conjugated polyethylene glycol acid (DSPE-PEG-COOH) and aminecyanine 5.5 (Cy5.5) were purchased from Sigma-Aldrich Trading Co., Ltd. (Shanghai, China). TP-10 was obtained from MedChemExpress (Shanghai, China). CTP (CSTSMLKAC) was supplied by QYAOBIO (Hubei, China). Deionized water (18.2 MΩ cm) was used throughout all the experiments. All reagents and chemicals were of analytical grade and used as received without further purification.

### Preparation and characterization of TP-10@CaP-CTP nanoparticles
The CaP nanoparticles were synthesized according to previous work[9,24]. Briefly, 5 mL of CaCl₂ (100 mM) and 5 mL of Na₃Cit (500 mM) were mixed, followed by the addition of 1 mL of TP-10 in DMSO (0.5 mg/mL). Then, 5 mL of Na₂HPO₄ (150 mM) was added prior to shaking for 30 min. Subsequently, the pH of the solution was adjusted to approximately 12 by adding NaOH (1.0 M), and the mixture was stirred for 1.5 h. The product (TP-10@CaP) was dialyzed against 500 mL of deionized water (with a MW cut-off of 3500 Da) for 3 days (replacing the water with fresh water every 24 h) and collected for further use. Moreover, the CaP nanoparticles were prepared by the same methods except that TP-10 was not added. The loading capacity and EE of TP-10 were determined by HPLC–MS/MS, and the detection conditions were the same as those used for the in vitro drug release experiment, as presented in the Supplementary Information.

For CTP (CSTSMLKAC, Supplementary Fig. 1) modification, TP-10@CaP was initially modified with DSPE-PEG-COOH. Five milligrams of DSPE-PEG-COOH was predissolved in DMSO and then added to the TP-10@CaP dispersion; the resulting mixture was stirred for 24 h. The TP-10@CaP-DSPE-PEG nanoparticles were collected by centrifugation and redispersed in 5 mL of deionized water. Then, EDC (35 mg) was added to the TP-10@CaP-DSPE-PEG dispersion, followed by the addition of NHS (53 mg) with ultrasonication. The mixture was stirred for 24 h to activate the carboxylic groups. Afterwards, 4 mL of CTP (2 mg/

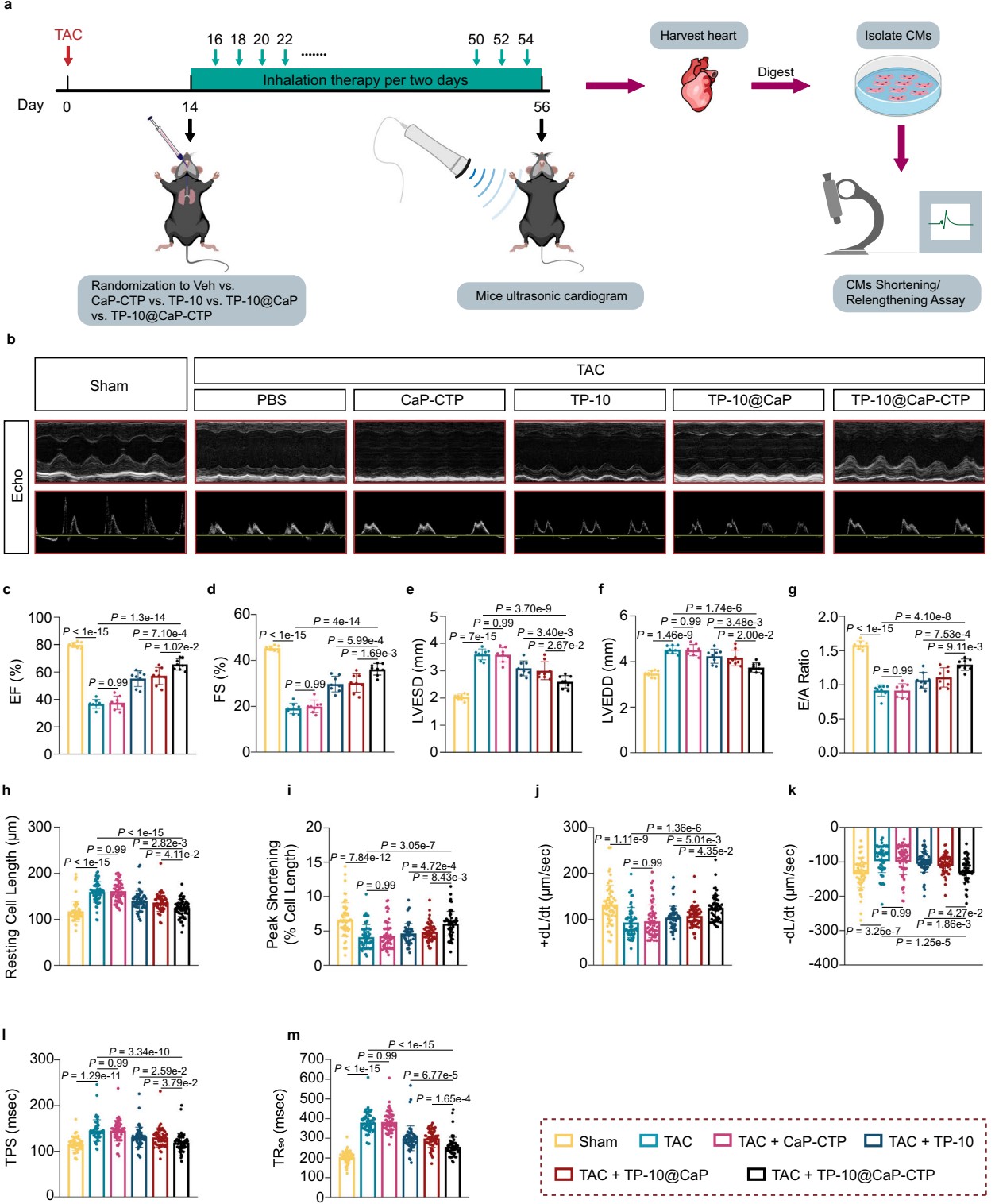

**Fig. 7 | The effect of inhaled TP-10@CaP-CTP on cardiac function. a** Schematic of the mouse cardiac function evaluation and CM shortening/relengthening assays. After six weeks of inhalation treatment with PBS, TP-10 (0.75 mg/kg/2 days), CaP-CTP (2.5 mg/kg/2 days), TP-10@CaP (2.5 mg/kg/2 days) or TP-10@CaP-CTP (2.5 mg/kg/2 days), the mice were subjected to ultrasound cardiography, and CMs were isolated from the left ventricle for shortening/relengthening assays. **b** Representative echocardiography images from each study group after six weeks of treatment. **c**–**g** Percent ejection fraction (EF%), percent fraction shortening (FS%), left ventricular end-systolic diameter (LVESD), left ventricular end-diastolic

diameter (LVEDD), and the E/A ratio. $n = 8$ mice in each group. **h**–**m** Primary CMs were isolated from each study group at the 8-week time point prior to mechanical assessment. Resting cell length, peak shortening (PS), maximal velocity of shortening (+dL/dt), maximal velocity of relengthening (−dL/dt), time to peak shortening (TPS), and time to 90% relengthening (TR$_{90}$) were measured. $n = 58$ CMs from 3 mice in each group. The results are presented as the mean ± SD. For **c**–**m**, statistical analysis was performed using one-way ANOVA with the Bonferroni multiple comparison correction. Source data are provided as a Source Data file.

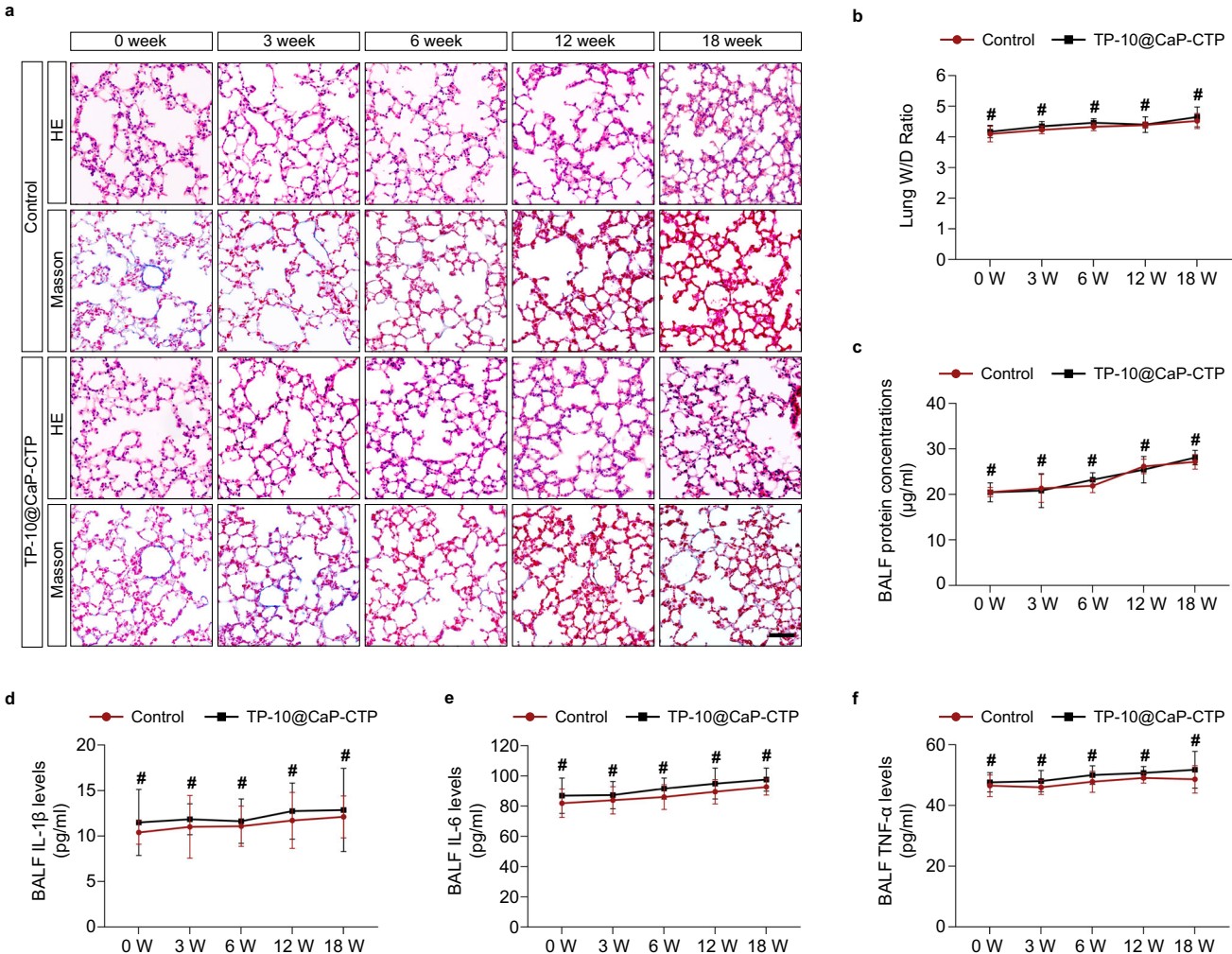

**Fig. 8 | Verification of the biosafety of inhaled TP-10@CaP-CTP nanoparticles in the lung.** Eight-week-old healthy male C57 mice were subjected to 18 consecutive weeks of inhalation treatment with TP-10@CaP-CTP (2.5 mg/kg/2 days) to verify lung safety after long-term inhalation treatment. **a** Representative H&E and Masson staining images of lung sections from the inhalation treatment and control groups at different time points (0, 3, 6, 12, and 18 weeks). Scale bar, 50 μm. **b** Lung W/D ratio. **c** Quantitative analysis of BALF protein concentrations and **d**–**f**, BALF levels of IL−1β, IL-6, and TNF-α at different time points. $n = 3$ biologically independent samples in each group. The results are presented as the mean ± SD. $\#P > 0.05$. For **b**–**f**, statistical analysis between two groups was determined using unpaired two-tailed $t$ tests. Source data are provided as a Source Data file.

mL) was added. After gentle stirring overnight, the product TP-10@CaP-CTP was collected by centrifugation and washed with deionized water and ethanol several times. The amount of CTP bound and EE were evaluated by a similar procedure and finally determined by a BCA assay (Beyotime, #P0012) with a low limit of detection of 0.23 mg/mL. Moreover, the CaP-CTP nanoparticles were prepared by the same process but TP-10@CaP was replaced with CaP nanoparticles. In addition, Cy5.5-labelled TP-10@CaP-CTP and TP-10@CaP nanoparticles were prepared through electrostatic adsorption. Briefly, 10 mL of a TP-10@CaP-CTP or TP-10@CaP dispersion (1 mg/mL) was mixed with 1 mL of an amine-Cy5.5 DMSO solution and stirred at room temperature overnight in the dark. The obtained Cy5.5-labelled TP-10@CaP-CTP and TP-10@CaP nanoparticles were collected by centrifugation, washed with deionized water and ethanol, and dispersed in PBS for further in vivo imaging.

### Cell culture
Primary CMs and CFs were isolated from male adult mice. Primary NRVMs were isolated from male neonatal rats. M199 medium supplemented with 2% BSA, 1% ITS, 1% BDM, 1% CD lipids and 1% penicillin/streptomycin was utilized to culture the CMs. Dulbecco's modified Eagle's medium (DMEM) supplemented with 10% foetal bovine serum and 1% penicillin/streptomycin was utilized to culture the CFs. AC16 cells were purchased from the Chinese Academy of Science Cell Bank (Shanghai, China). MLE-12 and MLFs were purchased from Shanghai Zhong Qiao Xin Zhou Biotechnology Co., Ltd. (Shanghai, China). AC16 and MLE-12 cells were cultured in DMEM supplemented with 10% foetal bovine serum and 1% penicillin/streptomycin. MLFs were cultured in mouse lung fibroblast complete culture medium (#PCM-M-06, Zhong Qiao Xin Zhou, Shanghai, China) supplemented with 5% foetal bovine serum, 5% IST and 1% penicillin/streptomycin. All cells were incubated in a humidified 37 °C, 5% $CO_2$ culture incubator (Thermo Fisher Scientific, Waltham, MA, USA) for subsequent experiments.

### Experimental animals
Eight-week-old male C57BL/6 J mice were purchased from Vital River Laboratory Animal Technology Co., Ltd. (Beijing, China). All animal experiments were performed according to the Guidelines for the Care and Use of Laboratory Animals published by the National Research Council (U.S.) Institute for Laboratory Animal Research and were approved by the Ethics Committee of Zhongshan Hospital, Fudan University, Shanghai, China (approval no. 2023-004). The mice were

given free access to food and water and housed under an alternating 12 h light–dark cycle in a room with constant temperature ($22 \pm 1\,°C$) and 50% relative humidity. We used age-matched male mice in all animal experiments to avoid possible shielding of estrogen on heart failure. Both sexes of rats were used to isolate neonatal ventricular cardiomyocytes since it is difficult to separate male and female in neonates. The mice were euthanized in a carbon dioxide ($CO_2$) chamber for experiments.

## TAC mouse model

A heart failure mouse model was established by TAC-induced pressure overload according to previous studies[23,46,47]. Briefly, eight-week-old male C57BL/6 J mice were anaesthetized by continuous administration of inhaled isoflurane (2%) during surgery. Mice were subjected to endotracheal intubation via a 22-gauge plastic catheter, which was then connected to a ventilator. Afterwards, the left chest was opened, and blunt dissection was performed at the proximal portion of the sternum to access the thoracic aorta. A 27-gauge needle was placed on the transverse aorta between the innominate region and the left common carotid artery. Then, 6-0 silk sutures were used to ligate the transverse aorta with the needle. Immediately after the aorta was completely occluded, the needle was removed, and the thoracic cavity was closed. Sham mice underwent the same procedure without ligation. Aortic velocity peak pressure was determined by in vivo echocardiography, and mice with gradients greater than 30 mmHg were used.

## Targeting ability of the nanoparticles in vivo

Four weeks after TAC or sham operation, the mice received intratracheal treatment with 50 µL of PBS, Cy5.5-labelled TP-10@CaP or TP-10@CaP-CTP. The mice were sacrificed at 0.5, 1, 3, 6, and 24 h to harvest heart tissue, which was then imaged with a Spectrum in vivo imaging system (IVIS) (VISQUE, InVivo Elite). Moreover, 1 h after inhalation or 3 h after I.V. injection, the hearts were harvested to generate frozen specimens, which were cut into 6 µm cryosections for immunofluorescence staining. In parallel, CMs were isolated from the mice and fixed for immunofluorescence staining.

## Mouse inhalation treatment

For inhalation treatment, a high-pressure microsprayer aerosolizer (BioJane BioTech, cat. no.: BJ-PW-M) was used to deliver inhalable agents to TAC model mice. Briefly, the mice were anaesthetized by intraperitoneal injection of pentobarbital sodium (30 mg/kg) for nebulization. Before tracheal cannulation, the mouths of the mice were opened with a laryngoscope to visualize the porch of the trachea. The tip syringe of the high-pressure microsprayer was gently fed into the main trachea, and the aerosols were rapidly delivered.

## In vitro TP-10 release from TP-10@CaP and TP-10@CaP-CTP

The in vitro release of TP-10 was measured using the dialysis bag diffusion method. 1 mg of TP-10@CaP and TP-10@CaP-CTP was dispersed in 5 mL of PBS and then incubated in dialysis bags (MWCO = 8–14k). Then, the dialysis bag was immersed in 10 mL of buffer at different pH values (7.4, 6.5 and 5.5) under stirring (300 rpm). Then, 1 mL of media was collected at given time intervals for further analysis and replaced with 1 mL of fresh buffer. The amount of released TP-10 was determined by using HPLC-ESI-MS/MS (EXPEC 5250, Agilent Technologies, Santa Clara, US). The chromatographic conditions are as follow: stationary phase is a reversed phase HPLC-column (ZORBAX RRHD Eclipse XDB-C18, 50 mm × 2.1 mm, 1.8 µm; Agilent Technologies, USA), a mixture of triethylamine-glacial acetic acid buffer solution (pH 3.5) and acetonitrile (87:12, v/v) served as mobile phase and the flow rate was fixed at 0.6 mL/min. For this established method, the limits of detection (LODs) of TP-10 were estimated as the minimum concentration determined with a signal-to-noise ratio of 3 and the limit of

quantitation (LOQ) values taken by signal-to-noise ratio of 10. The LOD and LOQ were determined to be 2.5 and 8 ng/mL, respectively.

## In vivo biodistribution of TP-10@CaP-CTP

To quantify the amount of TP-10 delivered to the myocardium and different organs, TAC model mice following 1 h TP-10@CaP or TP-10@CaP-CTP treatment through inhalation approach were sacrificed. The different organs including heart, liver, spleen, lung, and kidney were collected, washed and then homogenized in 1.0 mL of lysis buffer with superfine homogenizer. Then, 200 µL of tissue lysate was mixed with Triton X-100 under vortexing, following added 1.5 mL of the extraction solution (HCl–IPA). After incubated at −20 °C overnight, the samples were centrifugated and the amount of TP-10 was quantified by HPLC-MS/MS according to the chromatographic conditions described above.

## Adult mouse CMs (AMCMs) isolation

Primary CMs were isolated according to a previously published protocol[48]. Briefly, after sacrifice, EDTA buffer was injected into the right ventricle in situ. The ascending aorta was clamped, and the heart was then transferred to a dish containing EDTA buffer. Next, the left ventricle was perfused ex vivo using EDTA buffer, which was replaced with perfusion buffer when the thrombus was completely removed. Afterward, collagenase buffer containing collagenase II (Worthington, USA), collagenase IV (Worthington, USA) and protease XIV (Sigma-Aldrich, Singapore) was injected into the left ventricle until the digestion was completed. Then, the heart was dissociated by pipetting, and digestion was terminated with stop buffer. The isolated cells were filtered with a 100 µm pore-size strainer and then allowed to settle by gravity for 20 min. After filtration, CM pellets were sequentially resuspended in three intermediate calcium reintroduction buffers to gradually restore the calcium concentration to physiological levels. Then, the precipitates were collected. resuspended in culture medium, and cultured in a humidified 37 °C, 5% $CO_2$ culture incubator for subsequent experiments.

## Adult mouse CFs isolation

When the heart was dissociated and filtered with the 100 µm pore-size strainer, the CFs were found in the supernatant fraction. After 20 min of gravity settling, the supernatant was collected, centrifuged at 300 × g for 5 min, resuspended in fibroblast media, and plated in 6 cm dishes. The cell media was changed after 2 h of culture.

## Neonatal rat ventricular myocytes (NRVMs) isolation

NRVMs were isolated from 1- to 3-day-old Sprague-Dawley rats (Vital River Laboratory Animal Technology Co., Ltd, Beijing, China) using a cardiomyocyte isolation kit (Miltenyi Biotec, #130-098-373) as previously described[46]. The isolated cells were cultured at 37 °C in a humidified 5% $CO_2$ incubator for 2 h. Then, collected the supernatant which containing cardiomyocytes and transferred into another culture dish with Dulbecco's modified Eagle's medium (DMEM, Gibco) supplemented with 10% fetal bovine serum (FBS, Gibco) and 1% penicillin/streptomycin (Gibco) for 24 h. NRVMs were then subjected to subsequent experiments.

## Measurement of intracellular cAMP and cGMP levels

CMs were stimulated with phenylephrine (100 µM) or vehicle for 24 h, following by the treatment of 25 µg/mL CaP-CTP, TP-10@CaP, TP-10@CaP-CTP or vehicle for 1 h. The cAMP and cGMP levels were measured using cAMP and cGMP ELISA kits (Cayman, Ann Arbor, MI, USA) according to the manufacturer's protocol. Briefly, CMs were lysed with 1 mL HCl (0.1 M) for every 35 cm$^2$ surface area, and then the lysates were centrifuged at 1000 × g for 10 min. The supernatants were collected and diluted 1:2 with dilation buffer. The samples and reagents were added to 96-well plates and incubated at 4 °C overnight, followed by five washes with wash buffer. Ellman's reagent was added

to the plate and incubated at room temperature for 2 h in the dark. The plate absorbance was measured at 420 nm using a microplate reader.

Intracellular cAMP and cGMP levels in CFs were measured with a similar procedure. Briefly, CFs were stimulated with TGF-β (10 ng/mL) or vehicle for 24 h, following by the treatment of 25 µg/mL CaP-CTP, TP-10@CaP, TP-10@CaP-CTP or vehicle for 1 h. The subsequent steps were the same as those mentioned above.

## Boyden chamber migration assay

A Boyden chamber migration assay was performed to evaluate the migration of adult mouse CFs (8 µm pore size; Corning, Acton, MA, USA). Briefly, after 24 h of serum starvation, CFs were added to the upper chambers and cultured in serum-free DMEM. DMEM supplemented with 10% FBS and 1% P/S was added to the lower chambers. Then, the CFs were pretreated with 25 µg/mL CaP-CTP, TP-10@CaP, TP-10@CaP-CTP, TP-10@CaP-CTP with 1 µM PKG inhibitor DT-2 (MedChemExpress, Shanghai, China) or vehicle for 2 h, followed by treatment with TGF-β (10 ng/mL) or vehicle. After 24 h of incubation at 37 °C, the cells on the upper surfaces were removed with cotton swabs, and migrated cells on the lower surfaces were fixed and stained with a 0.1% crystal violet solution. For each Boyden chamber, the total cell numbers in six random fields were determined and analyzed.

## Wound healing scratch assay

CFs were seeded into 24-well plates and cultured in DMEM supplemented with 10% FBS and 1% P/S. When the cells reached ~100% confluence, they were serum starved for 24 h, and scratches were made with a 200 µL pipette tip. Then, the cells were treated with 25 µg/mL CaP-CTP, TP-10@CaP, TP-10@CaP-CTP, TP-10@CaP-CTP with 1 µM DT-2 or vehicle for 2 h, followed by 10 ng/mL TGF-β or vehicle. Wound closure was monitored for 24 h.

## CCK-8 assay

CF proliferation was evaluated by using the CCK-8 cell counting kit according to the manufacturer's instructions. Briefly, CFs were seeded in 96-well plates and cultured in DMEM supplemented with 10% FBS and 1% P/S overnight. Then, the CFs were serum starved for 24 h. Next, the CFs were pretreated with 25 µg/mL CaP-CTP, TP-10@CaP, TP-10@CaP-CTP, TP-10@CaP-CTP with 1 µM DT-2 or vehicle for 2 h, followed by treatment with TGF-β (10 ng/mL) or vehicle. After 24 h of treatment, CCK-8 reagent (10 µL) was added to each well, and the plates were incubated at 37 °C. The absorbance was measured at 450 nm using a microplate reader.

## Shortening/relengthening assay of ADCMs

The mechanical properties of the ADCMs were evaluated by a Softedge MyoCam system (IonOptix Corporation, Milton, MA, USA)[49]. Briefly, after isolation from the adult mouse, CMs were placed in a chamber under an inverted microscope (Olympus, IX-70) in contractile buffer (containing 135 mM NaCl, 10 mM HEPES, 1.0 mM MgCl₂, 1.0 mM CaCl₂, 10 mM glucose and 4.0 mM KCl (pH = 7.4)). Then, the CMs were electrically stimulated at 0.5 Hz, and relevant parameters were recorded, including resting cell length, peak shortening (PS), maximal velocity of shortening ($+dL/dt$), maximal velocity of relengthening ($-dL/dt$), time-to-PS (TPS), and time-to-90% relengthening (TR$_{90}$). The parameters were analyzed by IonOptix Softedge software.

## Echocardiography

Echocardiography was performed on TAC model and sham mice at 0, 2, 4, 6, 8 weeks by an expert laboratory technician who was blinded to the mouse information. The mice were anesthetized via continuous administration of inhaled 2% isoflurane. Echocardiography was performed on anesthetized mice using a multimode small animal ultrasound imaging system (Vevo 3100, FUJIFILM Visual Sonics, Canada). M-mode echocardiography on the short-axis was used to assess cardiac systolic function. The E/A ratio and aortic blood flow velocity were determined to evaluate mouse cardiac diastolic function. Pressure gradients were determined at 1 week post TAC operation.

## Lung wet/dry (W/D) ratio

After mouse sacrifice, the bilateral fresh lung tissues were harvested and weighed (wet weight). Then, the lung tissues were incubated in an oven at 60 °C for 48 h and weighed (dry weight).

## Bronchoalveolar lavage fluid (BALF) analysis

One milliliter of PBS was instilled into the trachea and flushed three times to collect BALF. The supernatant was stored after centrifugation (400 g, 10 min). To evaluate alveolar-capillary permeability in the lungs, the total protein level of the BALF was determined by the BCA Protein Assay Kit (Beyotime, #P0012). Inflammatory cytokine levels in the BALF were determined by enzyme-linked immunosorbent assay (ELISA).

## RNA extraction and quantitative RT–PCR

Total RNA was extracted using TRIzol (Invitrogen) and reverse-transcribed to cDNA using the PrimeScript™ RT Reagent Kit (cat. no. DRR037A, Takara, Japan). Gene expression levels were measured by mixing target primers with SYBR Green master mix (Yeasen, #11201ES03), and the reactions were performed with Bio-Rad's CFX96 (Bio-Rad). Each reaction was performed in triplicate. The primers were purchased from Tsingke Biological Technology (Shanghai, China) and are shown in Supplementary Table 2.

## ELISA

Serum and BALF inflammatory cytokine levels were determined by ELISA according to the manufacturer's instructions. The levels of IL-1β were measured using a mouse IL-1β ELISA kit (WELLBI, #EM30300M). The levels of IL-6 were measured using a mouse IL-6 ELISA kit (WELLBI, #EM30325M). The levels of TNF-α were measured using a Mouse TNF-α ELISA kit (WELLBI, #EM30536M). The levels of ANP were measured by a Mouse ANP ELISA kit (WELLBI, #EM30655M). The levels of Complement C3 were measured by a Mouse Complement C3 ELISA kit (WELLBI, #EM30760M). The levels of Complement C3b were measured by a Mouse Complement C3b ELISA kit (Biomatik, #EKC36654). The levels of Complement C3c were measured by a Mouse Complement C3c ELISA kit (Yojanbio, #YJ-E-94812Q). All of the absorbance values were measured using a microplate reader.

## Immunofluorescence staining

Frozen heart sections were fixed with acetone for 10 min at −20 °C. After washing three times with PBS, the sections were blocked with 5% bovine serum albumin (BSA) for 1 h at room temperature and then incubated with primary antibodies overnight at 4 °C. Afterward, the sections were washed with PBS three times and incubated with the corresponding fluorescence-labelled secondary antibodies for 1 h at room temperature. After four washes with PBS, the sections were counterstained with Antifade Mounting Medium with 2-(4-amidino-phenyl)-6-indolecarbamidine dihydrochloride (DAPI) (Beyotime, #P0131). For cell immunofluorescence staining, a similar procedure was performed, except the cell samples were fixed with 4% paraformaldehyde (PFA) (Beyotime, #P0099) for 10 min and permeabilized with 0.5% Triton X-100 (Beyotime, #P0096) for 10 min at room temperature. Images were captured using a confocal microscope (Olympus FV3000, Japan). The primary antibodies that were used were as follows: antibodies against cardiac troponin T (#ab209813, 1: 3000 for cardiomyocytes), antibodies against Col1α1 (#ab270993, 1:2000), antibodies against Vimentin (#ab24525, 1:300) and cardiac troponin T (#ab8295, 1: 2000 for myocardial tissue) were obtained from Abcam (Cambridge, UK); antibodies against α-smooth muscle actin (#19245 T, 1:500) were purchased from Cell Signaling Technology (MA, USA); antibodies against Discoidin domain receptor 2 (#sc-81707, 1:50) was

purchased from Santa Cruz Biotechnology (Dallas, USA); antibodies against CD31 (#DIA-310, 1:200)was purchased from BIOZOL (Eching, Germany); antibodies against Phalloidin (#40736ES75, 1:1000) was purchased from Yeasen (Shanghai, China).

## Western blot analysis

Cells were lysed in lysis buffer containing RIPA buffer (Beyotime, #P0013C), PMSF, protease inhibitors, and phosphatase inhibitors. The heart tissues were mixed with the abovementioned lysis buffer and ground in a homogenizer (Servicebio, #KZ-5F-3D) according to the manufacturer's instructions. Then, the lysates were centrifuged at $14,000 \times g$ (Thermo Fisher Scientific, 75004250) for 20 min at $4\,°C$, and the supernatants were collected for Western blotting. The protein concentrations were determined by BCA assay. Equal amounts of protein were added to a sodium dodecyl sulfate-polyacrylamide gel and separated by electrophoresis. Then, the proteins were transferred to PVDF membranes. After blocking with 5% BSA for 1 h at room temperature, the membranes were incubated with primary antibodies at $4\,°C$ overnight. Next, the membrane was washed three times with TBST and incubated with the HRP-conjugated secondary antibodies for 1 h at room temperature. Afterward, specific bands were imaged using a Bio-Rad detection system (Bio-Rad Laboratories, Hercules, CA, USA). The images were analyzed using ImageJ. The primary antibodies that were used were as follows: antibodies against AMPKα (#2532, 1: 1000), phospho-AMPKα (#2535, 1: 1000), and α-smooth muscle actin (#19245 T, 1:1000) were obtained from Cell Signaling Technology (MA, USA); antibodies against Col1α1 (#ab270993, 1:1000) were purchased from Abcam (Cambridge, UK).

## Statistical analysis and software

All the data are presented as the mean ± SEM. Two-tailed Student's $t$ tests were performed to compare two groups and identify differences. For multiple group comparisons, one-way ANOVA was conducted, followed by Bonferroni's multiple comparisons test. A $P$ value $< 0.05$ was considered to indicate statistical significance. All data analyses were performed with GraphPad Prism 8.0 software (San Diego, CA, USA).

## Reporting summary

Further information on research design is available in the Nature Portfolio Reporting Summary linked to this article.

## Data availability

The authors declare that all data needed to support the finding of this study are presented in the article, the Supplementary Information and the Source Data file. Source data are provided with this paper.

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

## Acknowledgements

This study was supported by the National Science Foundation of China (Grant No. 82001836, 82071933 and 82227803 to X.S., and 82030050, T2394534 to Y.Z.), the National Key Research and Development Program of China (2022YFC3400100 to Y.Z.), the Science and Technology Committee Foundation of Shanghai (grant No. 20JC1418400 and No. 20Y11912000 to Y.Z.), and the Excellent youth cultivation program of Shanghai Sixth People's Hospital (ynyq202203 to J.W.). We thank Dr Yu Zhang for designing Fig. 1.

## Author contributions

H.W., J.W., Y.Z., and X.S. conceived and designed the study. J.W., Y.Z., and X.S. supervised the project and commented on the project. H.W. and J.W. synthesized and characterized the materials. H.W., W.Z., and F.T. conducted in vitro and in vivo experiments. H.X., A.L., W.L., Y.L., N.Z., and X.C. contributed to the discussion. H.W., W.Z., F.T., and J.W. analyzed the data. H.W. and J.W. wrote the paper. All authors discussed the experimental procedures and results.

## Competing interests

The authors declare no competing interests.
