## [Peer Review File · Nature Communications]

REVIEWER COMMENTS

Reviewer #1 (Remarks to the Author):

The paper of Weng et al. deals with the preparation of CaP nanoparticles loaded with TP-10 and then functionalized with CTP to improve the targeting to myocardium. The nanoparticles were studied in vitro and in vivo by inhalation administration.

The paper is interesting and well written and despite it was largely inspired by a previous work (Miragoli et al. STM, 2018) it includes new features which can extend the approach of using inhalable nanoparticles for the treatment of cardiovascular diseases.

Specific comments:

The authors in the conclusion and in the abstract state that this work offers a paradigm for the use of inhalable drug delivery system to treat cardiovascular disease. In my opinion this is not completely true since the work of Miragoli et al. has already done this. I suggest that the authors focus more their attention on the new perspectives (such as the targeting approach) that this work can open in comparison to the previous results. Moreover, also the fact that this work corroborates previous data such as safety of CaP nanoparticles and biodistribution of CaP administered by inhalation are really good results that are important to highlight.

The authors have used a functionalization approach that includes different steps (Figure 1 a). It is important that the authors include in the paper the characterizations (DLS, TEM, EDX, FTIR, payload, surface charge) of the particles prepared in each step. This can give a clear picture of the strategy used and can help the readers to replicate this system.

It is not clear if the authors have performed a previous study to optimize the payload of the particles (both TP-10 and CTP) and if the amount of TP-10 and CTP is the maximum that CaP can reach, or this can be improved. The amount of drugs loaded can change the final characteristics of the nanoparticles, therefore it is important to know how the authors have selected these specific conditions.

Please specify better the conditions used for the dialysis. It is not enough to mention "three times".

Please state amount of water, time, type of membrane and change of water if any.

The authors should include in the paper data about stability of nanoparticles.

Please write what the spectra in Supplementary Figure 2 correspond to. Please include step by step the FT-IR spectra of the nanoparticles to clearly identify the EDC/NHS mechanism of loading.

Please describe the concentration of the nanoparticles used for the DLS and zeta potential measurements and the pH of the suspensions. Surface charge is a feature which is connected to the pH of the medium.

Please include the limit of detection of the methods used to quantify TP-10 and CTP. The efficiency of loading of CaP for both molecules is a parameter that is important to add.

The experiment of drug release at the different pHs was not described in the experimental part.

Please describe better the mechanism of CTP loading through the EDC/NHS approach. Please clarify the order of addition of EDC and NHS reagents.

Is DSPE-PEG-COOH commercially available? If not, please add how it was synthesized or include a reference to a previous work.

Data of surface charge in Supplementary Figure 3 are poorly described. One of the samples reported in this graph (i.e. CaP-CTP) is not described in the text and it is impossible to know how it was prepared. Please improve this part. The authors should be described why all the CaP (bare and functionalized with different molecules) have a similar surface charge. In my opinion the surface charge results do not demonstrate that CTP is attached to the surface of nanoparticles since the values are all similar.

Please describe how the nanoparticles were labeled with Cy5.5. Stability data of these labeled nanoparticles is also necessary.

It is very difficult to say that the size of particles by TEM is around 85 nm. In the TEM image, particles with different size appeared, so I suggest presenting this data as a range of values.

Please indicate how the CaP phase of the particles evolves during the synthesis. FT-IR spectrum of the CaP clearly demonstrates that the particles are crystalline, but it is not clear if a transformation from amorphous to crystalline occurs during the synthesis and functionalization.

Reviewer #2 (Remarks to the Author):

In this study, the authors reported the beneficial effect of PDE10A inhibition on antagonizing pathological cardiac remodeling and heart failure, using nanoparticles loading with TP-10 and labeled with a cardiac-targeting peptide (TP-10@CaP-CTP) via inhalation. The role of PDE10A deficiency and inhibition in pathological remodeling induced by pressure overload has been previously reported. The role of the CaP-CTP nanoparticle in cardiac targeting has also been known. The delivery of peptide-load nanoparticles via inhalation in heart failure has been described in existing literature. Thus, the novelty and significance of this study may be considered somewhat incremental. Given its focus on the effects of PDE10A inhibition and nanoparticle delivery on cardiac health, this study would be better for publication in a journal related to Pharmacology.

Reviewer #3 (Remarks to the Author):

This is an interesting paper that follows on from previous work by other groups on the use of inhalable calcium phosphate nanoparticles for enriched delivery of therapeutics to the heart. Here the authors use calcium phosphate nanoparticles loaded with TP-10 (a selective inhibitor of PDE10A) and surface

functionalised with a cardiac targeting peptide (CTP) in mice subjected to ventricular pressure overload (TAC). Following chronic administration (2.5 mg/kg every other day for a total of 42 days) by intra-tracheal nebulisation to TAC mice, several pathological features of the cardiac phenotype are reversed.

My majors' concerns are:

- The authors emphasize to have provided a new paradigm for the treatment of heart failure, using nebulization-based pulmonary drug delivery via calcium phosphate-based nanoparticles. However, the very same nanoparticle-based system and approach has already been shown to be effective for cardiac targeting by other groups (Di Mauro et al., *Nanomedicine* 2016, Miragoli et al. *Science Translational Medicine* 2018, Modica et al., *Circulation* et al., 2021). Thus, the statements in the abstract and along the text and discussion need to be properly revised without leaving the impression to the readers that the inhalable approach for the treatment of cardiac disease were discovered in this study. Notwithstanding that the relevance of the novel inhalable approach to reach the heart should still be described, I would recommend here to keep the focus on the real new findings and topic of the paper (as referred in lines 55-57; 358-362), which in fact builds on the previous technology and which the authors nicely apply for rescuing TAC-induced failing heart via the TP drug and the CTP targeting peptide.
- What is the encapsulation efficiency and the ratio of loaded vs. free-non encapsulated TP for both TP-10@CaP-CTP and TP-10@CaP?
- Information on product stability is missing and needs to be provided. In particular, since the authors are performing a long-term in vivo study, it should be clearly stated the maximum storage time for the TP-10@CaP-CTP and TP-10@CaP formulations used along their studies. Was the same formulation used throughout the treatment period (42 days) or were new formulations synthesized at any given time?
- How did the Authors determine the amount of CTP to use per amount of CaP? Was there a study done to determine the best CTP on CaP to reach the highest efficiency of cell targeting? Do the Authors know, or can they speculate on the CTP density of CaP?
- Referring to line 106-107, the Authors state "Therefore, we hypothesized that a CTP modification strategy to enable heart-targeted drug delivery, rather than the administration of a high dose of each inhalation medication complications." Are the authors expecting/knowing that the used nanoparticles (here described or others) to be toxic at high dose and thereby a targeting ligand is required? Does the author know the concentration at which the nanoparticles are toxic in vivo?
- Please, provide evidence if there are differences in TP-10 release from CaP depending on whether the nanoparticles are surface functionalized with CTP or not (see Supplementary Figure 4)
- The CTP peptide is expected to enrich the targeting of CaP nanoparticles to cardiac cells. Is there any in vitro study showing such enrichment of TP-10@CaP-CTP vs. TP-10@CaP with specific reference to cardiac and pulmonary cells? Although some in vivo study is provided, it is not clear the effect on selective enrichment. In addition, mice generally show profound pulmonary remodeling at 4 weeks post-

TAC, with an increase in muscularized lung vessels, myofibroblast proliferation, and general lung fibrosis. Since the CTP peptide has been shown to target cardiac fibroblasts (most likely myofibroblasts), it is curious to see that there are no differences at the level of pulmonary biodistribution in TAC mice when exposed to either TP-10@CaP or TP-10@CaP-CTP. This point should be discussed, and additional data is needed to clearly show that there is no pulmonary retention in the pathological setting (data from the IVIS study do not clearly quantify this).

- Is the enhanced cardiac targeting effect of CTP on CaP evident only in the TAC condition or also in the sham normal condition? Figure 2 shows no difference in cardiac accumulation when comparing TP-10@CaP in TAC vs. TP-10@CaP-CTP in sham. Is the CTP effect on targeting to the heart present only in the physiopathological state? This is an important piece of information that is required to be shown. The proper control of TP-10@CaP sham is missing (as well as in all the other studies presented in the manuscript). Also, the TP-10@CaP without CTP peptide is lacking in the iv group but I guess it can be spared in this study.
- Background signal from the free Cy5.5 targeting the lung and heart is not provided.
- How is it that the harvest time of the nanoformulation differs depending on the mode of administration, i.e. inhalation (1 hour) vs. iv (3 hours)? Does this not affect the outcome of the profile? Please comment.
- Figures 2 g,f are difficult to interpret and it is not easy to determine whether the signal originates from the compounds attached to the extracellular membrane or actually from the intracellular compartment. Z-stack analyses from x, y, z view are required to clearly show the effective internalization.
- Despite to a lower extent compared to the iv administration, biodistribution studies still show a relevant accumulation in liver and kidney of TP-10@CaP-CTP when inhaled. Any comparative consideration in regards of the dose used in the current study (2.5 mg/Kg) versus previous studies (0.5 mg/Kg, Modica et al., *Circulation* 2021) where accumulation to such filtrating organs appears lower?
- Any information regarding the effective administered amounts of formulations (in terms of CaP, CTP, and Cy5.5) versus the amount detected per mg in each analyzed organ? Providing a % distribution would help the reader understand the biodistribution. A comparative quantification within myocardium and other organ accumulation should complement the study to better understand the selective biodistribution. A similar comparative biodistribution table quantifying the detected signals in different organs is also missing for the therapeutic study. Since the authors have established an HPLC and MS method of analysis, I wonder if these approaches could be coupled to quantify the delivery of TP compound to the heart and other organs. In fact, it is important to have data showing whether TP-10 reaches the heart and tissues other than the heart after inhalation.
- Cell size measurements (Ref line 214) report only n=33-45 of analyzed cells, which is very low. In general, reliable studies report on hundreds of cells.

- Suppl. Fig. 6 does not include a marker for fibrosis. Therefore, it is difficult to identify the fibrotic areas. In addition, the very same myocardial area (fiber orientation and location in the ventricle) must be provided in all panels. Ref line 234-236. In addition, the number of hearts used for the histological analysis are not indicated in the text.

- Studies performed on cardiac fibroblasts (most likely myofibroblasts) lack adequate controls for cell identity and require quantification of genes such as discoidin domain receptor 2 (DDR2) and α -smooth muscle actin, which are needed to distinguish fibroblasts from myofibroblasts. In addition, as mentioned above, proper controls with TP-10@CaP condition without CTP are missing.

- It is curious that no monitoring was performed during the treatment study. An intermediate echocardiographic analysis is useful to understand the dynamics of the treatment. Also, no quantification of circulating biomarkers of cardiac stress are provided along this long study.

- The echocardiography table need to include additional parameters such as heart rate, LV wall thickness, and cardia output.

- Can the authors provide any rationale for the use of 0.75 mg TP. Was this a reference dose used in other studies of systemic exposure via IV? Please provide the reference

- I would rephrase the sentence in line 284-286 by stating that the iv effect in contrasting TAC HF was less pronounced compared to the inhalation approach.

- Do the authors have any in vitro/in vivo information on the effect of TP-10@CaP-MP on complement activation? In fact, the absence of any activation would support the biocompatibility of the formulation if a systemic exposure is achieved after inhalation. Also, this provide a relevant information as the current doses used in this study (2.5 and 7.5 mg/Kg) appear to be higher than previously used by other groups (0.5 mg/kg, Miragoli et al, Science Translational Medicine 2018, Modica et al, Circulation 2021)

My minors' concerns are:

- Please, provide more details about the type of lung complications to be expected and in reference to the kind of inhalable drugs to be used.

- Ref. to line 148. A reference to the previous protocol of synthesis must be clearly stated while more emphasis to the actual novelties (i.e. loaded drug and surface functionalization with the targeting peptide) shall be provided.

- Line 178 states "The accumulation of the designed nanoparticles in the heart was obviously enhanced by inhalation administration". An explanation of why this is "obvious" needs to be provided.

- Write if same doses were used in the injected and inhaled animals.

- Do you have any data providing the evidence whether pulmonary translocation occurs via intra- or inter-cellular translocation? Reference to line 190

- Please, provide details/reference for the adopted surgery protocol for TAC generation.
- Specify if the analysed ADCMs were from left ventricle or whole heart.
- The effective volume of nebulised formulation needs to be included in the methods.
- Line 431-433 “The potential mechanism by which this CTP binds to the diseased myocardium might mimic the binding of Titin to crystalline α -B, which is highly expressed in the ischemic myocardium”. The meaning of this statement and the link of CTP to Titin is not clear. Please clarify and rephrase the sentence.
- Some typos are present.

Point-by-point responses to reviewers' comments

Response to Reviewer #1

The paper of Weng et al. deals with the preparation of CaP nanoparticles loaded with TP-10 and then functionalized with CTP to improve the targeting to myocardium. The nanoparticles were studied in vitro and in vivo by inhalation administration.

The paper is interesting and well written and despite it was largely inspired by a previous work (Miragoli et al. STM, 2018) it includes new features which can extend the approach of using inhalable nanoparticles for the treatment of cardiovascular diseases.

Response: We thank the reviewer for the critical review and constructive feedback. We have enhanced the manuscript's clarity where confusion arose and performed additional experiments to elaborate the main findings of our work. Please find our detailed responses to your specific comments below.

Specific comments:

Q1. *The authors in the conclusion and in the abstract state that this work offers a paradigm for the use of inhalable drug delivery system to treat cardiovascular disease. In my opinion this is not completely true since the work of Miragoli et al. has already done this. I suggest that the authors focus more their attention on the new perspectives (such as the targeting approach) that this work can open in comparison to the previous results. Moreover, also the fact that this work corroborates previous data such as safety of CaP nanoparticles and biodistribution of CaP administered by inhalation are really good results that are important to highlight.*

Response: Thank you very much for this valuable suggestion, which is highly appreciated. According to your suggestion, we have now revised the relevant descriptions in the abstract and Discussion to highlight the main purpose of our article and the progressiveness compared with previous studies (Page 2 and Page 16-18).

Q2. *The authors have used a functionalization approach that includes different steps (Figure 1 a). It is important that the authors include in the paper the characterizations (DLS, TEM, EDX, FTIR, payload, surface charge) of the particles prepared in each step. This can give a clear picture of the strategy used and can help the readers to replicate this system.*

Response: Thanks for the suggestion. We have added these characterizations, including DLS, TEM, EDX, FTIR, XRD, payload, surface charge of the particles (TP-10@CaP, TP-10@CaP-DSPE-PEG and TP-10@CaP-CTP) prepared in each step (Supplementary Fig. 1, 3 and 4). In addition, X-ray photoelectron (XPS) was evaluated for further analysis of the containing elements (Supplementary Fig. 1h). These results together confirmed the successful preparation of TP-10@CaP-CTP. The results and discussion are also added in the revised manuscript (Page 5-6) and supplementary materials (Page 10,12).

Supplementary Fig. 1. Characterizations of different nanoparticles prepared in each step. **a-c**, TEM images of different intermediate product, including **a**, TP-10@CaP, **b**, TP-10@CaP-DSPE-PEG and **c**, TP-10@CaP-CTP. **(d-f)** EDX patterns of **d**, TP-10@CaP, **e**, TP-10@CaP-DSPE-PEG and **f**, TP-10@CaP-CTP. **g**, Fourier-transform infrared (FT-IR) spectra and **(h)** XPS spectra of different nanoparticles.

Supplementary Fig. 3. Powder X-ray diffraction patterns of different nanoparticles prepared in each step.

Supplementary Fig. 4. Zeta potentials of different prepared nanoparticles in PBS at pH 7.4 (n=3).

Q3. It is not clear if the authors have performed a previous study to optimize the payload of the particles (both TP-10 and CTP) and if the amount of TP-10 and CTP is the maximum that CaP can reach, or this can be improved. The amount of drugs loaded can change the final characteristics of the nanoparticles, therefore it is important to know how the authors have selected these specific conditions.

Response: Thank you for the comment. The loading of TP-10 and the CTP-conjunction amount were optimized before the final product was obtained. In order to provide a clearer explanation of the optimal synthesis of nanoparticles, we have presented the results of TP-10 loading and CTP-conjunction investigation in the revised manuscript (Supplementary Fig. 6). The loading capacity of TP-10 was determined to be a saturation level 37.2% at the ratio of $\text{CaCl}_2/\text{Na}_2\text{HPO}_4$ of 2:3 by high performance liquid chromatography (HPLC) with tandem mass spectrometric (MS/MS) method, and the maximum amount of CTP was determined to be 22.3% (w/w%) when the addition of different CTP amount (0-12 mg) and the amount of TP-10@CaP-DSPE-PEG is fixed at 15 mg. The results and discussion are also added in the revised manuscript (Page 7) and supplementary materials (Page 13).

Supplementary Fig. 6. a, The TP-10 loading and encapsulation efficiency analysis with different ratio of CaCl₂/Na₂HPO₄. **b,** The CTP bonding amount and encapsulation efficiency after mixing of TP-10@CaP-DSPE-PEG (15 mg) and different amount of CTP (n=3).

Q4. Please specify better the conditions used for the dialysis. It is not enough to mention “three times”. Please state amount of water, time, type of membrane and change of water if any. The authors should include in the paper data about stability of nanoparticles.

Response: Thanks. We have added more detailed descriptions of the dialysis process of TP-10@CaP-CTP (Page 20). Meanwhile, we have performed additional experiments in vitro to demonstrate the stability of nanoparticles (Supplementary Fig. 5). The obtained TP-10@CaP-CTP nanoparticles could be well-dispersed in different mediums (water, PBS, and DMEM), and showed a stable mean hydrodynamic diameters and surface charges during 7 d of incubation, indicating the good colloidal stability. These results have been added to the revised manuscript (Page 7).

The dialysis process of nanoparticles is provided as follows:

“The product (TP-10@CaP) was then dialyzed against 500 mL of deionized water (cutoff MW: 3500 Da) for 3 days (replace fresh water every 24 h) and collected for further use”.

Supplementary Fig. 5. In vitro stability of nanoparticles. **a,** the hydrodynamic size of TP-10@CaP-CTP dispersed in different mediums (water, PBS, and DMEM) for 7 days (n=3). **b,** Zeta potentials of TP-10@CaP-CTP dispersions in PBS (pH 7.4) for 7 days (n=5).

Q5. Please write what the spectra in Supplementary Figure 2 correspond to. Please include step by step the FT-IR spectra of the nanoparticles to clearly identify the EDC/NHS mechanism of loading.

Response: Thanks for your kind reminding. We have added the information corresponding to the spectrum. The FT-IR spectra of the stepwise product were provided to certify the desired synthesis (Supplementary Fig. 1g), and amide I and amide II bands at 1644 and 1343 cm⁻¹ were shown in the spectrum of TP-10@CaP-CTP, suggesting the successful formation of amide band through EDC/NHS reaction. We have added the related results and discussion in the revised manuscript (Supplementary

Fig. 1, Page 6).

Supplementary Fig. 1g, Fourier-transform infrared (FT-IR) spectra of different nanoparticles.

Q6. Please describe the concentration of the nanoparticles used for the DLS and zeta potential measurements and the pH of the suspensions. Surface charge is a feature which is connected to the pH of the medium.

Response: Thanks. The concentration of the nanoparticles (0.4 mg/mL) and the pH value (7.4) used for the DLS and zeta potential measurements were stated in the caption of Fig. 1d, Supplementary Fig. 3 and 4.

Q7. Please include the limit of detection of the methods used to quantify TP-10 and CTP. The efficiency of loading of CaP for both molecules is a parameter that is important to add.

Response: Thank you for your nice suggestion. We have added the limit of detection of the methods in the revised manuscript (Page 21). Related to our response to Q3 of yours, the loading/bonding efficiency of TP-10 and CTP were presented in the revised manuscript (Supplementary Fig. 6, manuscript page 7). The loading capacity of TP-10 was determined to be a saturation level 37.2% by HPLC-MS/MS method. Also, the limit of detection (LODs) of the methods (2.5 ng/mL for TP-10) will be covered in our response to Q8 of yours. Meanwhile, the maximum amount of CTP was determined to be 22.3% by BCA assay, and the LODs of this method is 0.23 mg/mL. These results and discussion are also added in the revised manuscript (Page 7 and 21) and supplementary materials (Supplementary Fig. 6).

Q8. The experiment of drug release at the different pHs was not described in the experimental part.

Response: Thanks for your suggestion. The following expressions about experimental process of drug release from TP-10@CaP and TP-10@CaP-CTP at the different pHs were added in the revised Supplementary Materials (Page 2).

“The in vitro release of TP-10 was measured using the dialysis bag diffusion method. 1 mg of TP-10@CaP and TP-10@CaP-CTP was dispersed in 5 mL of PBS and then incubated in dialysis bags (MWCO = 8k-14k). Then, the dialysis bag was immersed in 10 mL of buffer at different pH values (7.4, 6.5 and 5.5) under stirring (300 rpm). Then, 1 mL of media was collected at given time intervals for further analysis and replaced with 1 mL of fresh buffer. The amount of released TP-10 was determined by using HPLC-ESI-MS/MS (EXPEC 5250, Agilent Technologies, Santa Clara, US). The chromatographic conditions are as follow: stationary phase is a reversed phase HPLC-column (ZORBAX RRHD Eclipse XDB-C18, 50 mm × 2.1 mm, 1.8 μm; Agilent Technologies, USA), a mixture of triethylamine-glacial acetic acid buffer solution (pH 3.5) and acetonitrile (87:13, v/v) served as mobile phase and the flow rate was fixed at 0.6 mL/min. For this established method, the limits of detection (LODs) of TP-10 were estimated as the minimum concentration determined with a signal-to-noise ratio of 3 and the limit of quantitation (LOQ) values taken by signal-to-noise ratio of 10. The LOD and LOQ were determined to be 2.5 and 8 ng/mL, respectively.”

Q9. *Please describe better the mechanism of CTP loading through the EDC/NHS approach. Please clarify the order of addition of EDC and NHS reagents.*

Response: Thanks for the comment. We have described the mechanism of CTP bonding through the EDC/NHS approach in the revised manuscript (Page 5-6) together with a newly performed characterizations (as mentioned in our response to Q2 and Q5) for clearer demonstration of the successful CTP modification. Briefly speaking, TP-10@CaP nanoparticles were firstly modified with DSPE-conjugated polyethylene glycol acid (DSPE-PEG-COOH). In this step, the phosphate head of DSPE coordinates with the calcium of TP-10@CaP nanoparticles, introducing carboxyl groups to enable further modification. Then, the carboxyl-functionalized CaP (TP-10@CaP-COOH) was reacted with the amino group of CTP peptide by the EDC/NHS coupling reaction.

Q10. *Is DSPE-PEG-COOH commercially available? If not, please add how it was synthesized or include a reference to a previous work.*

Response: Thanks for the comment. DSPE-PEG-COOH is commercially available and purchased from the Sigma-Aldrich Trading Co., Ltd. (Shanghai, China). We have added the information of materials and reagents in the revised supplementary information (Page 2).

Q11. *Data of surface charge in Supplementary Figure 3 are poorly described. One of the samples reported in this graph (i.e. CaP-CTP) is not described in the text and it is impossible to know how it was prepared. Please improve this part. The authors should be described why all the CaP (bare and functionalized with different molecules) have a similar surface charge. In my opinion the surface charge results do not demonstrate that CTP is attached to the surface of nanoparticles since the values are all similar.*

Response: Thanks for spotting the discrepancies regarding surface charge of different

nanoparticles. We have now improved the description of surface charge in Supplementary Figure 3 (now Supplementary Fig. 4 in the revised manuscript, see below). Meanwhile, the CaP-CTP nanoparticles were prepared by the same process but replacing the TP-10@CaP with CaP nanoparticles.

For the surface charge of different nanoparticles, we agree that the difference in surface charges of all nanoparticles is not significant, even though they are different. Based on your comment of Q6, which reminds of the pH of the medium may have important effect on the surface charge. Unfortunately, we overlooked this issue in our previous measurements. Thus, we strictly controlled the pH of PBS medium (7.4) and re-evaluated the surface charge of different nanoparticles. As shown in Supplementary Fig. 4, the TP-10@CaP and bare CaP showed similar surface charge with about -20 mV due to the negligible impact of drug encapsulation on surface charge. The conjunction of DSPE-PEG-COOH on CaP nanoparticles changed the zeta potential to -34.8 ± 1.9 mV, while the formation of TP-10@CaP-CTP with CTP modification weakened the zeta potential to -12.6 ± 1.2 mV. The serial changes on zeta potential of each step further indicated a successive conjugation of COOH, PEG, and CTP due to the negative potential of COOH and positive potential of CTP, respectively. We have added the related results and discussion in the revised manuscript (Page 6).

Supplementary Fig. 4. Zeta potentials of different prepared nanoparticles in PBS at pH 7.4 (n=3).

Q12. Please describe how the nanoparticles were labeled with Cy5.5. Stability data of these labeled nanoparticles is also necessary.

Response: Thanks for the comment. We have now described the experimental process and mechanism of the Cy5.5 labeled nanoparticles. The TP-10@CaP-CTP and TP-10@CaP were labeled with the Cy5.5 through electrostatic adsorption due to the negative potential of the two nanoparticles and positive potential of Cy5.5. The following expressions about experimental process were added in the revised manuscript (Page 21), which reads “Meanwhile, the CaP-CTP nanoparticles were prepared by the same process but replacing the TP-10@CaP with CaP nanoparticles. In addition,

Cy5.5-labeled TP-10@CaP-CTP or TP-10@CaP were also prepared through electrostatic adsorption. Briefly, 10 mL of TP-10@CaP-CTP or TP-10@CaP dispersion (1 mg/mL) were mixed 1 mL of amine-Cy5.5 DMSO solution and stirred at room temperature overnight in the dark. The obtained Cy5.5-labeled TP-10@CaP-CTP or TP-10@CaP were collected by centrifugation, washed with deionized water and ethanol, followed by dispersing in PBS for further *in vivo* imaging”.

Then, the stability of Cy5.5-labeled TP-10@CaP-CTP or TP-10@CaP were evaluated by monitoring the UV-Vis-NIR absorption spectroscopy (Supplementary Fig. 8). After dispersed in PBS for 7 days, no obvious absorption peak change of Cy5.5 was observed on UV-Vis-NIR spectra, indicating that the labeling of Cy5.5 was stable. We have now improved the relevant descriptions in the revised manuscript (Page 8).

Supplementary Fig. 8. The UV-vis absorption spectra of TP-10@CaP-CTP, free Cy5.5 and Cy5.5-labeled TP-10@CaP-CTP in PBS for 7 days.

Q13. It is very difficult to say that the size of particles by TEM is around 85 nm. In the TEM image, particles with different size appeared, so I suggest presenting this data as a range of values.

Response: Thanks. According to your suggestion, we have added the standard deviation (SD) to present this data as a range of values (85 ± 3.7 nm) in the revised manuscript (Page 6).

Q14. Please indicate how the CaP phase of the particles evolves during the synthesis. FT-IR spectrum of the CaP clearly demonstrates that the particles are crystalline, but it is not clear if a transformation from amorphous to crystalline occurs during the synthesis and functionalization.

Response: Thanks for this question. We have now performed additional experiments to clarify this issue. To investigate the CaP phase of the particles evolves during the synthesis, X-ray diffraction (XRD) analysis was performed (Supplementary Fig. 3). Both these four types of nanoparticles showed similar XRD signals, which indicates that the XRD pattern of CaP phase was not affected surface modification and drug

loading. Meanwhile, the broad peak at around $2\theta = 31^\circ$ indicating an amorphous phase of CaP, while the other peaks are attributed to calcite (CaCO_3) (JCPDS 05-0586), an anhydrous phase of calcium carbonate (*J. Mater. Chem. B* 2015, 3, 7347-7354; *ACS Nano* 2019, 13, 12, 13985-13994). It should be noted that the XRD of CaP nanoparticles is difficult to monitor during the synthesis process due to the rapid formation of CaP in the mixture of CaCl_2 and Na_2HPO_4 .

Supplementary Fig. 3. Powder X-ray diffraction patterns of different nanoparticles prepared in each step.

Response to Reviewer #2

In this study, the authors reported the beneficial effect of PDE10A inhibition on antagonizing pathological cardiac remodeling and heart failure, using nanoparticles loading with TP-10 and labeled with a cardiac-targeting peptide (TP-10@CaP-CTP) via inhalation. The role of PDE10A deficiency and inhibition in pathological remodeling induced by pressure overload has been previously reported. The role of the CaP-CTP nanoparticle in cardiac targeting has also been known. The delivery of peptide-load nanoparticles via inhalation in heart failure has been described in existing literature. Thus, the novelty and significance of this study may be considered somewhat incremental. Given its focus on the effects of PDE10A inhibition and nanoparticle delivery on cardiac health, this study would be better for publication in a journal related to Pharmacology.

Response: We thank the reviewer for the critical review and constructive feedback. As evaluated by Reviewer 1 and 3, our work based on the previous technology and applied for rescuing TAC-induced failing heart via the TP-10 and the CTP targeting peptide. Also, this work corroborated previous data and confirmed the feasibility of inhalation approach for cardiac drug delivery. We have enhanced the manuscript's clarity where confusion arose and performed additional experiments to elaborate the main findings of our work in this round of revision. Now, please allow us to rephrase the novelty and significance of this study as follows for your check:

(1) Providing a promising delivery method for clinical application of TP-10.

Indeed, the role of PDE10A deficiency and inhibition in pathological remodeling

induced by pressure overload has been reported. However, research focusing on effective drug delivery for pathological myocardial therapy is still in its infancy, which is crucial for improving its bioavailability. We reported a promising delivery method for clinical application of TP-10 in prevention of heart failure. In addition to utilizing the role of TP-10 in myocardial remodeling, one of the main objectives of our work is to provide a delivery method of TP-10 for improving the efficiency. Our investigations represent a conceptual design of targeted TP-10 delivery system that couples the controllable release with the biocompatible feature of CaP mineral for active target delivery. Active target delivery of TP-10 on failing myocardium in vivo has been verified, as well as further confirmed the therapeutic mechanism.

(2) Proposing a heart-targeted peptide CTP engineering strategy to promote the drug accumulation on failing myocardium. We agree that the CTP utilized in targeted ischemic heart diseases is not a new concept, with other proofs-of-concept previously reported^{1, 2}, while no study has demonstrated the application of CTP in chronic heart failure. In this study, we not only verified the feasibility of CTP in chronic heart failure, but also demonstrated this targeting capacity only presented in the pathological state of heart. Furthermore, we found CTP was also able to target to fibrotic area of the failing heart, while is invalid on lung epithelial cell or lung fibroblasts. As commented by Reviewer 1 these results may provide a perspective for exploring myocardium target therapy by CTP targeted design (this content was provided in the Discussion Page 16 and 17).

(3) The lung complication of long-term inhalation treatment has been clarified. Previous studies regarding inhalable peptide-load CaP via inhalation in heart failure for heart drug delivery^{3, 4} given us great inspiration, while the long-term utilization induced lung complications were far from verification. Clinically speaking, adverse drug-induced respiratory reactions is a critical limitation in inhalable drugs for systemic delivery. For avoiding lung complications, we used a heart targeted peptide (CTP) for promoting specific heart drug delivery in this work, which can reduce dose of each inhalation medication. On the basis of verifying the feasibility of the designed nanoparticles administered by inhalation (including biodistribution, safety, cardiac targeting, etc.), we performed an 18-week monitors on the lung condition after inhalation of our designed TP-10@CaP-CTP nanoparticle in this work. After this long-term inhalation treatment, no obvious lung injury was observed (Fig. 7). This may be due to the targeting effect of failing myocardium targeting peptide CTP in nanoparticles, thus achieving greater myocardium accumulation reduce the therapeutic dosage and attenuate pulmonary burden during nebulization administration. In contrast, TP-10@CaP nanoparticles without CTP modification required a 3-fold higher dose to exerts therapeutic effects equivalent to those exerted by TP-10@CaP-CTP (shown in Supplementary Fig. 15). These results evaluated the biosafety of inhalable CaP, which further corroborated the previous studies^{3, 4}, and verified the safety of our designed TP-10@CaP-CTP on long-term management of chronic heart failure, as well as the lung complications. (This content was provided in the Discussion Page 17 and 18).

Take all together, the TP-10@CaP-CTP nanomedicine developed in this study provides a promising strategy for long-term management of chronic heart failure with

enhanced cardiopulmonary circulation delivery efficiency and therapeutic efficacy while minimizing unintended lung complication of inhalation delivery such as wheezing, bronchoconstriction, and losses in lung function.

Response to Reviewer #3

This is an interesting paper that follows on from previous work by other groups on the use of inhalable calcium phosphate nanoparticles for enriched delivery of therapeutics to the heart. Here the authors use calcium phosphate nanoparticles loaded with TP-10 (a selective inhibitor of PDE10A) and surface functionalised with a cardiac targeting peptide (CTP) in mice subjected to ventricular pressure overload (TAC). Following chronic administration (2.5 mg/kg every other day for a total of 42 days) by intra-tracheal nebulisation to TAC mice, several pathological features of the cardiac phenotype are reversed.

Response: We thank the reviewer for recognizing our work's significance and providing insightful feedback that has enabled us to enhance the rigor of the manuscript. We have also performed new experiments and carefully revised the manuscript according to the reviewer's suggestions. Our detailed responses can be found in the following.

Major comments:

Q1. *The authors emphasize to have provided a new paradigm for the treatment of heart failure, using nebulization-based pulmonary drug delivery via calcium phosphate-based nanoparticles. However, the very same nanoparticle-based system and approach has already been shown to be effective for cardiac targeting by other groups (Di Mauro et al., Nanomedicine 2016, Miragoli et al. Science Translational Medicine 2018, Modica et al., Circulation et al., 2021). Thus, the statements in the abstract and along the text and discussion need to be properly revised without leaving the impression to the readers that the inhalable approach for the treatment of cardiac disease were discovered in this study. Notwithstanding that the relevance of the novel inhalable approach to reach the heart should still be described, I would recommend here to keep the focus on the real new findings and topic of the paper (as referred in lines 55-57; 358-362), which in fact builds on the previous technology and which the authors nicely apply for rescuing TAC-induced failing heart via the TP drug and the CTP targeting peptide.*

Response: We highly appreciate for your kind comments and very nice suggestions. We have enhanced our Abstract and Discussion sections in the revised manuscript to make sure the advances of this work stand out (Page 2, see below). We are aware of the suggested studies by other groups and have now discussed it in the Introduction section (Page 3). Indeed, these leading works have given us great inspiration and encouraged us to continue confirming the applicability of the inhalable approach for the treatment of cardiovascular disease. As described by the reviewer, based on the previous works our work further presented a cardiac targeting peptide-modification strategy to enhance the accumulation of a potential drug (TP-10) in pathological myocardium, thereby exploring better therapeutic effects.

“...Inspired by the high efficiency of inhalation administration for the heart drug

delivery, we here present an inhalable a cardiac targeting peptide (CTP) modified calcium phosphate (CaP) nanoparticle for deliver TP-10 (a selective inhibitor of PDE10A). Equipped with excellent nebulization properties, The CTP modification significantly promoted nanoparticles targeting to cardiac myocytes and fibroblasts under pathological state of heart failure.... Through demonstrated the targeting capacity of CTP and further verified the biosafety of inhalable CaP nanoparticles in lung during long-term medication, this work provides a perspective for exploring myocardium target therapy and presents a potential promising clinical strategy for the long-term management of chronic heart failure....”

Q2. *What is the encapsulation efficiency and the ratio of loaded vs. free-non encapsulated TP for both TP-10@CaP-CTP and TP-10@CaP?*

Response: Thank you for the comment. We have measured the encapsulation efficiency (EE%) of TP-10 and CTP for TP-10@CaP or TP-10@CaP-CTP, respectively. The EE% of TP-10 was calculated by the equation given as:

$$EE\% = \frac{M_T - M_F}{M_T}$$

Where M_T is the total mass of TP-10 or CTP added initially, M_F is the mass of TP-10 or CTP in the supernatant measured by HPLC-MS/MS or BCA assay.

Similar with the loading capacity, the EE% of TP-10 was also influenced by the concentration of CaCl_2 and Na_2HPO_4 . We used an encapsulation method for TP-10@CaP to that reported by Miragoli et al. where more than half of the TP-10 was mixed with the calcium and the Na_2HPO_4 solutions to yield a significantly great EE (Supplementary Fig. 6). Conversely, the EE% of the TP-10 peaked at $51.4 \pm 3.1\%$ at the ratio of $\text{CaCl}_2/\text{Na}_2\text{HPO}_4$ of 2:3. Correspondingly, the ratio of loaded vs. free-non encapsulated TP-10 is calculated to be 1.03. In light of optimal loading capacity, concentration and encapsulation efficiency that would be required for future studies, we chose the nanoparticle formulation with a $\text{CaCl}_2/\text{Na}_2\text{HPO}_4$ ratio of 2:3 for all subsequent experiments.

In addition, the encapsulation efficiency of CTP was determined by a BCA method. The optimized EE% of CTP in the TP-10@CaP-CTP was $62.9 \pm 0.9\%$ when the amount of TP-10@CaP-DSPE-PEG and CTP during the preparation process are 15 mg and 8 mg, respectively. Similarly, the ratio of bonded vs. free-non bonded CTP is calculated to be 1.16 [4.3/(8-4.3)]. These results and discussion have now been included in the revised manuscript and supplementary materials (Page 7).

Supplementary Fig. 6. a, The TP-10 loading and encapsulation efficiency analysis with different ratio of $\text{CaCl}_2/\text{Na}_2\text{HPO}_4$. **b**, The CTP bonding amount and encapsulation efficiency after mixing of TP-10@CaP-DSPE-PEG (15 mg) and different amount of CTP (n=3).

Q3. Information on product stability is missing and needs to be provided. In particular, since the authors are performing a long-term in vivo study, it should be clearly stated the maximum storage time for the TP-10@CaP-CTP and TP-10@CaP formulations used along their studies. Was the same formulation used throughout the treatment period (42 days) or were new formulations synthesized at any given time?

Response: Thanks for your valuable suggestion. We have performed additional experiments in vitro to demonstrate the stability of nanoparticles (Supplementary Fig. 5). The obtained TP-10@CaP-CTP nanoparticles could be well-dispersed in different mediums (water, PBS, and DMEM), and showed a stable mean hydrodynamic diameters and surface charges during 7 d of incubation, indicating the good colloidal stability under physiological conditions.

According to the reviewer's suggestions, we have evaluated the maximum storage time for the TP-10@CaP-CTP and TP-10@CaP formulations, and we find the size of TP-10@CaP-CTP and TP-10@CaP decreased after 20 days of incubation. This suggests the activity of nanoparticles is indeed largely maintained in this period. Although this time is lower than the treatment period (42 days) we used in animal experiments, each administration were new formulations synthesized in our study. These results have been added to the revised manuscript (Page 7).

Supplementary Fig. 5. In vitro stability of nanoparticles. **a**, the hydrodynamic size of TP-10@CaP-CTP dispersed in different mediums (water, PBS, and DMEM) for 7 days (n=3). **b**, Zeta potentials of TP-10@CaP-CTP dispersions in PBS (pH 7.4) for 7 days (n=5).

Q4. How did the Authors determine the amount of CTP to use per amount of CaP? Was there a study done to determine the best CTP on CaP to reach the highest efficiency of cell targeting? Do the Authors know, or can they speculate on the CTP density of CaP?

Response: Thanks for the comment. The determination of CTP amount is as described earlier in our response to Q2 of yours. The amount of CTP in TP-10@CaP-CTP were

analyzed by the BCA assay. An enhanced trend in bonding amount of CTP was obtained when increasing the addition of CTP (TP-10@CaP-DSPE-PEG: 15 mg), the maximum amount of CTP was determined to be 22.3% (w/w%) when the addition of CTP amount is 8 mg (Supplementary Fig. 6, as described earlier in our response to Q2 of yours).

We have now performed more experiments in vitro to evaluate the effect CTP bonding on cell targeting. The intracellular uptake of TP-10@CaP-CTP at various CTP bonding amount was evaluated by confocal laser scanning microscopy (CLSM) using fluorescein Cy5.5-labeled TP-10@CaP-CTP. The intensity of Cy5.5 kept increasing when increasing the CTP amount on TP-10@CaP-CTP, further verifying that the efficiency of cell targeting augmented with the increased CTP bonding amount (Supplementary Fig. 12). As expected, the highest efficiency of cell targeting is achieved by using the TP-10@CaP-CTP with the highest CTP amount on TP-10@CaP-CTP (22.3%, w/w%). These results have been added to the revised manuscript (Page 10). Based on the above data, we can predict the density of CTP in TP-10@CaP-CTP is about 223 mg/g.

Supplementary Fig. 12. In vitro determination of targeting capacity of TP-10@CaP-CTP with different amounts of CTP. **a**, Representative images of the intracellular uptake of Cy5.5 labeled TP-10@CaP-CTP with different amounts of CTP (w/w%) in PE induced (pretreated with 100 μ M PE for 24 h) neonatal rat ventricular myocytes (NRVMs). Scale bar, 50 μ m. **b**, The quantitative assay of fluorescence signals from NRVMs. $n = 3$. Results are presented as mean \pm SD. Statistical analysis was performed using one-way ANOVA with Bonferroni multiple-comparison correction.

Q5. Referring to line 106-107, the Authors state “Therefore, we hypothesized that a CTP modification strategy to enable heart-targeted drug delivery, rather than the

administration of a high dose of each inhalation medication complications.” Are the authors expecting/knowing that the used nanoparticles (here described or others) to be toxic at high dose and thereby a targeting ligand is required? Does the author know the concentration at which the nanoparticles are toxic in vivo?

Response: We are sorry for the confusion caused due to unclear description. In this sentence, we mainly want to express that the increasing of cardiac accumulation is beneficial for improving the effectiveness of treatment, thereby avoiding the potential pulmonary complications that may occur during multiple doses of medication. Because, in previous studies, there is a tendency to cause bronchospasm, wheezing, reduction in lung function, and other lung complications in long-term utilization of inhalable drugs⁵⁻¹¹. In addition, some studies demonstrated hydroxyapatite nanoparticles, a crystalline phase of calcium phosphate will lead to cell apoptosis and possesses anti-tumor effect¹². In our design, CTP modification was performed for subsequently achieving an active-targeting performance, which can facilitate specific pathological myocardium accumulation of the nanoparticles, as well as avoiding lung complications and reduce potential cytotoxicity of inhalable CaP nanoparticles. We agree with the viewpoint of the reviewer that future work should pay more attention the safe or toxic dose of nanoparticles used to enhance the broad interest. We have revised this sentence for clearer description (Page 4).

“Therefore, we hypothesized that a CTP modification strategy to enable heart-targeted drug delivery for improving the efficiency, rather than the administration of a high dose of each inhalation medication complications or multiple-dose, which might facilitate specific pathological myocardium accumulation of drug and thus overcome the current obstacles of lung complications.”

Q6. *Please, provide evidence if there are differences in TP-10 release from CaP depending on whether the nanoparticles are surface functionalized with CTP or not (see Supplementary Figure 4)*

Response: Thanks for raising this issue. In the revised manuscript, we have evaluated the in vitro release of TP-10 from TP-10@CaP at different pHs (Supplementary Fig. 7). Similarly, only $\approx 9.5\%$ of TP-10 was released from the TP-10@CaP nanoparticles at pH 7.4. Compared with the profile of TP-10 from TP-10@CaP-CTP, the release ratio from TP-10@CaP-CTP dramatically increased to 46.2% at pH 6.5 and 67.7% at pH 5.5 within 12 h. By contrast, the released amount from TP-10@CaP over a span of 4 h increased to $\approx 49.3\%$ and $\approx 72.5\%$ at pH 6.5 and 5.5, respectively. It follows that the surface modification of DSPE-PEG and CTP may possess a protective effect on the internal CaP structure and thus slowing down the rate of degradation to a certain extent. The property of pH-responsive release of TP-10 will benefit the application of TP-10@CaP-CTP nanoparticles for heart failure therapy. These results have been added to the revised manuscript (Page 7).

Supplementary Fig. 7. The release profiles of TP-10 from **a**, TP-10@CaP-CTP and **b**, TP-10@CaP at different pHs (n = 3).

Q7. The CTP peptide is expected to enrich the targeting of CaP nanoparticles to cardiac cells. Is there any *in vitro* study showing such enrichment of TP-10@CaP-CTP vs. TP-10@CaP with specific reference to cardiac and pulmonary cells? Although some *in vivo* study is provided, it is not clear the effect on selective enrichment. In addition, mice generally show profound pulmonary remodeling at 4 weeks post-TAC, with an increase in muscularized lung vessels, myofibroblast proliferation, and general lung fibrosis. Since the CTP peptide has been shown to target cardiac fibroblasts (most likely myofibroblasts), it is curious to see that there are no differences at the level of pulmonary biodistribution in TAC mice when exposed to either TP-10@CaP or TP-10@CaP-CTP. This point should be discussed, and additional data is needed to clearly show that there is no pulmonary retention in the pathological setting (data from the IVIS study do not clearly quantify this).

Response: Thanks for raising this issue. We have now performed more *in vitro* experiments regarding the enrichment of TP-10@CaP-CTP and TP-10@CaP in neonatal rat ventricular myocytes (NRVMs), cardiac fibroblasts (CFs), mouse lung epithelial cells (MLE-12), and mouse pulmonary fibroblasts (MLFs). We found that the targeting capacity of CTP presented in phenylephrine (PE) stimuli NRVMs in a CTP-amounts dependent manner (Supplementary Fig. 12a, b). Besides, there was no overt difference of the intracellular uptake of nanoparticles between TP-10@CaP treated PE induced NRVMs and TP-10@CaP-CTP treated normal NRVMs (Supplementary Fig. 12c, d). These results indicated that CTP only targeted to the PE induced hypertrophic NRVMs. Moreover, similar results were observed in CFs, our data shown that CTP only targeted to TGF- β induced CFs, but not normal CFs (Supplementary Fig. 14).

For MLE-12 and MLFs, neither of which was able to be targeted by CTP (Supplementary Fig. 22 and Fig. 23, see below). These results suggested that CTP only exerts its targeting capacity in the pathological state of myocardial cells and cardiac fibroblasts rather than lung derived epithelial cells or fibroblasts. This may explain that no differences at the level of the distribution of TP-10@CaP and TP-10@CaP-CTP in the lung of TAC-operated mice after inhalation administration.

Moreover, we further increased the observation time of *ex vivo* imaging to 36 h, and no fluorescence signal was observed on lung tissue at that time point in every group

(Supplementary Fig. 10, see below), which suggested no retention of the nanoparticles in the lung. These results have been added to the revised manuscript (Page 8, 10) and make a discussion on this part of content (Page 18).

Supplementary Fig. 12. In vitro determination of targeting capacity of TP-10@CaP-CTP. **a**, Representative images of the intracellular uptake of Cy5.5 labeled TP-10@CaP-CTP with different amounts of CTP (w/w%) in PE induced (pretreated with 100 μ M PE for 24h) neonatal rat ventricular myocytes (NRVMs). Scale bar, 50 μ m. **b**, The quantitative assay of fluorescence signals from NRVMs. $n = 3$. **c**, Representative images of the intracellular uptake of Cy5.5 labeled TP-10@CaP in PE induced NRVMs and TP-10@CaP-CTP in normal NRVMs. Scale bar, 50 μ m. **d**, The quantitative assay of fluorescence signals from NRVMs after different treatments. $n = 3$. Results are presented as mean \pm SD. Statistical analysis was performed using one-way ANOVA with Bonferroni multiple-comparison correction.

Supplementary Fig. 14. In vitro determination of targeting capacity of TP-10@CaP and TP-10@CaP-CTP in AMCFs. **a**, Representative images of the intracellular uptake of Cy5.5 labeled TP-10@CaP and TP-10@CaP-CTP in TGF-β induced AMCFs. Scale bar, 50μm. **b**, The quantitative assay of fluorescence signals from AMCFs. n=3. Results are presented as mean ± SD. Statistical analysis was performed using one-way ANOVA with Bonferroni multiple-comparison correction.

Supplementary Fig. 22. In vitro determination of targeting capacity of TP-10@CaP and TP-10@CaP-CTP in MLE-12. **a**, Representative images of the intracellular uptake of Cy5.5 labeled TP-10@CaP and TP-10@CaP-CTP in MLE-12. Scale bar, 20 μm. **b**, The quantitative assay of fluorescence signals from MLE-12. n = 3. Results are presented as mean ± SD. Statistical analysis was performed using one-way ANOVA with Bonferroni multiple-comparison correction.

Supplementary Fig. 23. In vitro determination of targeting capacity of TP-10@CaP and TP-10@CaP-CTP in MLF. **a**, Representative images of the intracellular uptake of Cy5.5 labeled TP-10@CaP and TP-10@CaP-CTP in TGF- β induced MLF. Scale bar, 50 μ m. **b**, The quantitative assay of fluorescence signals from MLF. $n = 3$. Results are presented as mean \pm SD. Statistical analysis was performed using one-way ANOVA with Bonferroni multiple-comparison correction.

Supplementary Fig. 10. Ex vivo fluorescence images of lung tissue from mice received Cy5.5 labeled TP-10@CaP or TP-10@CaP-CTP treatment through inhalation approach or I.V. injection in TAC and Sham mice at 36 h. $n = 3$.

Q8. *Is the enhanced cardiac targeting effect of CTP on CaP evident only in the TAC condition or also in the sham normal condition? Figure 2 shows no difference in cardiac accumulation when comparing TP-10@CaP in TAC vs. TP-10@CaP-CTP in sham. Is the CTP effect on targeting to the heart present only in the physiopathological state? This is an important piece of information that is required to be shown. The proper control of TP-10@CaP sham is missing (as well as in all the other studies presented in the manuscript). Also, the TP-10@CaP without CTP peptide is lacking in the iv group but I guess it can be spared in this study.*

Response: Thanks for the comment. Yes, our result indicated that the cardiac targeting effect of CTP only perform in pathological conditions, while did not show significant

targeting effect in normal condition. Kanki et al. demonstrated that CTP targeted to the ischemic myocardium, while is invalid in the non-schemic myocardium¹. Our finding was consisted with their study. They also raised the hypothesis that CTP (as identified with partial sequence similarity to the cytoskeletal protein Titin) might preferentially bind to alpha-B crystalline, which is highly combined with cardiac Titin under ischemic state. This discussion has now been included in the revised manuscript (Page 17).

According to your kind suggestion, we have supplemented the TP-10@CaP in Sham group in all of the nanoparticle accumulation experiments (Fig. 2), as well as the in vitro therapy experiments (Supplementary Fig. 12). We found that the accumulation of TP-10@CaP nanoparticles in failing myocardium was less than that of TP-10@CaP-CTP nanoparticles. Furthermore, the therapeutic effect of TP-10@CaP was not significant when compared with the TP-10@CaP-CTP treatment (Fig. 3). These data further confirmed the results that the targeting capacity of CTP to the heart was presented only in the physiopathological state. We have added the related results and discussion in the revised manuscript (Page 8 and 9).

Q9. Background signal from the free Cy5.5 targeting the lung and heart is not provided.

Response: We have provided the background signal of free Cy5.5 in the lung and heart (Supplementary Fig. 9). Similar to the Cy5.5 labeled nanoparticles, free Cy5.5 passed pulmonary barrier resulted in a decaying signal in the lungs. However, the peak signal of the heart occurred in 0.5 h and decayed rapidly in the following time. This may be due to the rapid diffusion and/or metabolism of free Cy5.5, which led to faster pass of the pulmonary barrier but was unable retention in the myocardium. Moreover, no significant differences of free Cy5.5 accumulation in the heart and lung between Sham and TAC groups. We have added the related results and discussion in the revised manuscript (Page 8).

Supplementary Fig. 9. Background signal of free Cy5.5 in the lung and heart (n=3).

Q10. How is it that the harvest time of the nanoformulation differs depending on the mode of administration, i.e. inhalation (1 hour) vs. iv (3 hours)? Does this not affect the outcome of the profile? Please comment.

Response: Thank you for your comment. We selected the harvest time of tissues is mainly based on the accumulation of nanoparticle. According to the data presented in Fig 2a and b, the highest accumulation of TP-10@CaP-CTP after inhalation administration was observed at 1 h, while I.V. administration was at 3 h. Therefore, we chose these time points as the comparison time point, so as to compare the highest cardiac enrichment of the two drug delivery approaches. In addition, due to the excellent cardiac delivery efficiency of inhalation, the accumulation of TP-10@CaP-CTP nanoparticle in the heart is still higher than that of intravenous injection at 3 h. Thus, the choice of this different time points will not affect the experimental results and conclusion, that is, inhalation administration is more efficient than I.V. administration.

Q11. Figures 2 g,f are difficult to interpret and it is not easy to determine whether the signal originates from the compounds attached to the extracellular membrane or actually from the intracellular compartment. Z-stack analyses from x, y, z view are required to clearly show the effective internalization.

Response: Thanks for this nice suggestion. We have now provided the Z-stack analyses for Figure 2f and 2g in Supplementary Fig. 25. And confirmed that the signal of the compounds originated from the intracellular compartment.

Supplementary Fig. 25. Z-stack analyses from x, y, z view.

Q12. Despite to a lower extent compared to the iv administration, biodistribution studies still show a relevant accumulation in liver and kidney of TP-10@CaP-CTP when inhaled. Any comparative consideration in regards of the dose used in the current study (2.5 mg/Kg) versus previous studies (0.5 mg/Kg, Modica et al., Circulation 2021) where accumulation to such filtrating organs appears lower?

Response: Thanks for raising this question. Based on the previous studies on nanomedicine and our current research, although inhalation administration can reduce liver accumulation of nanoparticles, it cannot avoid their uptake. The accumulation of CaP nanoparticles in the liver can be attributed to absorption by the mononuclear

phagocyte system; those accumulated in the kidney are likely to arise from renal excretion. We have now performed more experiments to demonstrate the in vivo biodistribution of TP-10@CaP-CTP. More specific discussions will be provided in our response to Q13 later.

To briefly answer the second part, we are aware of the suggested study by Modica et al. The dose of TP-10@CaP-CTP in our approach is different from that of Modica et al.: First of all, Although the drug carriers and delivery methods used in both studies, the obtained nanoformulations (peptide-loaded CaP formulation by Modica et al) are different. Due to the different pharmacokinetics, pharmacokinetics, and biological distribution behaviors of different nanocomposites, there may be differences in the dosage of administration. Secondly, CaPs loaded with a therapeutic mimetic peptide was used to the treatment of cardiovascular diseases by Modica et al, while TP-10 was used as the main therapeutic ingredient for treatment. There are differences in the effective dosage of therapeutic drugs. Although our dose used is higher than the reported work, we further validate corroborates previous data such as safety of CaP nanoparticles and biodistribution of CaP administered by inhalation, as well as provides a new perspective in targeting the failing myocardium.

Q13. *Any information regarding the effective administered amounts of formulations (in terms of CaP, CTP, and Cy5.5) versus the amount detected per mg in each analyzed organ? Providing a % distribution would help the reader understand the biodistribution. A comparative quantification within myocardium and other organ accumulation should complement the study to better understand the selective biodistribution. A similar comparative biodistribution table quantifying the detected signals in different organs is also missing for the therapeutic study. Since the authors have established an HPLC and MS method of analysis, I wonder if these approaches could be coupled to quantify the delivery of TP compound to the heart and other organs. In fact, it is important to have data showing whether TP-10 reaches the heart and tissues other than the heart after inhalation.*

Response: Thanks for the comment. We have now provided detailed detection and comparative biodistribution table in the revised manuscript about the in vivo distribution and heart accumulation of TP-10@CaP and TP-10@CaP-CTP. To demonstrate this, we have quantified the detected TP-10 level in different organs (see Supplementary Table 1 attached below) after different treatments. The amount of TP-10 was quantified by HPLC-ESI-MS/MS. Regarding the TP-10 level, expressed as injected dose per gram of tissue (%ID/g), in the heart is calculated to be $5.74 \pm 1.17\%$ for TP-10@CaP-CTP, which was ca. 1.68-fold higher than that for TP-10@CaP ($3.41 \pm 0.68\%$ ID/g). Furthermore, the liver and kidneys showed rather high signals, in which the accumulation of CaP nanoparticles in the liver can be attributed to absorption by the mononuclear phagocyte system; those accumulated in the kidney are likely to arise from renal excretion. This corresponding data have now been included in the revised manuscript (Page 9) and supplementary materials (Page 35).

Supplementary Table 1

Summary of TP-10 level in different organs by HPLC-ESI-MS/MS from mice received TP-10@CaP or TP-10@CaP-CTP treatment through inhalation approach at 1 h. TP-10 level is expressed as injected dose per gram of tissue (%ID/g). Data are presented as mean \pm SD (n = 6). ND represents not detected.

	Heart	Liver	Spleen	Lung	Kidney
Blank	ND	ND	ND	ND	ND
TP-10@CaP	3.41 \pm 0.68	3.38 \pm 0.91	0.40 \pm 0.17	6.24 \pm 1.03	1.57 \pm 0.62
TP-10@CaP-CTP	5.74 \pm 1.17	3.07 \pm 0.74	0.46 \pm 0.13	6.46 \pm 1.15	2.27 \pm 0.51

Q14. Cell size measurements (Ref line 214) report only n=33-45 of analyzed cells, which is very low. In general, reliable studies report on hundreds of cells.

Response: According to your suggestion, we have increased the cell sizes (n = 91-102) for analysis. For your quick review, the updated Fig. 3e is attached below (please refer to its caption in the main text).

Fig. 3d, Representative images of CMs after treatment. CMs were then fixed and photographed under a microscope. Scale bar, 50 μ m. **e,** Cell surface area of CMs was determined. n = 91-102.

Q15. Suppl. Fig. 6 does not include a marker for fibrosis. Therefore, it is difficult to identify the fibrotic areas. In addition, the very same myocardial area (fiber orientation and location in the ventricle) must be provided in all panels. Ref line 234-236. In addition, the number of hearts used for the histological analysis are not indicated in the text.

Response: Thanks for raising these issues. We have included a marker for fibrosis in Supplementary Fig. 13 (Suppl. Fig. 6 in the previous version). Briefly speaking, we have utilized *Coll α 1* as the marker for identifying the fibrotic areas in the failing myocardium. The result indicated that CTP possessed the targeting capacity to the fibrotic areas of the myocardium, which further confirmed the targeting capacity of CTP in the failing heart and may provide potential research direction for the binding mechanism of CTP.

Moreover, we chose same myocardial area (ventricular septal area) where was fibrosis as the histological analysis position in all panels. Also, the number of hearts was 3 for each group, and we have added this information in the revised supplementary materials

Supplementary Fig. 13. The accumulation of Cy5.5 labeled TP-10@CaP-CTP nanoparticles in the fibrotic area of heart. **a**, Representative images of nanoparticles accumulation in the fibrotic area of heart after treated with PBS (blank control) and Cy5.5 labeled TP-10@CaP-CTP via inhalation. Scale bar, 50 μm. **b**, enlarged images of the distribution of Cy5.5 labeled TP-10@CaP-CTP in the fibrotic area of heart. **c**, The quantitative assay of Cy 5.5 fluorescence signals in the fibrotic areas. n = 3. Results are presented as mean ± SD. Statistical analysis between two groups was performed using unpaired two-tailed t-test.

Q16. *Studies performed on cardiac fibroblasts (most likely myofibroblasts) lack adequate controls for cell identity and require quantification of genes such as discoidin domain receptor 2 (DDR2) and α-smooth muscle actin, which are needed to distinguish fibroblasts from myofibroblasts. In addition, as mentioned above, proper controls with TP-10@CaP condition without CTP are missing.*

Response: Thanks for the comment. According to your valuable suggestion, we have now provided detailed evaluations in the revised manuscript about the quantification of the discoidin domain receptor 2 (DDR2) and α-smooth muscle actin (Fig. 4f). DDR2 was utilized as the identity for cardiac fibroblasts. While α-smooth muscle actin was utilized to distinguish fibroblasts from myofibroblasts. And Related to our response to Q8 of yours, we apologize for the ignore of TP-10@CaP treatment in the original version, we have now added TP-10@CaP treated group in all of the relative experiments in the studies performed on cardiac myocytes and cardiac fibroblasts (Fig. 2, 3 and 4).

Fig. 4f. Representative images of α -SMA expression in adult mice CFs after indicated treatments. Scale bar, 200 μ m.

Q17. It is curious that no monitoring was performed during the treatment study. An intermediate echocardiographic analysis is useful to understand the dynamics of the treatment. Also, no quantification of circulating biomarkers of cardiac stress are provided along this long study.

Response: Thanks for raising this important issue, which we have carefully addressed in the revised manuscript. We have performed a consecutive echocardiographic monitoring on the mice at 0, 2, 4, 6, and 8 weeks during the treatment (see the newly added Supplementary Fig. 18 below). Meanwhile, we have added more detailed quantification of circulating biomarkers of cardiac stress (plasma heart failure biomarker atrial natriuretic peptide (ANP) was chosen). We have added these data in the revised manuscript (page 12 and 13) and supplementary materials (Supplementary Fig. 18)

Supplementary Fig. 18. Consecutive echocardiographic and plasma ANP levels monitoring in mice. **a**, Representative imagines of echocardiography from each treatment group at 0, 2, 4, 6 and 8 weeks. **b-e**, the corresponding percentage of ejection fraction (EF%), percentage of fraction shortening (FS%), left ventricular end-systolic diameter (LVESD) and left ventricular end-diastolic diameter (LVEDD) after different treatments. $n = 6$. **f**, Quantification of plasma ANP levels at 0, 2, 4, 6 and 8 weeks after different treatments. $n = 8$. Results are presented as mean \pm SD. Statistical analysis was performed using one-way ANOVA with Bonferroni multiple-comparison correction.

Q18. The echocardiography table need to include additional parameters such as heart rate, LV wall thickness, and cardia output.

Response: Thanks for this nice suggestion. We have provided these parameters in the Supplementary Table 2, 3 and 4.

Q19. Can the authors provide any rationale for the use of 0.75 mg TP. Was this a reference dose used in other studies of systemic exposure via IV? Please provide the reference

Response: Thanks for raising this question. Related to our response to Q2 of yours, the

loading capacity of TP-10 was determined to be a saturation level 37.2% at the ratio of $\text{CaCl}_2/\text{Na}_2\text{HPO}_4$ of 2:3 by HPLC-MS/MS method (Supplementary Fig. 6). Considering that it is difficult to achieve 100% release of cargos from delivery systems in vivo, we have chosen the regular value (80% released) as the reference. Without considering the difference in delivery efficiency between free drugs and drug delivery systems, we chose an approximate value of 0.75 mg ($2.5 \text{ mg/kg} \times 0.367 \times 0.8 = 0.734 \text{ mg/kg}$) as the therapeutic dose for TP-10 to control the same dosage of TP-10 in every treatment group. The results and discussion are also added in the revised manuscript (Page 7) and supplementary materials (Page 2). Our currently adopted dose in vivo may actually serve as a high-bound dose. However, this needs to be verified in future work using other types of heart failure models constructed in animals.

Q20. *I would rephrase the sentence in line 284-286 by stating that the iv effect in contrasting TAC HF was less pronounced compared to the inhalation approach.*

Response: Thank you for your suggestion. We have rephrased this sentence in the revised manuscript (Page 13).

Q21. *Do the authors have any in vitro/in vivo information on the effect of TP-10@CaP-MP on complement activation? In fact, the absence of any activation would support the biocompatibility of the formulation if a systemic exposure is achieved after inhalation. Also, this provide a relevant information as the current doses used in this study (2.5 and 7.5 mg/kg) appear to be higher than previously used by other groups (0.5 mg/kg, Miragoli et al, Science Translational Medicine 2018, Modica et al, Circulation 2021)*

Response: Thanks for raising this issue. We have now performed more experiments in vitro and in vivo to evaluate the effect of TP-10@CaP-CTP on complement activation using ELISA assay (Supplementary Fig. 21). Plasma levels of C3 and its activation fragment C3b, C3c were not significantly elevated in the TP-10@CaP-CTP treated groups. These data indicated that TP-10@CaP-CTP nanoparticles have no effect on complement activation in the experimental doses. This may be due to the CaP is the main inorganic component of biological hard tissues and its good biocompatibility. Thus, the designed CaP nanoparticles in this study will not lead to inflammation response, as well as the complement activation. The results and discussion are also added in the revised manuscript (Page 15)

Supplementary Fig. 21. In vitro (a-c) and in vivo (d-e) effect of TP-10@CaP-CTP on complement activation. a-c, for in vitro experiment, 0.5 mL Plasma from mice was incubated for 30 min with various amounts (0, 5, 10, 20, 40 µg) of TP-10@CaP-CTP and complement activation was monitored by ELISA assay for C3, C3b and C3c levels. d-e, for in vivo experiment, various amounts (0, 2.5, 5, 7.5, 10 mg/kg) of TP-10@CaP-CTP nanoparticles were administrated via inhalation. Plasma was collected after 1 h and subjected to ELISA assay. n = 5. Results are presented as mean ± SD. Statistical analysis was performed using one-way ANOVA with Bonferroni multiple-comparison correction.

My minors' concerns are:

q1. Please, provide more details about the type of lung complications to be expected and in reference to the kind of inhalable drugs to be used.

Response: Thanks for raising this suggestion. We have provided some type of lung complications in Introduction (Page 4, as below). We refer to the studies on inhalable insulin for diabetes and inhalable levodopa for Parkinson's disease. Long-term clinical studies found that inhalation treatment will lead to wheezing and bronchoconstriction, and result in a gradually greater losses in lung function. Even in some studies, lung cancer cases were reported.

Indeed, the lung complications of our current strategy has not been understood clearly. Furthermore, the inhalable drugs may encounter side effects of the lungs even though the TP-10@CaP-CTP used in our experiments indicated safety. Moreover, further toxicity studies using a much higher dose (e.g., tenfold of the therapeutic dose) is needed before clinical translation. The therapeutic effect in the current platform has not reached a substantial advancement over current clinical practice, and thus, future investigations should focus on the systematic majorization of nanomedicine-based

inhalation therapy, including various parameters involving lung complications, drug loading, and the efficiency of the targeted administration to the heart. This discussion has now been included in the revised manuscript (Page 19).

Page 4: *“In previous studies, long-term utilization of inhalable insulin will lead to wheezing and bronchoconstriction, and result in a gradually greater losses in lung function. In long-term clinical trials, some lung cancer cases were reported in patients received inhalable insulin. Besides, inhalable levodopa was utilized to the treatment of Parkinson’s disease, but also be limited to use clinically due to its increased risk of bronchospasm. How to avoid lung complications is crucial for fulfilling the requirement of efficient inhalation delivery.”*

q2. *Ref. to line 148. A reference to the previous protocol of synthesis must be clearly stated while more emphasis to the actual novelties (i.e. loaded drug and surface functionalization with the targeting peptide) shall be provided.*

Response: We have added more detailed descriptions of the synthesis process of TP-10@CaP-CTP (Page 6-7) together with the reference (ref. 9, 12 and 24) in the revised manuscript (Page 5) as follows.

“In brief, TP-10@CaP nanoparticles were prepared a biomineralization-inspired strategy by mixing TP-10 with CaCl₂ and Na₂HPO₄ according to the previous studies by Miragoli et al^{9, 12, 24}. The transmission electron microscopy (TEM) image showed that TP-10@CaP was homogeneous and spherical with a diameter of about 73 nm (Supplementary Fig. 1). Then, TP-10@CaP nanoparticles were modified with 1,2-distearoyl-sn-glycero-3-phosphoethanolamine conjugated polyethylene glycol acid (DSPE-PEG-COOH) to obtain the carboxyl-functionalized CaP (TP-10@CaP-DSPE-PEG). Unlike TP-10@CaP, TP-10@CaP-DSPE-PEG showed a smooth organic surface under observation by TEM. After that, TP-10@CaP-DSPE-PEG was reacted with the amino group of CTP peptide (CSTSMKAC, Supplementary Fig. 2) by the EDC/NHS coupling reaction to form the final product (TP-10@CaP-CTP).”

q3. *Line 178 states "The accumulation of the designed nanoparticles in the heart was obviously enhanced by inhalation administration". An explanation of why this is "obvious" needs to be provided.*

Response: We apologize for any confusion caused by the inappropriate description here. During the revision, we become aware that this state in the previous version of our manuscript might be unsuitable for summarized the following findings in this part. We have deleted this sentence in the revised manuscript.

q4. *Write if same doses were used in the injected and inhaled animals.*

Response: Thanks for the comment. Actually, the same doses were used in the injected and inhaled animals. we have checked the dose used throughout the manuscript and added this information in our revised manuscript (page 30).

q5. *Do you have any data providing the evidence whether pulmonary translocation*

occurs via intra- or inter-cellular translocation? Reference to line 190

Response: Thanks for this important question. To demonstrate this, we have performed an additional immunofluorescent staining on the lung after inhalation of Cy5.5 labeled TP-10@CaP-CTP nanoparticles. We found that the fluorescence signal of Cy5.5 both existed in the CD31 marked alveolar epithelial cells (green arrows) and the joint between cells (red arrows). These results indicated that pulmonary translocation might realize via both intra- and inter-cellular translocation. We have added a newly immunofluorescent staining results for clearer demonstration of the hybrid magnetic field (Supplementary Fig. 24) in the revised version.

Supplementary Fig. 24. Pulmonary translocation of Cy5.5 labeled TP-10@CaP-CTP. The fluorescence signal of Cy5.5 both existed in the CD31 marked alveolar epithelial cells (green arrows) and the joint between cells (red arrows).

q6. Please, provide details/reference for the adopted surgery protocol for TAC generation.

Response: Thanks for raising this issue. We have now added more detailed descriptions of the TAC-induced pressure overload together with the reference (ref. 23, 34, 35 in the revised manuscript) for clearer demonstration of the adopted surgery protocol in the revised manuscript (Page 22). For your quick review, the updated protocol is attached below.

“A heart failure mouse model was established by TAC-induced pressure overload according to the previous studies^{23, 34, 35}. Briefly, eight-week-old male C57BL/6J mice were anesthetized by continuous administration of inhaled isoflurane (2%) while the surgery was performed. Mice were subjected to endotracheal intubation by a 22-gauge plastic catheter, which then connected to a ventilator. After that, the left chest was opened and blunt dissection was performed at the proximal portion of the sternum to access the thoracic aorta. A 27-gauge needle was placed on the transverse aorta between the innominate and left common carotid. The 6-0 silk sutures were used to ligate the transverse aorta with the needle. After completely occluding the aorta, the

needle was immediately removed, and the thoracic cavity was closed. Sham mice underwent the same procedure without ligation. Aortic velocity peak pressure was determined by in vivo echocardiography and mice with gradients greater than 30 mmHg were used.”

q7. *Specify if the analysed ADCMs were from left ventricle or whole heart.*

Response: Thanks very much for your kind reminding. We have now specified the analysed ADCMs were from left ventricle in this work, and added relative description to the revised manuscript (page 31 and 36).

q8. *The effective volume of nebulised formulation needs to be included in the methods.*

Response: Thank you for noting the missing information. We have now specified the effective volume of nebulised formulation in the revised manuscript (Page 30).

q9. *Line 431-433 “The potential mechanism by which this CTP binds to the diseased myocardium might mimic the binding of Titin to crystalline α -B, which is highly expressed in the ischemic myocardium”. The meaning of this statement and the link of CTP to Titin is not clear. Please clarify and rephrase the sentence.*

Response: We apologize for the confusion caused due to the insufficient explanation of the targeting mechanism of CTP to the diseased myocardium. We have rephrased the sentence (Page 17).

“A cytoskeletal protein Titin in heart was identified with partial sequence similarity to the CTP (CSTSMLKAC). Kanki, S. et al raised the hypothesis that CTP might preferentially bind to alpha-B crystalline, which is highly combined with cardiac Titin under ischemic state.”

q10. *Some typos are present.*

Response: Thank you for your careful review. We have overhauled the entire manuscript to address this issue spotted by the reviewer. The text has been thoroughly revised for conciseness and clarity, the figures (incorporating new data) have been enhanced with clear information for better visibility, and grammar/spelling mistakes are carefully corrected.

References:

1. Kanki, S. et al. Identification of targeting peptides for ischemic myocardium by in vivo phage display. *J Mol Cell Cardiol* **50**, 841-848 (2011).
2. Wang, X. et al. Engineered Exosomes With Ischemic Myocardium-Targeting Peptide for Targeted Therapy in Myocardial Infarction. *J Am Heart Assoc* **7**, e008737 (2018).
3. Miragoli, M.A.-O. et al. Inhalation of peptide-loaded nanoparticles improves heart failure. *Sci Transl Med* **10**, eaan6205. (2018).
4. Modica, J. et al. Nano-miR-133a Replacement Therapy Blunts Pressure Overload-Induced Heart Failure. *Circulation* **144**, 1973-1976 (2021).

5. White S. et al. EXUBERA: pharmaceutical development of a novel product for pulmonary delivery of insulin. *Diabetes Technol Ther* **7**, 896-906 (2005).
6. Balducci, A.G. et al. Pure insulin highly respirable powders for inhalation. *Eur J Pharm Sci* **51**, 110-117 (2014).
7. Goldberg T Fau - Wong, E. & Wong, E. Afrezza (Insulin Human) Inhalation Powder: A New Inhaled Insulin for the Management Of Type-1 or Type-2 Diabetes Mellitus. *P&T* **40**, 735-741 (2015).
8. Heinemann, L. & Parkin, C.G. Rethinking the Viability and Utility of Inhaled Insulin in Clinical Practice. *J Diabetes Res* **2018**, 4568903 (2018).
9. Martinez-Raga, J. et al. 1st International Experts' Meeting on Agitation: Conclusions Regarding the Current and Ideal Management Paradigm of Agitation. *Front Psychiatry* **9**, 54 (2018).
10. Anderson, S. et al. Inhaled Medicines: Past, Present, and Future. *Pharmacol Rev* **74**, 48-118 (2022).
11. Ye, Y., Ma, Y. & Zhu, J. The future of dry powder inhaled therapy: Promising or discouraging for systemic disorders? *Int J Pharm* **614**, 121457 (2022).
12. Masouleh, M.P., Hosseini, V., Pourhaghgouy, M. & Bakht, M.K. Calcium Phosphate Nanoparticles Cytocompatibility Versus Cytotoxicity: A Serendipitous Paradox. *Curr Pharm Des* **23**, 2930-2951 (2017).

REVIEWER COMMENTS

Reviewer #1 (Remarks to the Author):

I appreciate the effort of the authors to reply to all my comments and to provide new characterizations on the materials.

However, I recommend more effort to improve the clarity of the paper since the data about the chemical characterizations of the materials were only simply reported as new figures in the SI without a clear description of chemical significance. For example, the appearance of several new bands (not only those of Amide I and II) in the FTIR spectrum of TP-10@CaP-DSPE-PEG is not reported and is not well discussed.

HPLC data of Fig.S2 was not discussed in the text.

The authors simply claimed in the text that the new characterizations give an indication of functionalization with TP-10 and CTP, but they do not report why and how this happens and the related chemical meanings.

The authors have indeed included some tentative chemical explanations in the reply to authors, but these were not reported in the text (see for examples replies to questions 9 and 11), so I suggest to include these findings also in the text and they should be clearly supported by analytical evidences. It is important that they are more analytical to evaluate the evidence that they collected about the chemistry of materials.

The XRD patterns reported in Figure 3 of SI are full of noise and it is very difficult to analyze. I suggest to repeat the test to collect more reliable spectra.

Zeta potential values are similarly only reported but the chemical meaning of the changes was not reported.

Please state why the size of the samples in the different media is different.

English grammar and style should be extensively revised. Some typos are also present.

Reviewer #2 (Remarks to the Author):

The authors have tried to address my concern regarding the novelty of this study. I still have some concerns regarding the experimental methods and data interpretation.

(1) Figure 2, the TP-10@CaP-CTP nanoparticles appear highly concentrated in the atrium of the heart. Cyclic nucleotide changes have been implicated in atrial vibration. Is PDE10A also expressed in atrial CMs?

(2) Fig. 2a and Supplementary Fig. 15, cAMP and cGMP levels are currently shown by fold changes. Their levels should be shown with real concentration units.

(3) cAMP/cGMP levels were measured after more than 24 hours of treatment by nanoparticles with TP-10 in CMs and CFs, which is under the same time course that authors show the functional consequences of CM hypertrophy and CF activation. The changes of cAMP/cGMP often occur very fast at relatively early

time points at the signaling pathway. The reasoning for accessing cAMP/cGMP at such a late time point is unclear. There is a possibility that cAMP/cGMP level changes may not simply reflect the consequence of PDE10A inhibition.

(4) Based on the results in Fig. 2a and Supplementary Fig. 15, PDE10A represents the major PDE isozyme in CMs and CFs. This is because the TP-10@CaP-CTP abolished cAMP/cGMP reduction by PE or TGF β up to 80-90%. However, several other PDEs were also reported in CMs or CFs. How do authors explain the observations?

(5) The culture condition for the hypertrophy study using adult mouse CMs is unclear. Are these CMs retaining the capability of contraction? Adult mouse CMs have difficulty being in culture for 24 hours if retaining contraction.

(6) In Figs. 3b and c, the authors analyzed the activation of AMPK by TP-10. The rationale for choosing AMPK should be mentioned. The role of AMPK in cardiac hypertrophy is controversial. Some studies showed that AMPK is activated during cardiac hypertrophy.

(7) Fig. 4, the authors chose KT5823 as a PKG inhibitor in CFs. However, a previous study reported that KT5823 worked in a cell-free system but not intact cells (JBC Vol. 275, pp. 33536-33541, 2000).

(8) Fig. 5 and 6, the authors included 0.75 mg/kg TP-10 to compare with other TP-10 nanoparticles, such as 2.5 mg/kg TP-10@CaP-CTP, which is great. The authors predicted that free TP-10 (0.75 mg/kg) is equivalent to TP-10@CaP-CTP (2.5 mg/kg) by simple calculation considering a saturation level of 37.2% at the ratio of CaCl₂/Na₂HPO₄ of 2:3 and 80% release of cargos. However, TP-10 is highly protein-bound in the culture medium or in vivo (J Med Chem. 2009; 52:5188–5196) (J Pharmacol Exp Ther. 2014; 349:138–154). Thus, TP-10 has often been used at much higher concentrations than its IC₅₀ (e.g., \approx 3 mg/kg in mice) in previous studies (J Pharmacol Exp Ther. 2008; 325:681–690)(Circulation. 2020; 141:217–233.). Therefore, the effect of free TP-10 in this study is very likely underestimated due to a low concentration.

Reviewer #3 (Remarks to the Author):

The authors have generated a large amount of new data and addressed most of my initial concerns. In this new format, the study provides an interesting documentation for the use of inhalation delivery in the long-term management of chronic heart failure.

However, I was still puzzled by a few points, as described below.

In the abstract, line 10 and elsewhere, the authors refer to a low dosage of the treatment, but it is not clear whether they are referring to the actual active ingredient, the nanoparticle, or something else. Please clarify this, and if possible, it would be interesting to refer to a relevant published study for comparison. For example, mentioning the TP dose used in the study from reference 23 (lines 116-120)

would be of great help. This would also support other statements such as those in lines 107-111.

Line 95. With reference to the statement “However, the lung complications which greatly limit the wide application of inhalation route clinically, are still need to be explored”, the author might consider to discuss a new recent publication (Alogna et al., JACC 2024 Lung-to-Heart Nano-in-Micro Peptide Promotes Cardiac Recovery in a Pig Model of Chronic Heart Failure) where the very same CaPs (engineered in a dry powder of microparticles containing drug-loaded CaPs) were tested in a (1 week long) dose range finding study in rats and no adverse effects at any of the increasing doses were reported.

Line 107-11. I believe the sentence/message needs to be rephrased to better convey the intended message. In fact, I expect the potential pulmonary complication to be similar whether one uses the same amount of CaP or the CaP-CTP nanoformulation. In fact, I see no reason why CTP would reduce the potential pulmonary complication that CaP might have, unless the authors believe that coating CaP with CTP is expected to provide some "tox-reduction" effect in the lung. In my opinion, the only added value of CTP is expected systemically by influencing the biodistribution towards an enriched accumulation in the heart, thus reducing the required therapeutic dose to be administered.

Line 129-132. Please rephrase. The molecular mechanism of action of TP-10 has already been explored/elucidated by other studies and here I would recommend using different wording, as the same authors write later in the discussion (line 532-535).

Fig. 1b and Supplementary Fig. 1, please mention when post-synthesis TEM analysis was performed.

Fig2a,d,e,f,g,h. Please correct the positioning of the panels (columns) by adopting the same order when the group is presented in sham and TAC (i.e. blank, TP-10@CaP-CTP, TP-10@CaP).

Fig 2d, labeling/description is missing for all panels. Are different time points per group shown, or slices of heart or time points? Please clarify.

Fig. 2b: Please indicate in the figure or legend which graph is for the heart or lung.

Most importantly, it would be relevant to show from the fluorescence study a comparative percentage of dye accumulation from different organs. Such data would then facilitate the comparison with those obtained from the TP analyses in Supplementary Table 1, providing a nice biodistribution study based on either nanoparticle (Cy5.5) or drug (TP-10) tracking.

Line 254-258, The TP alone control is missing in TP quantification from the biodistribution study.

Supplementary Fig. 12c,d. It is not clear to me why the data of TP-10@CaP in -PE conditions are not shown. Please clarify or show. In fact, it would be nice to see the two CaP preparations (i.e. +/- CTP) in +/-PE in the same densitometry plot.

Supplementary Fig. 13. Please include the same results also for TP-10@CaP, which should be presented as a comparative control in each assay performed. In fact, in the current study, the authors do

implement the CaP data from the literature with the CTP targeting peptide.

Supplementary Fig. 14, 23. Please, include the TP-10@CaP control in PBS condition.

Supplementary Fig. 18. Please add symbols for statistical analysis in graph and relative description in legend.

Supplementary Table 1. Data from inhaled free TP-10 were not included, but represent an important control. In particular, because free TP-10 was used in the therapeutic experiment.

In Supplementary Figures 22 and 23, the data are expressed as %Cy5.5 per field, whereas in Supplementary Figures 12 and 14, the data are expressed as %Cy5.5 per cell. It would be better to use the same method of data presentation to allow some sort of comparative analysis between experiments.

Line 474-477. Still do not understand the rationale for including this information. CTP was used in this paper to enrich the delivery of the nanoparticle to cardiomyocytes (and cardiac fibroblasts) by a possible binding to an unidentified membrane receptor that is enriched on such cells. Even if CTP can bind to alpha-b-crystallin due to homology to cardiac titin, what is the relevance for extracellular targeting since both alpha-b-crystallin and titin are cytosolic? Sentences 518-524 I think are better for the topic.

Supplementary Fig. 1: Why a different scale bar is used in b (100 nm) compared to a, c (200 nm)

Please add references to the studies mentioned in lines 104-107 that refer to “drug delivery carriers that were modified by CTP”

Supplementary Fig. 18 should be renumbered as Supplementary Fig. 16 as presented earlier than the actual Supplementary Fig. 16-17

Ref24 is not correct to be included in line 628 as no TAC is performed on this paper.

Line 408-413- Do the authors have any information from the literature on the toxicity data of TP-10 or CTP? Do the authors propose that the observed effect is a phenomenon due to one of the components or the entire formulation? Although performed in TAC mice, did the authors observe any incremental lung issue with the 7.5 mg/kg/2 days dose regimen of TP-10@CaP as used in the tox assay of Supplementary Figure 19 where TP-10@CaP-CTP has been used?

Finally, I suggest a thorough revision of English grammar by a native speaker.

Point-by-point responses to reviewers' comments

Response to Reviewer #1 (Remarks to the Author)

Q1. I appreciate the effort of the authors to reply to all my comments and to provide new characterizations on the materials.

However, I recommend more effort to improve the clarity of the paper since the data about the chemical characterizations of the materials were only simply reported as new figures in the SI without a clear description of chemical significance. For example, the appearance of several new bands (not only those of Amide I and II) in the FTIR spectrum of TP-10@CaP-DSPE-PEG is not reported and is not well discussed.

Response: We would like to thank the reviewer for comprehensive feedback on improving the manuscript. We have thoroughly revised the manuscript for enhanced clarity and added extra experimental data for scientific rigor.

Thank you very much for this value suggestion, which is highly appreciated. According to your suggestion, we have now enhanced the description of chemical significance of every characterization result in the revised manuscript for better visibility. Meanwhile, for enhanced clarity of the chemical characterizations, we further integrated the relevant data into **Figure 2** of the main text and revised the relevant part in the revised manuscript (**Page 5-6**).

Fig. 2. Characterizations of different nanoparticles prepared in each step. a-c, TEM images of different intermediate product, including **a**, TP-10@CaP, **b**, TP-10@CaP-DSPE-PEG and **c**, TP-10@CaP-CTP collected at 12 h after preparation. **d**, Elemental mappings of Ca, P, C, and N elements of TP-10@CaP-CTP. **e**, XPS and **f**, FT-IR spectra of different nanoparticles. **g**, Hydrodynamic diameter of different prepared nanoparticles in water (0.4 mg/mL). **h**, Zeta potentials of different prepared nanoparticles in PBS at pH 7.4. n = 3. **i**, TP-10 loading and encapsulation efficiency analysis with different ratio of CaCl₂/Na₂HPO₄. n=3.

Q2. HPLC data of Fig.S2 was not discussed in the text.

Response: Thank you for your comment. This is the identification report of the synthesized CTP peptide. We have added the discussion of HPLC data (**now Supplementary Fig. 1**) in the revised manuscript (**Page 5**).

Q3. The authors simply claimed in the text that the new characterizations give an indication of functionalization with TP-10 and CTP, but they do not report why and how this happens and the related chemical meanings.

Response: Thank you for the comment. We apologize for the oversimplified description of these characterizations that caused confusion. We have also added more detailed descriptions of chemical significance of every characterization (data presented in **Fig. 2, Supplementary Fig. 1-4**) in the revised manuscript to avoid confusion (**Page 5-6**).

Q4. The authors have indeed included some tentative chemical explanations in the reply to authors, but these were not reported in the text (see for examples replies to questions 9 and 11), so I suggest to include these findings also in the text and they should be clearly supported by analytical evidences. It is important that they are more analytical to evaluate the evidence that they collected about the chemistry of materials.

Response: Thanks for your kind reminding. Related to our response to Q1 and Q3 of yours, we have provided a more accurate description in the revised manuscript. Also, according to your suggestion, we have included the findings about the chemical explanations together with the corresponding data support for clearer demonstration of the preparation process (**Page 5-6**).

Q5. The XRD patterns reported in Figure 3 of SI are full of noise and it is very difficult to analyze. I suggest to repeat the test to collect more reliable spectra.

Response: Thanks. We have re-examined the XRD measurement of different nanoparticles. The noise and XRD patterns have been improved to a certain extent. It should be noted that the CaP-based nanoparticles showed no obvious diffraction peaks only a broad peak occurred at around $2\theta = 30^\circ$, indicating an amorphous phase

of this product. Thus, it is difficult to obtain a XRD spectra without any noise. For your quick review, the updated **Supplementary Fig. 3**. is attached below.

Supplementary Fig. 3. Powder X-ray diffraction patterns of different nanoparticles prepared in each step.

Q6. Zeta potential values are similarly only reported but the chemical meaning of the changes was not reported.

Response: We have enhanced the Discussion sections about the zeta potential values changes in the revised manuscript for better visibility, and the chemical meaning of the changes is carefully indicated (**page 6, Line 168-173, see below**).

“Additionally, the TP-10@CaP and bare CaP nanoparticles showed similar surface charges of approximately -20 mV due to the negligible impact of drug encapsulation on the surface charge. However, modification of the CaP nanoparticles with DSPE-PEG-COOH changed the zeta potential to -34.8 ± 1.9 mV due to the negative charge of the carboxyl groups, confirming the formation of the lipid layer. In contrast, due to the presence of primary amino groups on CTP, the zeta potential of TP-10@CaP-CTP decreased to -12.6 ± 1.2 mV after the formation of amide bonds (Fig. 2h).”

Q7. Please state why the size of the samples in the different media is different.

Response: Thanks for your comment. The hydrodynamic size of TP-10@CaP-CTP was enhanced in DMEM, which contains a mixture of biomolecules such as amino acids, sugars, vitamins, and salts. In this case, the interaction of nanoparticles with biomolecules of medium will inevitably increase the hydrodynamic size (nm). This phenomenon occurs in many calcium-based biomineral (Chem 2020, 6, 1391–1407; J. Am. Chem. Soc. 2018, 140, 2165–2178). Despite all this, the TP-10@CaP-CTP nanoparticles showed a stable mean hydrodynamic diameters with less than 200 nm, which are highly useful for biological assays.

Q8. English grammar and style should be extensively revised. Some typos are also present.

Response: Thanks for your kind comment. The text has been thoroughly revised for conciseness and clarity, and grammar/spelling mistakes are carefully corrected. Also, we have repolished the language in the revised version by the Nature Editing Services (**Fig. R1**).

SPRINGER NATURE | Author Services

March 27, 2024

Dear Haobo Weng,

Thank you for choosing Springer Nature Author Services. This manuscript, titled "An inhaled cardiac-targeting peptide-modified nanomedicine prevents pressure overload-related heart failure based on cardiopulmonary circulation," is very interesting. The paper was edited for grammar, phrasing, and punctuation. In addition, many edits were made to further improve the flow and readability of the text. Below, we highlight the areas of this paper that we focused on in our edit.

In cases where the meaning of the text was not clear, revisions were made to convey the information with increased clarity and reduced ambiguity.

Some edits were made to improve conciseness by trimming unnecessary words and streamlining the flow of your manuscript.

Articles are an important aspect of the English language, including the definite article "the" and the indefinite articles "a" and "an." Our edits focused on improving article use, which is often strongly dependent on context and field conventions.

Comments were left if further clarification would be helpful or confirmation of the meaning of the text was necessary. Please review these comments and all our changes carefully for more detailed suggestions, as well as to ensure that the final version of the manuscript is fully accurate.

Thank you again for using our editing services; we wish you the best of luck with your submission.

Best regards,

Beth K.
Senior Editor
Springer Nature Author Services

Figure R1. English editing certificate of our paper.

Response to Reviewer #2 (Remarks to the Author)

The authors have tried to address my concern regarding the novelty of this study. I still have some concerns regarding the experimental methods and data interpretation.

Response: We thank the reviewer for the careful review and constructive feedback. We have enhanced the manuscript's clarity where confusion arose and performed additional experiments to elaborate the main findings of our work. Please find our detailed responses to your specific comments below.

Q1. Figure 2, the TP-10@CaP-CTP nanoparticles appear highly concentrated in the atrium of the heart. Cyclic nucleotide changes have been implicated in atrial vibration. Is PDE10A also expressed in atrial CMs?

Response: Thank you for your question. The fluorescent imaging signal represents the relative amount of the nanoparticles in the heart tissues, and the high fluorescent imaging signal in atrium probably related to the lower density of atrial tissue than ventricular tissue. Thus, the result of this ex-vivo fluorescence signal might only be used to illustrate the degree of accumulation of nanoparticles in heart. And whether PDE10A or our designed TP-10@CaP-CTP nanoparticles are implicated in atrial fibrillation remains unclear, it might need to be verified in future work by performing other relative experiments and assays. In our present study, we may only focus on the design and application of heart-targeted peptide CTP engineering strategy to promote the drug accumulation on pressure overload induced heart failure treatment.

Q2. Fig. 2a and Supplementary Fig. 15, cAMP and cGMP levels are currently shown by fold changes. Their levels should be shown with real concentration units.

Response: We thanks for your valuable suggestion. Related to Q3, we have now re-tested the cAMP/cGMP levels in the CMs and CFs. And the data have been shown in the manner of real concentration units. These results were shown in the revised manuscript (Page 31, Fig. 4c) and revised supplementary materials (Page 18, Supplementary Fig. 15).

Fig. 4c, CMs intracellular cAMP and cGMP levels were determined by ELISA. n=16.

Supplementary Fig. 15. CFs intracellular cAMP and cGMP levels.

Q3. cAMP/cGMP levels were measured after more than 24 hours of treatment by nanoparticles with TP-10 in CMs and CFs, which is under the same time course that authors show the functional consequences of CM hypertrophy and CF activation. The changes of cAMP/cGMP often occur very fast at relatively early time points at the signaling pathway. The reasoning for accessing cAMP/cGMP at such a late time point is unclear. There is a possibility that cAMP/cGMP level changes may not simply reflect the consequence of PDE10A inhibition.

Response: We thank the reviewer for pointing out this issue. We apologize for the confusion caused due to unclear description of the methodological in the previous version of our manuscript. Actually, the CMs and CFs were stimulated with PE or TGF-β for 24 h. After different treatments for 1 h, the CMs or CFs were harvested for following detection. We choose 1 h as the time point for detection is due to the reason that it takes time for TP-10 released from the CaP carriers. And the results of pre-experiment indicated that intracellular cAMP and cGMP levels in PE-treated CMs were reached to peak at 1 h after the treatment of TP-10@CaP-CTP (as shown below). To ensure the accuracy of the results, we have added more detailed descriptions of the experimental procedure (Page 4, Line 95-106 in supplementary materials) together with a newly performed measurement of cAMP/cGMP levels for clearer demonstration of this experiments. These results have now been included in the revised manuscript (Page 31, Fig. 4c) and revised supplementary materials (Page 18, Supplementary Fig. 15).

Intracellular cAMP and cGMP levels in PE-treated CMs at different time points after treatment with TP-10@CaP-CTP.

Q4. Based on the results in Fig. 2a and Supplementary Fig. 15, PDE10A represents the major PDE isozyme in CMs and CFs. This is because the TP-10@CaP-CTP abolished cAMP/cGMP reduction by PE or TGF β up to 80-90%. However, several other PDEs were also reported in CMs or CFs. How do authors explain the observations?

Response: Thanks for mentioned this valuable issue. This phenomenon may be due to more released TP-10 inside the AMCM in the TP-10@CaP-CTP group by the targeted uptake. By contrast, the uptake of TP-10@CaP and free TP-10 by CMs is similar, and showed less effect on abolishing cAMP/cGMP reduction induced by PE or TGF- β . Meanwhile, we found that this phenomenon was presented in other previous studies. For example, in Knight et al's work (Proc Natl Acad Sci U S A. 2016 Nov 8;113(45): E7116-E7125.), they treated CMs with Ang II, leading to the reduction of intracellular cAMP levels. However, a PDE1 inhibitor (IC86340) completely reversed cAMP levels in CMs under the treatment of Ang II. Thus, there may be other potential mechanism for explaining this phenomenon, while more studies are still required in the future. We agree with the viewpoint of the reviewer that other PDEs presented in CMs or CFs may also affect the cAMP/cGMP level. Future work will pay more attention the detail mechanism of TP-10@CaP-CTP on CMs or CFs.

Q5. The culture condition for the hypertrophy study using adult mouse CMs is unclear. Are these CMs retaining the capability of contraction? Adult mouse CMs have difficulty being in culture for 24 hours if retaining contraction.

Response: Thanks for the comment. We apologize for the confusion caused due to unclear description of the culture condition for the hypertrophy study in the previous version of our manuscript. In fact, the isolation and culture protocols were according to the previous study by Ackers-Johnson, M. et al (Circ Res 119, 909-920 (2016)). For most experiments related to CMs, the culture medium contained 2,3-butanedione monoxime (BDM) for inhibition of CMs contracted (The detailed ingredients of the culture medium for adult mouse CMs were mentioned in Manuscript (Page 20, Line 560-561)). Expect for the experiments of CMs shortening/relengthening assay, the CMs were resuspend in the contractile buffer without BDM (The detailed ingredients

of the culture medium for CMs shortening/relengthening assay were mentioned in **Supplementary materials Line 139-140**.

Q6. *In Figs. 3b and c, the authors analyzed the activation of AMPK by TP-10. The rationale for choosing AMPK should be mentioned. The role of AMPK in cardiac hypertrophy is controversial. Some studies showed that AMPK is activated during cardiac hypertrophy.*

Response: Thanks for the comment. We have added the rationale for choosing AMPK in the revised manuscript (**Page 10, Line 274-278, see below**). As we know, AMPK is a natural energy sensor in mammalian cells that plays an essential role in regulating energy homeostasis. Previous studies found that both intrinsic and extrinsic activation AMPK could protect against cardiac hypertrophy and heart failure (Circ Res. 2012 Aug 31;111(6):800-14.). And numerous studies have demonstrated that AMPK is a critical effector for the prevention of pressure overload induced cardiac hypertrophic (Circulation. 2017 Nov 21;136(21):2051-2067.) (J Cell Mol Med. 2022 Feb;26(3):855-867.). Thus, we should be able to assume that the activation of AMPK can improve the pathological process of cardiac hypertrophic and heart failure.

“Numerous previous studies have demonstrated the effect of the cAMP/AMPK axis on various cell types after AMPK activation³¹. AMPK is a natural energy sensor in mammalian cells that plays an essential role in energy homeostasis, can protect against cardiac hypertrophy³²⁻³⁴ and has numerous beneficial effects on heart failure^{31, 32, 35, 36}. Moreover, cAMP can lead to AMPK activation in a protein kinase A (PKA)- or exchange protein activated by cAMP (EPAC)-dependent manner³¹. This evidence prompted us to explore whether the therapeutic effect of TP-10@CaP-CTP was dependent on cAMP-induced AMPK activation. First, we determined the phosphorylation level of AMPK in AMCMs by WB. AMPK phosphorylation was significantly reduced in PE-stimulated AMCMs, but this effect was reversed by treatment with TP-10@CaP-CTP (Fig. 4i, j). Then, we used the AMPK inhibitor compound C (CC) and found that the therapeutic effect of TP-10@CaP-CTP on cardiac hypertrophy was blocked by CC (Fig. 4k, l), as indicated by the CM cell surface area. Moreover, CC abolished the effects of TP-10@CaP-CTP on the downregulation of hypertrophic marker genes (Fig. 4m-o).”

In addition, we also added the rationale for choosing cGMP in CFs in the revised manuscript (**Page 11, Line 313-315**):

“According to the above results, TP-10 reversed the reduction in cAMP and cGMP levels in CFs stimulated by TGF- β (Supplementary Fig. 15). As an effector of cGMP, activated PKG exerts antifibrotic effects on heart failure, thus inhibiting cardiac fibrosis and improving cardiac function^{37, 38}. This finding prompted us to explore whether the inhibitory effect of TP-10@CaP-CTP nanoparticles on TGF- β -stimulated CFs was cGMP/PKG dependent.”

Q7. *Fig. 4, the authors chose KT5823 as a PKG inhibitor in CFs. However, a previous study reported that KT5823 worked in a cell-free system but not intact cells*

(JBC Vol. 275, pp. 33536-33541, 2000).

Response: We thank the reviewer for this professional discussion and the recommended study. We followed your suggestion and read this article carefully (JBC Vol. 275, pp. 33536-33541, 2000). It is true that T5823 was failure to inhibit the PKG-mediated response in the intact cells, which raised a reconsideration on the use of KT5823 as a PKG inhibitor in intact cells. However, they only verified this phenomenon in two cell types (human platelets and rat mesangial cells). The actual effect of KT5823 in CFs was still unclear. Indeed, in previous biomedical applications, KT5823 has been successfully applied as a PKG inhibitors in the in vivo and vitro study in a study regarding ventricular arrhythmias (Circ Res. 2021 Sep 3;129(6):650-665.). Meanwhile, it has been used in a study regarding heart ischemia/reperfusion injury (Molecules. 2020 Jul 28;25(15):3426.). These application of KT5823 may be due to the fact that it can effectively inhibit PKG in the heart. To avoid confusion and ensure the accuracy of our result, we would like to use another PKG inhibitor (DT-2) to replace KT5823 in our study. The detailed descriptions of data together with a newly performed evaluation were added to the revised manuscript (Page 33, Fig. 5 and Page 11, Line 316-318).

Fig. 5. The effect of TP-10@CaP-CTP on cardiac fibroblasts.

a, Representative images of CF wound closure after the indicated treatments for 24 h. Scale bar, 500 μm . **b**, Representative images of migrated CFs after the indicated treatments for 24 h. Scale bar, 200 μm . **c**, Statistical analysis of CF wound closure. $n = 5$. **d**, Statistical analysis of migrated CFs. $N = 5$. **e**, CCK8 assay to evaluate the cell proliferation capacity of adult mouse CFs. $n = 20$. **f**, Representative images of α -SMA expression in adult mouse CFs after the indicated treatments. Scale bar, 200 μm . **g**, Quantitative assessment of fluorescence intensity. $n = 3$. **h**, Western blotting analysis of α -SMA and $\text{Coll1}\alpha 1$ in adult mouse CFs after the indicated treatments. GAPDH was used as a loading control. **i**, Quantitative assessment of α -SMA expression. $n = 6$. **j**, Quantitative assessment of $\text{Coll1}\alpha 1$ expression. $n = 6$. **k**, mRNA levels of the indicated genes in adult mouse CFs after the indicated treatments. $n=6$. The results are presented as the mean \pm SD. For **c**, **d**, **e**, **g**, **i**, **j**, and **k**, statistical analysis was performed using one-way ANOVA with the Bonferroni multiple comparison correction.

Q8. *Fig. 5 and 6, the authors included 0.75 mg/kg TP-10 to compare with other TP-10 nanoparticles, such as 2.5 mg/kg TP-10@CaP-CTP, which is great. The authors predicted that free TP-10 (0.75 mg/kg) is equivalent to TP-10@CaP-CTP (2.5 mg/kg) by simple calculation considering a saturation level of 37.2% at the ratio of $\text{CaCl}_2/\text{Na}_2\text{HPO}_4$ of 2:3 and 80% release of cargos. However, TP-10 is highly protein-bound in the culture medium or in vivo (J Med Chem. 2009; 52:5188–5196) (J Pharmacol Exp Ther. 2014; 349:138–154). Thus, TP-10 has often been used at much higher concentrations than its IC_{50} (e.g., ≈ 3 mg/kg in mice) in previous studies (J Pharmacol Exp Ther. 2008; 325:681–690)(Circulation. 2020; 141:217–233.). Therefore, the effect of free TP-10 in this study is very likely underestimated due to a low concentration.*

Response: We are appreciated that you mentioned us with this valuable issue. We agreed that this is an important aspect for clinics and the effect of free TP-10 may be underestimated in our in vivo experiments. However, in another sense, these results also manifested the excellent efficacy of our CTP modified CaP nanoparticle-mediated delivery strategy. The CTP modification together with inhalation delivery significantly improved the accumulation of TP-10 in the diseased myocardium. Benefit from this advantage of the delivery strategy, TP-10 can effectively reverse the pathological process of chronic heart failure in such a low dosage.

As shown in **Fig. 6** and **Fig. 7**, the therapeutic effects of free TP-10 and TP-10@CaP are similar. However, based on the previous study (Sci Transl Med. 2018 Jan 17;10(424): eaan6205.), inhalation delivery of CaP loaded with LTCC mimetic peptide (MP) for the treatment of diabetic cardiomyopathy showed much better effect than that in inhalation delivery free MP. Generally speaking, the therapeutic effect of TP-10@CaP should be better than that of free TP-10 due to the blessing of delivery system. But our in vivo evaluations shown the different results. This controversy may be as stated by the reviewer stated, that is, the level of TP-10 in myocardium from TP-10@CaP or free TP-10, were far from the minimal effective concentrations due to the highly protein-bound characteristic of TP-10 in vivo. The majority of TP-10 was

bound with protein in the circulation and myocardium. In contrast, TP-10 delivery by the designed TP-10@CaP-CTP promoted much more TP-10 accumulated in the myocardium, which may contribute to achieve the effective concentration.

In the newly added experiments, we also verify the effects of different administration dosage of TP-10 on heart failure. Although inhalation treatment of TP-10 in the dosage of 3 mg/kg/2 day can partially ameliorate heart failure, the effects of inhalation treatment of 1.5 mg/kg TP-10 or 3 mg/kg TP-10 are both inferior to that of inhalation of 2.5 mg/kg TP-10@CaP-CTP (the amount of TP-10 is around 0.75mg) on preventing the pathological process of heart failure and improving cardiac function (Supplementary Fig. 18).

Based on the above analysis, we think the usage of low dosage of free TP-10 (0.75 mg/kg TP-10) might not influence the conclusion that the TP-10@CaP-CTP nanomedicine developed in this study provides a promising strategy for long-term management of chronic heart failure with enhanced cardiopulmonary circulation delivery efficiency and therapeutic efficacy while minimizing unintended lung complication of inhalation delivery. We have added this context to the revised manuscript (Page 12, Line 346-351, and Page 16, Line 440-447) and revised supplementary materials (Page 22-23).

Supplementary Fig 18. The effect of inhalation treatment of different dosage of free TP-10 in pressure overload induced heart failure mouse model.

Response to Reviewer #3 (Remarks to the Author)

The authors have generated a large amount of new data and addressed most of my initial concerns. In this new format, the study provides an interesting documentation for the use of inhalation delivery in the long-term management of chronic heart failure. However, I was still puzzled by a few points, as described below.

Response: We thank the reviewer for recognizing the potential of our work and providing constructive feedback. Below are our detailed responses to your comments.

Q1. *In the abstract, line 10 and elsewhere, the authors refer to a low dosage of the treatment, but it is not clear whether they are referring to the actual active ingredient, the nanoparticle, or something else. Please clarify this, and if possible, it would be interesting to refer to a relevant published study for comparison. For example, mentioning the TP dose used in the study from reference 23 (lines 116-120) would be of great help. This would also support other statements such as those in lines 107-111.*

Response: We thank for your kindly suggestion. The “low dosage” here was refers to the dosage of TP-10@CaP-CTP (2.5mg/kg/2 days) that compared to the high dosage of 7.5mg/kg/2 days. To avoid confusion, we have followed your suggestion and clarified this in the revised manuscript (**Page 2, Line 46, Page 12, Line 350-351**). In addition, we also added the content of comparison of the dosage in our study with the previous study in the revised manuscripts (**Page 15, Line 433-436, Ref. 23**).

“In addition, the dosage of TP-10@CaP-CTP nanoparticles inhaled in this study (2.5 mg/kg/2 days, with a content of TP-10 that is equivalent to approximately 0.75 mg/kg/2 days) was much lower than that used in a previous study (3.2 mg/kg/day TP-10 for the prevention of heart failure by I.V. injection)²³”

Q2. *Line 95. With reference to the statement “However, the lung complications which greatly limit the wide application of inhalation route clinically, are still need to be explored”, the author might consider to discuss a new recent publication (Alogna et al., JACC 2024 Lung-to-Heart Nano-in-Micro Peptide Promotes Cardiac Recovery in a Pig Model of Chronic Heart Failure) where the very same CaPs (engineered in a dry powder of microparticles containing drug-loaded CaPs) were tested in a (1 week long) dose range finding study in rats and no adverse effects at any of the increasing doses were reported.*

Response: Thanks. We are aware of the suggested study by Alogna et al. and have now cited it in the revised manuscript (ref 43). Alogna et al systematically evaluated the lung condition after two weeks of daily inhalation treatment and demonstrated that dry powder of microparticles containing drug-loaded CaPs could reduce lung congestion and damage. Furthermore, in the one-week daily inhalation toxicity study, they verified the safety profile of the dpCaP nanoparticles. Also, as Alogna et al. put it in the section of “STUDY LIMITATIONS”: *....the treatment in the present study was administered for a short time and therefore did not allow investigating efficacy and safety of a chronic regimen.* Compared with their study, we have a longer monitoring period (18 weeks) on lung status. Our data indicated that long-term inhalation

treatment at a low dose (2.5 mg/kg/2 days for mice) or short-medium term inhalation treatment at a high dose (7.5 mg/kg/2 days for mice) might not lead to lung injury over inhalation treatment. We have added this discussion in the revised manuscript (**Page 17, line 467-470**). Also, we have rephrased the sentence (**Page 3, Line 88-89**).

“However, the potential lung complications of chronic regimen, which may limit the wide application of drug administration via inhalation clinically, still need to be explored”.

Q3. *Line 107-11. I believe the sentence/message needs to be rephrased to better convey the intended message. In fact, I expect the potential pulmonary complication to be similar whether one uses the same amount of CaP or the CaP-CTP nanoformulation. In fact, I see no reason why CTP would reduce the potential pulmonary complication that CaP might have, unless the authors believe that coating CaP with CTP is expected to provide some "tox-reduction" effect in the lung. In my opinion, the only added value of CTP is expected systemically by influencing the biodistribution towards an enriched accumulation in the heart, thus reducing the required therapeutic dose to be administered.*

Response: Thanks for your valuable question. We agree that the side effect of CaP or the CaP-CTP nanoformulation on lung may be similar. Based on the existing literatures, we know that calcium phosphate nanoparticles have prominent biocompatibility, and in our study, we also verified its biosafety by a series of vitro and vivo experiments. However, some study also found that calcium phosphate nanoparticles lead to cell apoptosis which might be induced by Ca²⁺ released (Curr Pharm Des. 2017;23(20):2930-2951.). The Ca²⁺ released from calcium phosphate nanoparticles could induce human hepatoma cells death (World J Gastroenterol 2003; 9: 1968-71.), vascular smooth muscle cells death (Circ Res. 2008 Aug 29;103(5): e28-34.) and so on.

Thus, to further confirm the safety of our designed CaP nanoformulation in high dosage (7.5 mg/kg/2 days), we followed your suggestion in Q21 and performed relative experiments. We harvested the lung tissue after six weeks of inhalation treatment with different nanoformulations. We demonstrated that both inhalation treatment of TP-10@CaP in the dosage of 7.5 mg/kg/2 days and TP-10@CaP-CTP in the dosage of 2.5 mg/kg/2 days can ameliorate pulmonary congestion and injury, which induced by chronic heart failure (**Supplementary Fig. 20**). Based on these data from us and other research groups (Sci Transl Med 10, ean6205. (2018), J Am Coll Cardiol 83, 47-59 (2024)), we believe that even in a high dosage (like 7.5 mg/kg/2 days), inhalation treatment of the CaP or the CaP-CTP nanoformulation are safety for lung during the short- to medium-term inhalation treatment.

Therefore, we would like to replace the results of the long-term evaluation of high dosage of TP-10@CaP-CTP on lung safety with the results of the condition of the lung after six-week of inhalation treatment. The relative results and discussion were added in the revised manuscript (**Page 14, Line 394-400** and **Page 17, 475-479**) and revised supplementary materials (**Page 26, Supplementary Fig. 20**).

Moreover, to avoid confusion, we have rephrased the sentence (**Page 4, Line 97-102**).

“Although the potential binding mechanism remains unclear.....Therefore, we hypothesized that the CTP modification strategy to enable heart-targeted drug delivery to improve the accumulation of therapeutic agents in the heart rather than the administration of a high dose or multiple-doses of each inhalation medication, thus reducing the required therapeutic dose to be administered.”

Supplementary Figure 20. Lung safety verification after six weeks of inhalation treatment in TAC mice.

a, Representative images of H&E and Masson staining of the lung sections after six weeks of inhalation treatment in TAC mice. n=6. b, The lung W/D ratio, c, Quantitative assessment of BALF protein concentrations, and d-f, BALF levels of IL-1β, IL-6, and TNF-α. n=6. The results are presented as the mean ± SD. # P > 0.05. For b-f, statistical analysis was performed using one-way ANOVA with the Bonferroni multiple comparison correction.

Q4. Line 129-132. Please rephrase. The molecular mechanism of action of TP-10 has already been explored/elucidated by other studies and here I would recommend using different wording, as the same authors write later in the discussion (line 532-535).

Response: We apologize for the confusion caused due to the insufficient explanation. We have rephrased the sentence according to your nice suggestion as follows. However, we know TP-10 is a novel selective inhibitor of PDE10A. Previous study found that TP-10 can regulate intracellular cAMP and cGMP levels in cardiomyocytes and cardiac fibroblasts. Besides, it was demonstrated that TP-10 contributes to DOX induced cardiomyocyte death and mitochondrial dysfunction via cGMP/PKG/Top2β signaling and contributes to DOX induced cardiomyocyte atrophy via both cAMP/PKA and cGMP/PKG signaling (Circ Res. 2023 Jul 7;133(2):138-157.). Thus,

the molecular mechanism of action of TP-10 on the prevention of DOX-induced cardiotoxicity was explored. However, the detail mechanism of TP-10 on heart failure still far from elucidated. We only know that TP-10 might contribute to heart failure by regulating cAMP and cGMP level in CMs and CFs (Circulation. 2020 Jan 21;141(3):217-233.). Whereas, which signaling pathways implicates in the HF-prevention effects of TP-10, still unknown. And our study initially explored the potential mechanism of TP-10 on prevention of HF. Maybe there are some misunderstands in these sentences in Line 129-132, we have rephrased these sentences and made them clearer (**now in Page 4, Line 115-121**).

“The TP-10@CaP-CTP nanoparticles also regulated intracellular cAMP and cGMP levels in both CMs and CFs. Exploiting the heart-targeting capacity of CTP, the inhaled TP-10@CaP-CTP nanoparticles effectively attenuated pathological cardiac hypertrophy and cardiac fibrosis remodelling, thereby improving cardiac function in a pressure overload-induced heart failure mouse model. We elucidated that TP-10@CaP-CTP might contribute to CM pathological hypertrophy via cAMP/AMPK signalling and inhibit CF activation in a cGMP/PKG-dependent manner.”

Q5. *Fig. 1b and Supplementary Fig. 1, please mention when post-synthesis TEM analysis was performed.*

Response: We have added more detailed descriptions of the TEM analysis of different nanoparticles in the revised manuscript (please refer to the captions of **Fig. 2** in the main text).

Q6. *Fig2a,d,e,f,g,h. Please correct the positioning of the panels (columns) by adopting the same order when the group is presented in sham and TAC (i.e. blank, TP-10@CaP-CTP, TP-10@CaP).*

Response: Thanks. We have corrected the positioning of the panels by adopting the same order (now **Fig. 3** in the revised manuscript).

Fig. 3. Heart targeting capacity of TP-10@CaP-CTP at the physiological, histological and cellular levels.

Q7. Fig 2d, labeling/description is missing for all panels. Are different time points per group shown, or slices of heart or time points? Please clarify.

Response: Thank you for this suggestion. We have added relative labels for all panels (see Fig. 3 attached above).

Q8. Fig. 2b: Please indicate in the figure or legend which graph is for the heart or lung.

Most importantly, it would be relevant to show from the fluorescence study a comparative percentage of dye accumulation from different organs. Such data would then facilitate the comparison with those obtained from the TP analyses in Supplementary Table 1, providing a nice biodistribution study based on either nanoparticle (Cy5.5) or drug (TP-10) tracking.

Response: We thank for suggesting this detail. We have now added the relevant labels in Fig. 2b (now Fig. 3b). In addition, based on your suggestion, the ex vivo fluorescence imaging for observing the biodistribution of Cy5.5-labelled TP-10@CaP-CTP, TP-10@CaP and free Cy5.5 after administrated via inhalation in various organs was also performed (Supplementary Fig. 11). As expected, TAC mice received TP-10@CaP-CTP showed highest fluorescence signal in the heart compared to mice received TP-10@CaP or free Cy5.5. Besides, mice received TP-10@CaP-CTP showed lower fluorescence signal in the major extra-pulmonary organs. These results were consisted with the findings of the TP-10 level in different organs (Supplementary Table 1), which was measured by HPLC-ESI-MS/MS. Although there are differences in quantitative standards between the two methods, the trend of biodistribution remains consistent. These results further demonstrated that the specific heart targeting capacity of TP-10@CaP-CTP nanoparticle. We have now added the results and discussion of this matter in the revised manuscript (Page 9, Line 237-239) and revised supplementary materials (Page 14).

Supplementary Fig. 11. Quantitative assessment of Cy5.5 fluorescence intensities of various organs from mice after inhalation treatment with Cy5.5-labelled TP-10@CaP-CTP, TP-10@CaP and free Cy5.5. Tissues were harvested at 1 h after inhalation treatment and subjected to ex vivo fluorescent imaging detection (n=3).

Q9. Line 254-258, The TP alone control is missing in TP quantification from the biodistribution study.

Response: Thanks for your valuable suggestion. Per your suggestion, we tested the TP-10 levels after the free TP-10 treatment. The biodistribution data of TP-10 in myocardial and other major organs in TP-10 control group was available in

Supplementary Table 1. As expected, the myocardial TP-10 level in TP-10 control group was significantly lower than that in TP-10@CaP-CTP group. These results are also added in the revised manuscript (**Page 9, Line 243**) and supplementary materials (**Page 33**)

Q10. *Supplementary Fig. 12c, d. It is not clear to me why the data of TP-10@CaP in -PE conditions are not shown. Please clarify or show. In fact, it would be nice to see the two CaP preparations (i.e. +/- CTP) in +/-PE in the same densitometry plot.*

Response: Thank you for this valuable suggestion. We have now added the data of TP-10@CaP in -PE conditions and TP-10@CaP-CTP in +PE conditions in this experiment (**now presented in Fig. 4a, b**). There was no significant difference of nanoparticle uptake in TP-10@CaP treated PBS or PE induced NRVMs and TP-10@CaP-CTP treated normal NRVMs. Consisted with the data in **Supplementary Fig. 12**, TP-10@CaP-CTP treated and PE induced NRVMs group showed highest nanoparticle uptake behavior. These results and discussion are added now in the revised manuscript (**Page 9, line 257-260**).

Fig. 4a, Representative images of the intracellular uptake of Cy5.5-labelled TP-10@CaP and TP-10@CaP-CTP in PBS- or PE-induced (pretreated with 100 μM PE for 24 h) neonatal rat ventricular myocytes (NRVMs). Scale bar, 50 μm. **b,** Quantitative analysis of fluorescence signals from NRVMs. n = 3.

Q11. *Supplementary Fig. 13. Please include the same results also for TP-10@CaP, which should be presented as a comparative control in each assay performed. In fact, in the current study, the authors do implement the CaP data from the literature with the CTP targeting peptide.*

Response: Thanks. We have now added the data of TP-10@CaP group in **Supplementary Fig. 13**. As shown in the immunofluorescence images, more TP-10@CaP-CTP nanoparticles were accumulated in the myocardium than that of TP-10@CaP nanoparticles.

Supplementary Fig. 13. The accumulation of Cy5.5-labelled TP-10@CaP or TP-10@CaP-CTP nanoparticles in the fibrotic area of heart.

Q12. *Supplementary Fig. 14, 23. Please, include the TP-10@CaP control in PBS condition.*

Response: Thanks for your valuable suggestion. We have now added the data of TP-10@CaP control in PBS group in Supplementary Fig. 14 and 23 (**now Supplementary Fig. 14 and Fig. 24 in the supplementary materials**). TP-10@CaP-CTP treated TGF- β induced CFs presented significantly more nanoparticles accumulation. By contrast, no overt difference was observed on other treatment groups (**Supplementary Fig. 14**), this indicated that CTP might also target to the pathological state of CFs. Meanwhile, the immunofluorescence imaging show that there were no significant differences of nanoparticle accumulation in MLFs

(Supplementary Fig. 24), which is a type of pulmonary fibroblasts cells, indicating pulmonary fibroblasts cells were not targeted by this CTP-modified nanomedicine.

Supplementary Fig. 14. In vitro determination of targeting capacity of TP-10@CaP and TP-10@CaP-CTP in AMCFs.

Supplementary Fig. 24. In vitro determination of targeting capacity of TP-10@CaP and TP-10@CaP-CTP in MLFs.

Q13. Supplementary Fig. 18. Please add symbols for statistical analysis in graph and relative description in legend.

Response: Thanks for your kindly suggestion. We have added the symbols and relative description in the revised supplementary materials (see below, now **Supplementary Fig. 16, Page 19**).

Supplementary Fig. 16. Consecutive echocardiographic and plasma ANP levels monitoring in mice.

When TAC+TP-10@CaP-CTP vs TAC+PBS group, *** represents $P < 0.05$, **, $P < 0.01$, ***, $P < 0.001$.

When TAC+TP-10@CaP-CTP vs TAC+TP-10 group, # represents $P < 0.05$, ##, $P < 0.01$, ###, $P < 0.001$.

When TAC+TP-10@CaP-CTP vs TAC+TP-10@CaP group, † represents $P < 0.05$, ††, $P < 0.01$, †††, $P < 0.001$.

When TAC+PBS vs Sham group, ‡ represents $P < 0.05$, ‡‡, $P < 0.01$, ‡‡‡, $P < 0.001$.

When TAC+PBS vs TAC+ CaP-CTP group, ★ represents $P < 0.05$, ★★, $P < 0.01$, ★★★, $P < 0.001$.

Q14. Supplementary Table 1. Data from inhaled free TP-10 were not included, but represent an important control. In particular, because free TP-10 was used in the therapeutic experiment.

Response: Related to our response to Q9 of yours, we tested the TP-10 levels in myocardial and other major organs after the free TP-10 treatment in **Supplementary Table 1 (Page 33 in supplementary materials)**.

Q15. In Supplementary Figures 22 and 23, the data are expressed as %Cy5.5 per field, whereas in Supplementary Figures 12 and 14, the data are expressed as %Cy5.5 per cell. It would be better to use the same method of data presentation to allow some sort of comparative analysis between experiments.

Response: Thanks for spotting the discrepancy. We apologize for the confusion caused by our careless of a mistaken descriptions of Supplementary Figures 22 and 23 in the original manuscript. We have now provided the correct figure (total fluorescence intensity/numbers of cell) and matching descriptions in the revised manuscript (please refer to its caption in the main text). The revised figures were shown in the revised supplementary materials (**now Supplementary Figures 23 and 24**).

Q16. Line 474-477. Still do not understand the rationale for including this information. CTP was used in this paper to enrich the delivery of the nanoparticle to cardiomyocytes (and cardiac fibroblasts) by a possible binding to an unidentified membrane receptor that is enriched on such cells. Even if CTP can bind to alpha-b-crystallin due to homology to cardiac titin, what is the relevance for extracellular targeting since both alpha-b-crystallin and titin are cytosolic? Sentences 518-524 I think are better for the topic.

Response: Thanks for your valuable suggestion. We apologize for the confuse, our original intention is to discuss the bind mechanism of CTP. Kanki, S. et al proposed the hypothesis that CTP might bind to alpha-B crystalline, but they finally failure to identify it. Thus, the bind mechanism still be unclear. In our study, we found that CTP can bind to cardiac myocytes and fibroblasts rather than pulmonary epithelial cells and fibroblasts. However, the detail mechanism of this phenomenon required more experiments to explore it. Due to the subsequent description, we have also continued to discuss this topic, and rephrase the sentences in Line 474-477. (**now shown in Page 16, Line 458-463**).

Q17. Supplementary Fig. 1: Why a different scale bar is used in b (100 nm) compared to a, c (200 nm).

Response: Thanks for the comment. Different scales are caused by the different testing times of the samples. We have now unified them in Supplementary Fig. 1 (**now Fig. 2b, see below**) for clearer demonstration of the TEM images in the revised

manuscript to avoid confusion.

Fig. 2. TEM image of different intermediate product, including a, TP-10@CaP, b, TP-10@CaP-DSPE-PEG, and c, TP-10@CaP-CTP nanoparticles collected at 12 h after preparation.

Q18. Please add references to the studies mentioned in lines 104-107 that refer to “drug delivery carriers that were modified by CTP”.

Response: Thanks for your kindly notice. We have added the references in revised manuscript (**Page 4, Line 99**).

21. Wang, X. et al. Engineered Exosomes With Ischemic Myocardium-Targeting Peptide for Targeted Therapy in Myocardial Infarction. *J Am Heart Assoc* 7, e008737 (2018).

22. Kanki, S. et al. Identification of targeting peptides for ischemic myocardium by in vivo phage display. *J Mol Cell Cardiol* 50, 841-848 (2011).

Q19. Supplementary Fig. 18 should be renumbered as Supplementary Fig. 16 as presented earlier than the actual Supplementary Fig. 16-17.

Response: Thanks for you kindly suggestion. We have revised the order of **Supplementary Fig. 16-19** in the revised manuscript (**Page 12, 13**) and supplementary materials (**Page 19-25**).

Q20. Ref24 is not correct to be included in line 628 as no TAC is performed on this paper.

Response: Thanks. The references in line 628 are 23, 34 and 35, not 24 (**now are 23, 46, 47**). TAC is performed in these papers.

Q21. Line 408-413- Do the authors have any information from the literature on the toxicity data of TP-10 or CTP? Do the authors propose that the observed effect is a phenomenon due to one of the components or the entire formulation? Although performed in TAC mice, did the authors observe any incremental lung issue with the

7.5 mg/kg/2 days dose regimen of TP-10@CaP as used in the tox assay of Supplementary Figure 19 where TP-10@CaP-CTP has been used?

Response: Thanks for this valuable question. Firstly, we do not find any evidences about the toxicity data of TP-10 or CTP in previous studies. Based on the existing literatures about the calcium phosphate nanoparticles, we speculated this phenomenon may be due to the entire formulation, and more likely to be related with the excessive Ca^{2+} release from the nanoparticles. However, we have performed the animal experiment for observing lung condition after inhalation treatment on TAC mice. As shown in **Supplementary Fig. 20**, both inhalation treatment of TP-10@CaP in the dosage of 7.5 mg/kg/2 days and TP-10@CaP-CTP in the dosage of 2.5 mg/kg/2 days can ameliorate pulmonary congestion and injury, which induced by chronic heart failure. More specific discussions have been provided in our response to Q3 before, and we summarized that high dosage of our designed TP-10@CaP-CTP and other CaP-based formulations after inhalation is safe in a short-medium term. However, for long-term usage of inhalable formulations, a low dosage may be one of the conditions to ensure safety.

we have replaced the results of the long-term evaluation of high dosage of TP-10@CaP-CTP on lung safety with the results of the condition of the lung after six-week of inhalation treatment. The relative results and discussion were added in the revised manuscript (**Page 14, Line 394-400** and **Page 17, 475-479**) and revised supplementary materials (**Page 26, Supplementary Fig. 20**).

Supplementary Figure 20. Lung safety verification after six weeks of inhalation treatment in TAC mice.

Q22. Finally, I suggest a thorough revision of English grammar by a native speaker.

Response: Thanks for your kind suggestion. The text has been thoroughly revised for conciseness and clarity, and grammar/spelling mistakes are carefully corrected. Also, we have repolished the language in the revised version by the Nature Editing Services (Fig. R1).

SPRINGER NATURE | Author Services

March 27, 2024

Dear Haobo Weng,

Thank you for choosing Springer Nature Author Services. This manuscript, titled "An inhaled cardiac-targeting peptide-modified nanomedicine prevents pressure overload-related heart failure based on cardiopulmonary circulation," is very interesting. The paper was edited for grammar, phrasing, and punctuation. In addition, many edits were made to further improve the flow and readability of the text. Below, we highlight the areas of this paper that we focused on in our edit.

In cases where the meaning of the text was not clear, revisions were made to convey the information with increased clarity and reduced ambiguity.

Some edits were made to improve conciseness by trimming unnecessary words and streamlining the flow of your manuscript.

Articles are an important aspect of the English language, including the definite article "the" and the indefinite articles "a" and "an." Our edits focused on improving article use, which is often strongly dependent on context and field conventions.

Comments were left if further clarification would be helpful or confirmation of the meaning of the text was necessary. Please review these comments and all our changes carefully for more detailed suggestions, as well as to ensure that the final version of the manuscript is fully accurate.

Thank you again for using our editing services; we wish you the best of luck with your submission.

Best regards,

Beth K.
Senior Editor
Springer Nature Author Services

Figure R1. English editing certificate of our paper.

REVIEWER COMMENTS

Reviewer #1 (Remarks to the Author):

I appreciate the efforts made by the authors in addressing my concerns. Nearly all of them have been clarified in this round of revision. However, I have one remaining comment regarding the XRD patterns of the samples. The patterns appear to be noisy, likely due to the low amount of material present. As a result, it is challenging to distinguish whether the samples are crystalline or amorphous. In my view, not all the samples are amorphous. To further analyze this aspect, I suggest examining the FTIR spectra of the samples and calculating the splitting factor of phosphate bands. The authors may find guidance on this method in the work of Di Mauro et al. (ref. 24). Following this adjustment, I believe the paper is ready for publication.

Reviewer #2 (Remarks to the Author):

This reviewer's concerns have been adequately addressed.

Reviewer #3 (Remarks to the Author):

The authors have addressed most of my concerns and I have no further questions. There is only one open question about the effective TP concentration in TP-10@CaP formulations without the targeting peptide. This is relevant information that is needed. In fact, with the latest revision, the effective TP concentration in nanoparticles was only included for the full TP-10@CaP-CTP formulation.

Responses to reviewers' comments

Please revise your manuscript, addressing all the remaining issues raised by the reviewers. In addition, please revise the title as this should not be longer than 15 words, and shorten the abstract to max 150-200 words.

Response: Thank you very much for this value suggestion. Below is a detailed description of the changes we have made and a point-by-point response to the reviewers' comments. Also, we have revised the title as "An inhaled cardiac-targeting peptide-modified nanomedicine prevents pressure overload-related heart failure based on cardiopulmonary circulation", and shorten the abstract to 200 words in the revised manuscript (Page 2).

Point-by-point responses to reviewers' comments

Response to Reviewer #1 (Remarks to the Author)

I appreciate the efforts made by the authors in addressing my concerns. Nearly all of them have been clarified in this round of revision. However, I have one remaining comment regarding the XRD patterns of the samples. The patterns appear to be noisy, likely due to the low amount of material present. As a result, it is challenging to distinguish whether the samples are crystalline or amorphous. In my view, not all the samples are amorphous. To further analyze this aspect, I suggest examining the FTIR spectra of the samples and calculating the splitting factor of phosphate bands. The authors may find guidance on this method in the work of Di Mauro et al. (ref. 24). Following this adjustment, I believe the paper is ready for publication.

Response: We would like to thank the reviewer for comprehensive feedback on improving the manuscript. We have thoroughly revised the manuscript for enhanced clarity and added extra experimental data for scientific rigor.

Regarding the crystallinity of products, we apologize for not being able to obtain satisfactory XRD patterns of the samples during multiple experiments. We thank the reviewer for the recommended study by Di Mauro et al. From the Supplementary Fig. 3 (enlarged images of Fig. 2f), it can be observed that a broad absorption band at approximately 602-618 cm^{-1} , indicating the amorphous of TP-10@CaP. In contrast, the absorption band in this region turned narrow and two bands at 602 and 618 cm^{-1} (ν_4 P-O vibrations of PO_4^{3-}) were appeared in the spectra of TP-10@CaP-DSPE-PEG and TP-10@CaP-CTP, which indicated that amorphous TP-10@CaP turned into a more crystalline state, following a surface modification process. Also, we have determined the splitting factor of these products to evaluate the crystallinity degree of CaP-based materials from FT-IR spectra. According to the calculation method reported by Di Mauro et al, we measured the sum of the heights of the valleys at 602 and 618 cm^{-1} and divided by the height of the peak between them at $\sim 609 \text{ cm}^{-1}$. For TP-10@CaP where a single broad peak exists, a value for the splitting function of zero is obtained as the heights of the valleys are equal to zero. The amorphous and crystalline states of these CaP-based materials were also confirmed by the measurement of splitting factors (**Supplementary Fig. 3**). These results have now

been included in the revised manuscript (Page 6, Line 162-167).

Supplementary Fig. 3. (a) FT-IR spectra of different prepared nanoparticles (enlarged image of Fig. 2f). (b) Splitting factors of different prepared nanoparticles was calculated by the sum of the heights of the valleys at 602 and 618 cm^{-1} and divided by the height of the peak between them at $\sim 609 \text{ cm}^{-1}$. All heights were measured above a baseline drawn from approximately 645-530 cm^{-1}).

Response to Reviewer #2 (Remarks to the Author)

This reviewer's concerns have been adequately addressed.

Response: Thanks very much for your recommendation of our work. Thanks again for the careful review and constructive feedback.

Response to Reviewer #3 (Remarks to the Author)

The authors have addressed most of my concerns and I have no further questions. There is only one open question about the effective TP concentration in TP-10@CaP formulations without the targeting peptide. This is relevant information that is needed. In fact, with the latest revision, the effective TP concentration in nanoparticles was only included for the full TP-10@CaP-CTP formulation.

Response: We would like to thank the reviewer for recognition of this research and useful comments for us to improve the scientific rigor of the manuscript.

Since TP-10@CaP-CTP were synthesized based on TP-10@CaP, followed by the surface functionalization and modification, we guarantee to use the same TP-10 amount for the preparation of TP-10@CaP nanoparticles to ensure consistency. We have now added the effective TP-10 content in TP-10@CaP formulations (57.6%) in the revised manuscript (**Page 7, Line 180-184**). It should be noted that we synthesized TP-10@CaP with a drug loading of about 37.2% for subsequent animal experiments, which is mainly to maintain consistency with TP-10@CaP-CTP at the dosage level of monolithic structure (2.5 mg/kg).

REVIEWERS' COMMENTS

Reviewer #1 (Remarks to the Author):

The authors have satisfactorily addressed all of my concerns.